# MidSteer: Optimal Affine Framework for Steering Generative Models

Tatiana Gaintseva [* 1 2]   Andrew Stepanov [* 3]   Ziquan Liu [1]   Martin Benning [4]   Gregory Slabaugh [1]
Jiankang Deng [5]   Ismail Elezi [2]

## Abstract

Steering intermediate representations has emerged as a powerful strategy for controlling generative models, particularly in post-deployment alignment and safety settings. However, despite its empirical success, it currently lacks a comprehensive theoretical framework. In this paper, we bridge this gap by formalizing the theory of concept steering. First, we establish a link between steering and affine concept erasure, proving that the standard approach for removing unwanted behaviors is a special case of LEACE (a closed-form method for affine erasure). Next, we formulate a principled theoretical framework for concept switching, LEACE-Switch, and characterize the assumptions under which it provides an optimal affine solution. Building on this analysis, we then introduce Mid-Steer (**Mi**nimal **D**isturbance concept **Steer**ing), a more general affine framework for concept manipulation that relaxes these assumptions and enables directed, minimal-disturbance transformations. We demonstrate that MidSteer performs favorably across a range of tasks, modalities, and architectures, including vision diffusion models and large language models.

## 1. Introduction

Generative models such as Large Language Models (LLMs) and vision diffusion models have achieved remarkable progress in recent years (Yang et al., 2024b) (Naveed et al., 2023). However, controlling model outputs to enforce desirable behaviors or suppress harmful ones remains challenging (Bartoszcze et al., 2025). Yet, this capability is necessary for improving model safety, reliability, alignment, and usefulness in downstream applications.

Concept steering of intermediate representations is an increasingly popular technique that has already proven simple yet powerful for controlling behavior in LLMs (Panickssery et al., 2024). Recently it was also shown to be applicable to vision diffusion models (Gaintseva et al., 2026). It changes the intermediate representations of a generative model during generation by adding a "steering vector" that encodes a target concept. This approach has proven effective for tasks such as erasing unwanted behaviors (toxicity, nudity) or amplifying desirable features (helpfulness, truthfulness) (Panickssery et al., 2024; Singh et al., 2024; Gaintseva et al., 2026). However, despite its simplicity, its theoretical foundations remain underdeveloped, with most work being highly empirical (Zou et al., 2023; Wehner et al., 2025). Existing methods largely rely on heuristic vector manipulations, which can introduce unintended side effects and lack a solid theoretical basis and guarantees (Raedler et al., 2025; Anthropic, 2024). Moreover, naive steering often perturbs unrelated features, undermining the minimal disturbance principle that is critical to maintaining model quality and coherence.

Recently, strong theoretical foundations have been developed for concept erasure. (Ravfogel et al., 2023) introduced the notion of log-linear guardedness. Based on it, (Belrose et al., 2025) developed LEACE, an affine concept erasure framework to remove undesired information from model representations for downstream tasks. However, these methods do not naturally extend to other forms of concept manipulation, such as switching, where the goal is to replace one concept with another rather than just erasing it.

In this work, we address these gaps by developing a unified theoretical framework for affine steering of generative models. We begin by proving a formal equivalence between the standard steering methodology and LEACE, demonstrating that the widely used heuristic of steering for concept erasure is a special case of optimal affine concept erasure. Building on this foundation, we extend the framework from erasure to concept switching. We first consider the setting of bidirectional switching, in which a binary concept partitions the dataset and the goal is to invert the linear dependence

[1]Queen Mary University of London, London, UK [2]Huawei Noah's Ark, London, UK [3]QuantumLight Capital, London, UK [4]University College London, London, UK [5]Imperial College London, London, UK. Correspondence to: Tatiana Gaintseva <t.gaintseva@qmul.ac.uk>.

*Proceedings of the 43rd International Conference on Machine Learning*, Seoul, South Korea. PMLR 306, 2026. Copyright 2026 by the author(s).

of the representation on the concept label. Under these assumptions, we derive LEACE-Switch, an optimal affine transformation that performs a complete and symmetric concept swap while minimally disturbing the representation.

We examine the scope of this formulation and show that its assumptions, dataset partitioning and global label inversion define a practically relevant, though restricted, regime. We address more general settings in which the concepts involved do not jointly span the entire dataset, or where asymmetric, one-directional transformations are desired. We introduce MidSteer (Minimal Disturbance Concept Steering), a generalized framework for affine concept manipulation that enables precise switching while minimizing interference with unrelated properties of the representation.

Through experiments with LLMs and vision diffusion models, we demonstrate that LEACE-Switch and MidSteer achieve more reliable concept switching than vanilla steering, allowing controllable generation with minimal side effects. Our results highlight the value of grounding steering methods in theory and provide practical tools for aligning generative models with desired behaviors.

In summary, our **contributions** are as follows:

- We establish a **formal theoretical connection** between standard activation steering and affine concept erasure, showing that commonly used steering heuristics for concept erasure are special cases of LEACE.

- We **extend the affine erasure framework to concept switching** and introduce **LEACE-Switch**, an optimal affine formulation for concept swapping under the assumption that a binary concept partitions the dataset.

- We then relax the dataset partitioning and symmetry requirements for the task of concept switching, and introduce **MidSteer** (Minimal Disturbance Concept Steering), a **generalized affine framework for concept manipulation** that enables precise and directed concept switching with provably minimal interference to unrelated representation components.

- We **empirically validate** LEACE-Switch and MidSteer across modalities and architectures, including LLMs and vision diffusion models, demonstrating improved controllability and reduced side effects compared to existing steering and erasure methods.

## 2. Related Work

**Activation steering and representation manipulation**. Activation steering has emerged as a lightweight approach for controlling generative models by modifying intermediate representations, particularly in LLMs (Turner et al., 2023) (Bartoszcze et al., 2025) (Rimsky et al., 2024) and more recently in diffusion models (Tumanyan et al., 2023) (Kwon et al., 2023) (Gaintseva et al., 2026). Most existing methods rely on heuristic vector addition, often derived from mean activation differences, and provide limited guarantees on optimality or side effects. Our work formalizes steering as an affine transformation problem and studies when such interventions are provably minimal and well-posed.

**Affine concept erasure and guardedness** A related line of work focuses on removing concept information from representations. INLP (Ravfogel et al., 2020) and RLACE (Ravfogel et al., 2023) iteratively project out linear subspaces predictive of a protected attribute. LEACE (Belrose et al., 2025) provides a closed-form affine solution for optimal linear concept erasure under the guardedness framework, minimizing representational disturbance while enforcing zero covariance with the concept label. Our work builds directly on this theory, showing that standard erasure-mode steering is a special case of LEACE, and extending the framework beyond erasure. Next, SPLINCE (Holstege et al., 2025) study oblique projections that erase protected attributes while preserving task-relevant subspaces. While these methods address constrained erasure, they do not consider concept switching, which requires translating concept dependence rather than erasing it. Nevertheless, they motivate the importance of minimal-disturbance objectives with structural constraints, which our work addresses in the switching setting.

**Distributional alignment and representation surgery**. Representation Surgery (Singh et al., 2024) derives affine transformations that match class-conditional means, and optionally covariances, between source and target distributions. This approach performs distributional alignment under Gaussian assumptions and is well-suited to tasks where class-conditional statistics fully characterize the desired transformation. In contrast, MidSteer operates on cross-covariances between representations and concept indicators, preserving the global linear structure of the representation space and explicitly minimizing changes outside the concept-mediating subspace. As a result, MidSteer targets concept switching rather than full distribution matching, and the two approaches optimize distinct objectives.

**In summary**, while prior methods address erasure or distributional alignment, our work is the first to formalize concept switching as a distinct affine problem and to derive closed-form solutions with explicit minimal-disturbance guarantees across modalities.

## 3. Preliminaries

### 3.1. Steering internal representations of models

We formalize **activation steering** as the manipulation of internal model representations to control the presence of a

specific concept $c$ in the model's output. This is achieved by adding a scaled steering vector $s_c$ to the intermediate hidden activity $h$ during inference. The **steering vector** $s_c$ is constructed from the concept-conditional means of the hidden activity. Let $h$ be a random vector in $\mathbb{R}^d$ representing the activity at a particular layer, and let $C \in \{0, 1\}$ represent the presence or absence of concept $c$. The steering vector $s_c \in \mathbb{R}^d$ is defined as the difference of these means:

$$s_c = \mathbb{E}[h|C = 1] - \mathbb{E}[h|C = 0], \tag{1}$$

$s_c$ can be optionally post-processed (to have unit norm).

The general steering intervention $f$ controls the concept's expressiveness via a scalar $\alpha \in \mathbb{R}$ representing the steering strength and direction. Omitting the subscript $c$, the operation is:

$$f(h, s) = h + \alpha s \tag{2}$$

We highlight two essential special cases, determined by the choice of the intervention function $f$.

**Concept Erasure.** It aims to erase all information aligned with the concept $c$ from the activation vector $h$. This is achieved by projecting $h$ onto the subspace orthogonal to the steering direction $s$. The projection $\langle h, s \rangle s$ estimates the conceptual component, which is then removed:

$$f_{\text{delete}}(h, s) = h - \langle h, s \rangle s \tag{3}$$

**Concept switch.** Multiplying projection by 2 results in a **Householder reflection** of the vector $h$ across the hyperplane orthogonal to $s$:

$$f_{\text{switch}}(h, s) = h - 2\langle h, s \rangle s \tag{4}$$

This transformation effectively substitutes the component of $h$ aligned with $c$ with its opposite, thereby substituting the representation of concept $c$ with the representation of its absence.

If steering is applied to specific layers (e.g., self-attention outputs in LLMs or cross-attention layers in vision models), both Equation 3 and Equation 4 can be incorporated directly into the model's weight matrices. This allows for achieving **zero inference overhead**, a critical advantage for deployment in large-scale applications.

### 3.2. Affine guardedness and covariance control

Our subsequent results are formulated as minimal-disturbance affine transformations under constraints on the cross-covariance between representations and concept indicators. We use the guardedness framework of Ravfogel et al. (2023); Belrose et al. (2025) to justify why cross-covariance is a meaningful object for formalizing linear concept information in representations. In particular, for the task of concept erasure under linear-affine predictors and squared

loss, Belrose et al. (2025) show that making a concept linearly unpredictable from a representation is equivalent to enforcing zero covariance between the representation and the concept label. Belrose et al. (2025) formulated the following notion of guardedness:

**Definition 3.1** (Guardedness, Belrose et al. 2025)**.** Consider a $k$-class classification task over jointly defined random vectors $X$ and $Z$, where $X$ takes values in $\mathbb{R}^d$ and $Z$ is a one-hot label vector taking values in

$$\mathcal{Z} = \{\mathbf{z} \in \{0, 1\}^k \mid \|\mathbf{z}\|_1 = 1\},$$

with $\mathbb{P}(Z = j) > 0$ for each class $j$. Let $\eta(\cdot; \theta) : \mathbb{R}^d \to \mathbb{R}^k$ be a predictor from a function class

$$\mathcal{V} = \{\eta(\cdot; \theta) \mid \theta \in \Theta\},$$

assumed to contain all constant functions, and let $\mathfrak{L}$ be a class of losses $\mathcal{L} : \mathbb{R}^k \times \mathcal{Z} \to [0, \infty)$. Let $\chi$ denote the set of random vectors of finite first moment taking values in $\mathbb{R}^d$ and jointly defined with $Z$.

We say that $X$ $(\mathcal{V}, \mathfrak{L})$-**guards** $Z$ if, for every $\mathcal{L} \in \mathfrak{L}$, it maximizes the minimum achievable prediction loss:

$$X \in \underset{X' \in \chi}{\arg\max} \inf_{\theta \in \Theta} \mathbb{E}\big[\mathcal{L}\big(\eta(X'; \theta), Z\big)\big].$$

Equivalently, $X$ is maximally uninformative about $Z$ for predictors in $\mathcal{V}$ and losses in $\mathfrak{L}$.

The full guardedness definition is general, but the key consequence for our purposes is its linear characterization. For linear–affine predictors with squared loss, guardedness is equivalent to zero cross-covariance between the representation and the label.

**Theorem 3.2** (Linear guardedness, Belrose et al. 2025)**.** *The following statements are equivalent:*

- *$X$ linearly guards $Z$ in the sense of Definition 3.1;*

- *every component of $X$ has zero covariance with every component of $Z$, i.e.*

$$\text{Cov}(X, Z) = 0.$$

Therefore, in the affine setting, concept erasure can be expressed as a covariance-control problem: find a transformation of the representation that removes linear dependence on the concept label. At the same time, to preserve unrelated information, the transformation should perturb the representation as little as possible. These two desiderata lead to the LEACE objective: among all affine maps that enforce $\text{Cov}(AX + b, Z) = 0$, choose the one closest to the identity transformation in expected squared distance.

**Theorem 3.3** (LEACE, Belrose et al. 2025). *Let $X$ and $Z$ be random vectors taking values in $\mathbb{R}^d$ and $\mathbb{R}^k$, respectively, each with finite second moments. Define $\Sigma_{XX} = \mathrm{Cov}(X, X) \in \mathbb{R}^{d \times d}$, $\Sigma_{XZ} = \mathrm{Cov}(X, Z) \in \mathbb{R}^{d \times k}$ and assume $\mathrm{Im}(\Sigma_{XZ}) \subseteq \mathrm{Im}(\Sigma_{XX})$. Then the optimization problem*

$$\min_{\substack{A \in \mathbb{R}^{d \times d} \\ b \in \mathbb{R}^d}} \mathbb{E}\left[\|AX + b - X\|_2^2\right] \quad s.t. \quad \mathrm{Cov}(AX+b, Z) = 0 \tag{5}$$

*has the following solution, almost surely:*

$$\widehat{A} = I - W^+(W\Sigma_{XZ})(W\Sigma_{XZ})^+ W, \tag{6}$$

$$\widehat{b} = \mathbb{E}[X] - \widehat{A}\,\mathbb{E}[X], \tag{7}$$

*where $W = (\Sigma_{XX}^{1/2})^+$ is the whitening transformation.*

Here and throughout, $A^+$ denotes the Moore–Penrose pseudo-inverse. For a positive semidefinite symmetric matrix $A = VSV^\top$, we write $A^{1/2} = VS^{1/2}V^\top$, where $S^{1/2}$ is obtained by taking square roots of the diagonal entries of $S$.

# 4. Theoretical Results

We theoretically analyze connections between steering approaches (Sec. 3.1) and LEACE (Sec. 3.2), and derive a novel framework for affine concept steering called MidSteer, that unifies and generalizes all these approaches.

This section is organized as follows. We first consider the task of concept erasure and establish a formal connection between the standard steering setup for erasure and LEACE, showing that Eq. 3 arises as a special case of the LEACE framework. We then extend LEACE to the task of optimal affine concept switching and derive **LEACE-Switch**, demonstrating that Eq. 4 is a special case of this bidirectional switching formulation.

Next, we characterize the assumptions under which LEACE-Switch provides an optimal solution and introduce **MidSteer**, a more general affine framework for concept manipulation that enables directed, minimal-disturbance transformations without requiring dataset-wide label inversion. Finally, through experiments on large language models and vision diffusion models, we show that MidSteer consistently outperforms vanilla steering and LEACE-based switching, achieving precise concept control while preserving unrelated features of the generated outputs.

## 4.1. Connection between steering in erasure mode and LEACE

In this section, we show that steering in erasure mode (Eq. 3) is a special case of LEACE. We formulate and prove a following corollary to the Theorem 3.3:

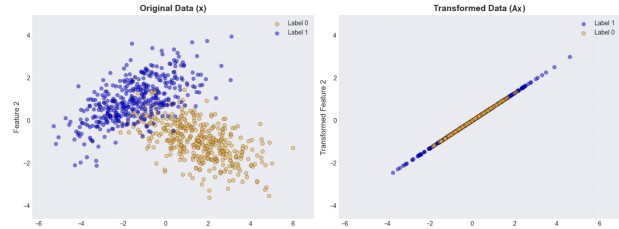

*(a)* Illustrative example of affine concept erasure. In this case the affine transformation is satisfying $\mathrm{cov}(AX + b, Z) = 0$. This figure is inspired by (Belrose et al., 2025).

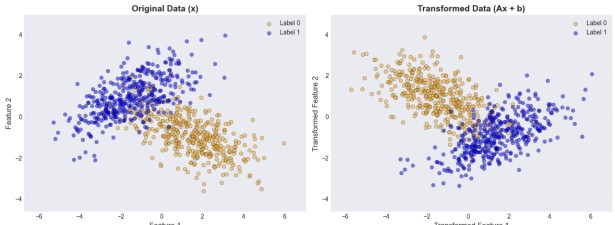

*(b)* Illustrative example of affine concept switching. In this case the affine transformation is satisfying $-\mathrm{cov}(AX + b, Z) = \mathrm{cov}(X, Z)$.

*Figure 1.* Illustrative example of affine concept erasure and affine concept flipping frameworks.

**Corollary 4.1.** *Let $X$ be a standardized random vector in $\mathbb{R}^d$, i.e., it has zero mean $E[x] = 0$ and unit covariance matrix $\Sigma_{XX} = I$. Let $C \in \{0, 1\}$ be a concept indicator variable. Let $s$ be defined as in Eq. 1. Let $f_{delete}$ be defined as in Eq. 3. Then $f_{delete}$ as a function of $h$ minimizes*

$$\min_{f \in \mathrm{Aff}(\mathbb{R}^d \mapsto \mathbb{R}^d)} \mathrm{E}[\|f(X) - X\|^2] \; s.t. \; \mathrm{Cov}(f(X), C) = 0 \tag{8}$$

This theorem states that steering in erasure mode can be seen as LEACE under the assumptions that the whitening matrix is the identity and that the mean of all vectors is zero. The proof is found in the Appendix D.1.

## 4.2. Concept switching

We generalize beyond a single binary concept, and consider the task of transforming the representation of one concept $c_1$ into another $c_2$. We call this task *concept switching*.

More formally, let $c_1$ and $c_2$ denote two distinct concepts, each associated with a subset of the data distribution (not necessarily jointly covering it), and their corresponding binary indicators $C_1, C_2 \in \{0, 1\}$. The goal of a *concept switch* operation is to construct a transformation $f$ such that samples exhibiting concept $c_1$ after transformation exhibit concept $c_2$, while preserving all other factors of variation as much as possible. Formally, this requires modifying the dependence of the representation on $C_1$ and introducing a

desired dependence on $C_2$, without enforcing any relationship between $C_1$ and $C_2$ outside their observed support.

We now adapt LEACE (Theorem 3.3) framework to the task of concept switching, and formulate a theoretical framework for the optimal affine concept switching, LEACE-Switch. Additionally, we show that steering in switching mode Eq. 4 can be seen as a special case of this framework. We then characterize the scope of LEACE-Switch by making explicit the assumptions under which it provides an optimal solution. Finally, we introduce MidSteer, a generalized affine framework for concept manipulation that relaxes these assumptions and enables directed, minimal-disturbance concept switching in more general settings.

### 4.2.1. LEACE FOR CONCEPT SWITCHING

Recall that $Z \in \{0,1\}^k$ denotes a binary concept vector. We now consider the task of *bidirectional concept switching*, in which the concept encoded by $Z$ is assumed to partition the dataset (i.e., $P(Z = 1) + P(Z = 0) = 1$), so that each sample belongs either to the concept or to its complement. Under this assumption, concept switching can be interpreted as a global inversion of the linear dependence between the representation and the concept label.

Within the covariance-based guardedness framework of LEACE, such an inversion admits a natural affine formulation. Rather than removing linear information about the concept as in concept erasure, we seek to preserve the magnitude of the linear dependence while reversing its sign. This leads to the following constraint:

$$\text{Cov}(f(X), \mathbf{1}^k - Z) = \text{Cov}(X, Z), \qquad (9)$$

where $\mathbf{1}^k$ denotes a $k$-dimensional vector of ones. By linearity of covariance, this condition is equivalent to

$$\text{Cov}(f(X), Z) = -\text{Cov}(X, Z). \qquad (10)$$

See Fig. 1b for a geometric illustration of this.

Equation 9 constitutes the affine analogue of a *perfect concept flip* within the guardedness framework. In LEACE (Theorem 3.3), concept erasure is achieved by enforcing $\text{Cov}(f(X), Z) = 0$, ensuring that the transformed representation carries no linear signal about the concept. In contrast, the constraint above enforces a complete inversion of this signal: samples that previously correlated positively with the concept now correlate equally strongly with its complement, and vice versa. This corresponds to a symmetric, dataset-wide concept swap and mirrors the role of reflection-based switching in vanilla steering.

Analogous to Theorem 3.3, we now characterize the optimal affine transformation that satisfies the constraint in Eq. 9 while minimally disturbing the original representation:

**Theorem 4.2** (LEACE-Switch, Optimal Concept Switching). *Let $X, Z, \Sigma_{XX}, \Sigma_{XZ}$ be defined as in Theorem 3.3. Assume $\text{Im}(\Sigma_{XZ}) \subseteq \text{Im}(\Sigma_{XX})$. Then the optimization problem*

$$\min_{\substack{A \in \mathbb{R}^{d \times d}, \\ b \in \mathbb{R}^d}} \text{E}\left[\|AX + b - X\|_2^2\right] \qquad (11)$$

$$\text{s.t. } \text{Cov}(AX + b, Z) = -\text{Cov}(X, Z) \qquad (12)$$

*has the following solution (almost surely):*

$$\widehat{A} = I - 2W^+(W\Sigma_{XZ})(W\Sigma_{XZ})^+ W, \qquad (13)$$

$$\widehat{b} = \mathbb{E}[X] - \widehat{A}\mathbb{E}[X], \qquad (14)$$

The proof is found in the Appendix D.2. Also note, since the transform is affine, it can be incorporated into the weight matrix of the model, thus achieving zero inference overhead. Similar to corollary 4.1, we now show that concept switching as defined in Eq. 4 is a special case of Theorem 4.2.

**Corollary 4.3.** *Let $X$ be a standardized random vector in $\mathbb{R}^d$, i.e. it has zero mean $\mathbb{E}[x] = 0$ and unit covariance matrix $\Sigma_{XX} = I$. Let $C \in \{0,1\}$ be a concept indicator variable. Let $s$ be defined as in Eq. 1. Let $f_{switch}$ be defined as in Eq. 4. Then $f_{switch}$ as a function of $h$ minimizes*

$$\min_{f \in \text{Aff}(\mathbb{R}^d \mapsto \mathbb{R}^d)} \text{E}[\|f(X) - X\|^2] \qquad (15)$$

$$\text{s.t. } \text{Cov}(f(X), C) = -\text{Cov}(X, C) \qquad (16)$$

The proof is found in the Appendix Sec. D.3. This theorem shows that steering in switching mode can be seen as LEACE-Switch under the assumptions that the whitening matrix is the identity and the mean of all vectors is zero.

**Scope of LEACE-Switch.** While the constraint in Eq. 9 precisely captures the algebraic notion of inverting a linear concept signal, it does so under a specific set of assumptions that define the regime in which LEACE-Switch is theoretically well-posed.

First, Eq. 9 assumes that the binary concept variable $Z$ partitions the dataset, i.e., $P(Z = 1) + P(Z = 0) = 1$. Under this condition, the linear dependence between $X$ and $Z$ is defined globally over the data distribution and admits a dataset-wide inversion. When instead considering two concepts $c_1$ and $c_2$ with corresponding indicators $Z_1$ and $Z_2$ that do not jointly span the dataset, this assumption is violated. In such cases, the notion of a dataset-wide bidirectional concept flip is no longer well-posed, and Theorem 4.2 does not apply. Second, even when $Z$ does partition the dataset, the constraint in Eq. 9 enforces a *complete* inversion of the linear dependence on the concept label. As a result, LEACE-Switch implements a symmetric transformation: representations associated with $c_1$ are mapped to

those of $c_2$, and representations associated with $c_2$ are simultaneously mapped to those of $c_1$. While this behavior is desirable in settings that explicitly require bidirectional swapping, it may be overly restrictive in applications where only a one-directional transformation is intended.

Together, these considerations delineate the scope of LEACE-Switch as an optimal solution for symmetric concept inversion under dataset partitioning. In the next section, we relax these assumptions and introduce a more general formulation for affine concept manipulation that enables directed, minimal-disturbance concept switching.

### 4.3. Optimal Affine Concept Manipulation

We now move beyond bidirectional concept switching and formulate a more general framework for affine concept manipulation. Our goal is to enable *directed* concept transformations that do not require dataset-wide label inversion or the assumption that the involved concepts jointly span the entire distribution.

Let $Z_1$ and $Z_2$ represent indicators of two groups of concepts $C_s = \{c_1^{(s)}, \ldots, c_l^{(s)}\}$ and $C_t = \{c_1^{(t)}, \ldots, c_l^{(t)}\}$. Let $Z = (Z_1, Z_2)$, where $Z_1, Z_2 \in \{0,1\}^l$. Our goal is to have $\mathrm{Cov}(f(X), Z_1) = \mathrm{Cov}(X, Z_2)$, meaning for each $i$, the concept $c_i^{(s)}$ maps to to $c_i^{(t)}$. Now we formulate the following theorem:

**Theorem 4.4** (MidSteer, affine optimal concept manipulation). *Let $X, Z$ be defined as in Theorem 4.2 and assume $k = 2l$ for $l > 0$. Let $Z = (Z_1, Z_2)$, where $Z_1, Z_2 \in \{0,1\}^l$. Let $\Sigma_{XZ_i} = \mathrm{Cov}(X, Z_i), i \in \{1,2\}$ be the cross-covariance matrices between $X$ and $Z_i$. Assume $\mathrm{Im}(\Sigma_{XZ_i}) \subseteq \mathrm{Im}(\Sigma_{XX})$ for $i \in \{1,2\}$. Assume $\Sigma_{XZ_1}$ has full column rank: $\mathrm{rk}\left(\Sigma_{XZ_1}\right) = l$. Let $W = (\Sigma_{XX}^{1/2})^+$ be the whitening transformation. Let $\Sigma_{WX,Z_i} = W\Sigma_{XZ_i}$.*

*Then we have the following optimization problem:*

$$\min_{\substack{A \in \mathbb{R}^{d \times d} \\ b \in \mathbb{R}^d}} \mathrm{E}[\|AX + b - X\|_2^2] \quad (17)$$

$$s.t. \quad \mathrm{Cov}(AX + b, Z_1) = \mathrm{Cov}(X, Z_2) \quad (18)$$

*which has the following solution (almost surely):*

$$\widehat{A} = I + W^+(\Sigma_{WX,Z_2} - \Sigma_{WX,Z_1})\Sigma_{WX,Z_1}^+ W \quad (19)$$

$$\widehat{b} = \mathbb{E}[X] - \widehat{A}\mathbb{E}[X] \quad (20)$$

The proof can be found in Sec. D.4. Unlike LEACE, we now have two covariance matrices for each group of concepts, and do not require the labels to be flipped, but translated from one group to another.

**On the full-rank assumption** in Thm. 4.4. Rank deficiency indicates redundancy among the source concept directions.

In that case, one can restrict to an independent subset or work in the effective subspace via a pseudoinverse. In our experiments $l = 1$, so the condition reduces to $\Sigma_{XZ_1} \neq 0$, i.e., the source concept must have a nontrivial linear footprint at the chosen layer. If not, the affine update is degenerate and MidSteer is not meaningfully applicable there.

**Connection between MidSteer and LEACE.** Note that if $Z_2$ is a constant indicator (e.g., representing no item in the dataset), then $\mathrm{Cov}(X, Z_2) = 0$, and the MidSteer formula turns into concept erasure (LEACE). Thus we refer to MidSteer as *affine optimal concept manipulation*, as it generalizes several concept manipulation tasks (erasure, switch) under one framework.

### 4.4. Steering strength

Let us now introduce the steering strength $\beta$ for erasure and switching. Note that in vanilla steering (i.e. steering defined by Eq. 3 and Eq. 4), we can unify Eq. 3 and Eq. 4:

$$f(h, s) = h - \beta \cdot \langle h, s \rangle s \quad (21)$$

For LEACE, $\widehat{b}$ is unaffected, and Eq. 6 and Eq. 13 become:

$$\widehat{A} = I - \beta \cdot W^+(W\Sigma_{XZ})(W\Sigma_{XZ})^+ W \quad (22)$$

Now, $\beta = 1$ represents concept erasure, $0 < \beta < 1$ represents lesser degree of erasure. $\beta = 2$ represents concept switching, and $\beta > 2$ refers to switching where the switched concept is more expressed than the base concept.

For MidSteer, we update Eq. 19 as:

$$\widehat{A} = I + \beta \cdot W^+(\Sigma_{WX,Z_2} - \Sigma_{WX,Z_1})\Sigma_{WX,Z_1}^+ W \quad (23)$$

$\beta = 1$ represents normal steering mode, $\beta < 1$ represent steering where $Z_2$ has less expression compared to what $Z_1$ had in original representation. Vice versa, $\beta > 1$ represent more expression for $Z_2$ compared to $Z_1$.

## 5. Experiments

### 5.1. Experimental setup

We conduct experiments comparing MidSteer to vanilla steering and LEACE-Switch on the task of concept switching. We focus on concept switching in the main experiments, as concept erasure is recovered as a special case when the target concept is constant. However, we provide comparison of all the approaches on concept erasure in the Appendix Sec. L. We consider two modalities: (1) text, for which we use Large Language Models (LLMs); (2) images, for which we use Diffusion Models. As for concepts, we consider both *concrete* nominal concepts, such as object categories, and *abstract safety-related* concepts, such as toxicity, helpfulness, violence, and peace. We show that MidSteer improves

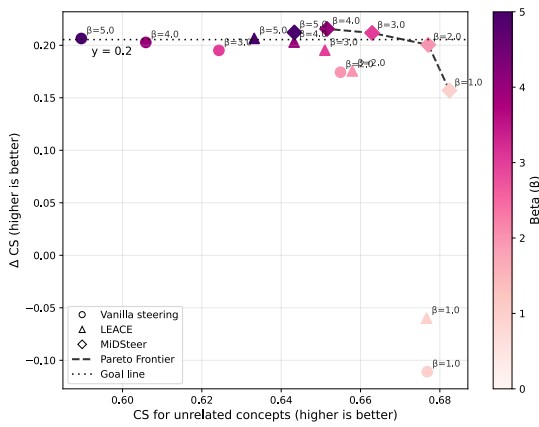

*(a)* $\Delta CS$ vs CS of unrelated concepts
for the Llama-2-7b model.

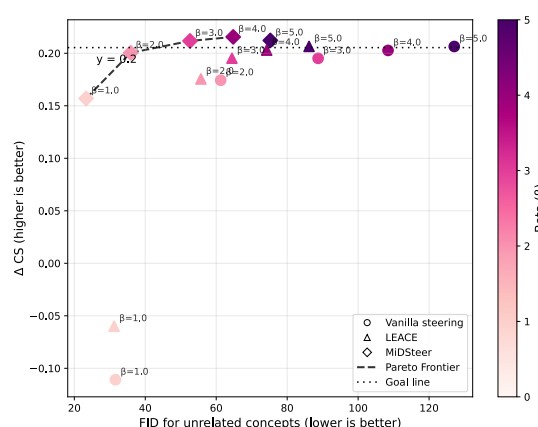

*(b)* $\Delta CS$ vs 1 - BERT Precision on MMLU
for the Llama-2-7b model.

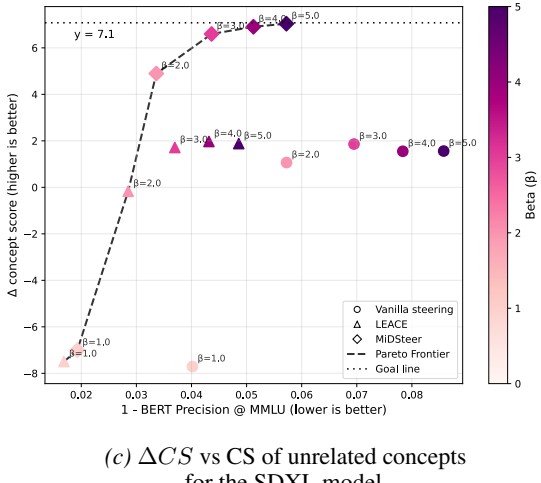

*(c)* $\Delta CS$ vs CS of unrelated concepts
for the SDXL model.

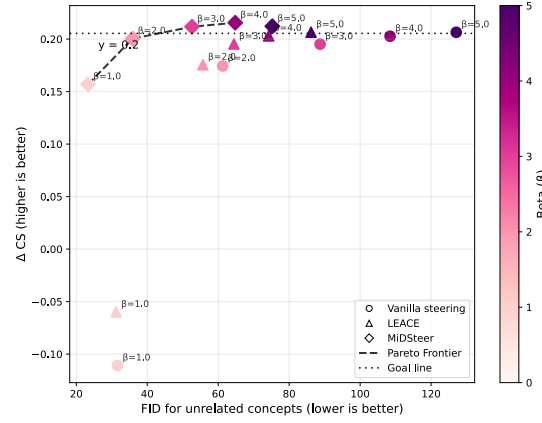

*(d)* $\Delta CS$ vs FID of unrelated concepts
for the SDXL model.

*Figure 2.* Pareto efficiency frontiers for concept *switching* experiments with steering, LEACE, and MidSteer highlighting different $\beta$s.

directed concept switching for both concrete categories and safety-relevant behavioral and visual concepts.

**General covariance estimation.** In each case, given a pair of concepts ($c_s$, $c_t$) to switch, we estimate class-conditional covariances $\Sigma_{XZ_1}, \Sigma_{XZ_2}$ based on a sample of size $N = 1000$, and self-covariances $\Sigma_{XX}$ on a sample of size $M = 50000$. We provide details about datasets used for $\Sigma_{XZ_1}, \Sigma_{XZ_2}$ and $\Sigma_{XX}$ estimation for each experiment described below in the Appendix Sec. E. We also ablate the number of prompts needed for estimating $\Sigma_{XX}$ in the Appendix Sec. H, and provide pseudocode for the algorithm on covariance estimation in the Sec. A.

**Concrete concept switching setup.** As no standard benchmarks currently exist for concept switching, we introduce a dedicated evaluation setup. Our procedure is largely based on established concept erasure benchmarks used for vision generative models (Lyu et al., 2024; Wu et al., 2025). To

test switching from the source concept $c_s$ to the target concept $c_t$, we use 80 template prompts prompting the model to generate output related to $c_s$ or $c_t$. For each prompt we run 10 such generations varying the random seed. We run the generation on these prompts with and without steering. Templates for LLMs and diffusion models can be found in Sec. G. We use *Concept Score (CS)* to estimate the amount of concepts $c_s$ and $c_t$ present in the model's output. In the case of ideal switching, CS for concept $c_t$ should be high, and CS for concept $c_s$ should be low for prompts related to both $c_s$ or $c_t$.

To evaluate content preservation beyond the switched concepts, we additionally generate outputs for four unrelated concepts $\{c_i\}_{i=1}^{4}$. These concepts are chosen to span varying levels of semantic proximity to $c_s$ and $c_t$. We compute CS over $\{c_i\}_{i=1}^{4}$ to quantify the extent to which these concepts are retained in the generated outputs, which ideally should not decrease under steering. This design enables a

more fine-grained comparison of concept switching methods with respect to semantic interference.

We use the following pairs $p_i = (c_s \rightarrow c_t)$ of concrete concepts: $p_1$ = ("Horse" → "Motorcycle"), $p_2$ ="Dog" → "Cat", $p_3$ ="Chihuahua" → "Muffin"). Note that none of these pairs span the whole dataset. Corresponding unrelated concepts $t_i = \{c_i\}_{i=1}^4$ are : $t_1$ = ("Cow", "Dog", "Pig", "Legislator"), $t_2$ = ("Cow", "Wolf", "Pig", "Legislator"), $t_3$ = ("Cat", "Dog", "Wolf", "Legislator").

**Safety-related concept switching.** We additionally evaluate abstract safety-related switching. In LLMs, we switch *toxicity* to *helpfulness*. The corresponding unrelated concepts are ("sarcasm", "creativity", "politeness", "mathematics"). In diffusion models, we evaluate the analogous visual safety setting of switching *violence* to *peace*, and the corresponding unrelated concepts are ("sadness", "calm", "anger", "nature"). This directly tests the motivating use case of suppressing harmful behavior while preserving or increasing useful behavior.

As in the concrete-concept setup, we use 80 template prompts with 10 random seeds per concept. The only exception is the toxicity source concept. For toxicity, we use RealToxicityPrompts with prompt toxicity $\geq 0.5$ instead of template prompts, since we found that instruction-tuned LLMs often refuse to produce toxic continuations from generic toxicity templates. RealToxicityPrompts therefore provides a more reliable evaluation set for measuring whether steering reduces toxic behavior in settings where the base model can still exhibit toxicity.

**Details on LLM experiments.** We test on instruction-tuned Llama 2 (Touvron et al., 2023) and Qwen 2.5 (Yang et al., 2024a) models. We apply steering at every self-attention (SA) layer ans use SA activations corresponding to the last token in prompt.

We calculate the *Concept Score (CS)* for *concrete* concepts using LLM as a judge. More specifically, we use the Llama-3.1-8B-Instruct (Dubey et al., 2024) model, and prompt it to estimate the amount of concept $c$ present in the generated output on a scale from 0 to 10. Prompts used are outlined in the Appendix Sec. F. We additionally report results using GPT-4o-mini as LLM judge in the Appendix Sec. J to support reliability of using LLM as a judge for scoring. For safety-related LLM switching, we measure toxicity using the Detoxify BERT classifier (Hanu & Unitary team, 2020), and helpfulness using ArmoRM reward scores (Wang et al.), while keeping LLM as a judge approach for assessing preservation of unrelated concepts.

We additionally test how much outputs for testing concepts differ with and without steering by calculating BERT scores (Zhang et al., 2020) on the MMLU (Hendrycks et al., 2021) dataset generations. Lower values of BERT Precision

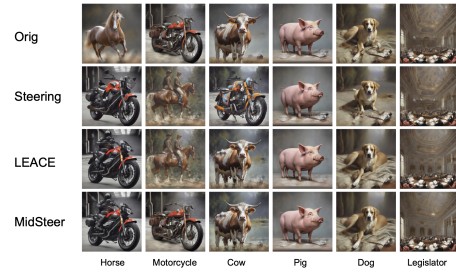

*Figure 3.* Qualitative results on switching to steer "horses" into "motorcycles". While all methods similarly successfully performed switching from "horse" to "motorcycle", vanilla steering (CASteer) and LEACE fail when presented with prompt for the target concept ("motorcycle"), unable to distinguish between forward and reverse steering. CASteer also additionally failed on the "cow" concept, and more significantly altered images of the concept "dog".

*Table 1.* Results on SDXL when flipping from "horse" to "motorcycle". Reported are CLIP-scores (cs) and FID for target and non-target concepts.

| method | strength | horse src-cs ↓ | horse tgt-cs ↑ | motorcycle src-cs ↓ | motorcycle tgt-cs ↑ | motorcycle fid ↓ | cow cs ↑ | cow fid ↓ | pig cs ↑ | pig fid ↓ | dog cs ↑ | dog fid ↓ | legislator cs ↑ | legislator fid ↓ |
|---|---|---|---|---|---|---|---|---|---|---|---|---|---|---|
| orig | - | 71.0 | 49.1 | 51.8 | 70.7 | - | 72.7 | - | 71.8 | - | 66.3 | - | 60.8 | - |
| CASteer | 2.0 | 52.1 | 69.5 | 68.3 | 52.9 | 212.4 | 70.9 | 42.7 | 71.9 | 18.9 | 66.1 | 28.6 | 60.9 | 24.6 |
| LEACE | 2.0 | 51.2 | 68.8 | 67.6 | 53.3 | 207.6 | 72.2 | 25.2 | 71.7 | 12.6 | 66.1 | 20.8 | 60.6 | 28.2 |
| MidSteer (ours) | 1.0 | 51.2 | 68.7 | 51.9 | 70.7 | 12.7 | 72.2 | 23.9 | 71.8 | 12.4 | 66.1 | 20.7 | 60.7 | 27.2 |

represent more change in the underlying output.

**Details on diffusion model experiments.** For visual diffusion models, we test on SDXL (Podell et al., 2024) and SANA (Xie et al., 2025) models. Following recent work (?), we apply steering on activations of every cross-attention (CA) layer. CA activations corresponding to all images patches are used.

We use CLIP score (Hessel et al., 2021) as *Concept Score (CS)* for both concrete and safety-related concepts. Additionally, we measure the change between generations based on the testing concepts when steering is applied, by calculating FID (Heusel et al., 2017) between images generated by steered model and vanilla model. Higher values of FID represent more change to the underlying image.

## 5.2. Experimental results

Across both concrete and safety-related settings, MidSteer provides the best trade-off between successful directed concept switching and preservation of non-target content. Vanilla steering can often induce the target concept, but it also causes stronger interference and frequently acts symmetrically, modifying prompts that already contain the target concept. LEACE-Switch improves preservation relative to vanilla steering, but inherits the limitation of bidirectional concept inversion. MidSteer avoids this failure mode by directly matching the source-concept cross-covariance to the target-concept cross-covariance.

**Concrete concept switching.** We compare results on con-

*Table 2.* Safety-related concept switching on Llama-2-7B-chat and SDXL. Bold indicates the best intervention result in each column.

| | *Toxicity → Helpfulness, Llama-2-7B-chat* | | | |
|---|---|---|---|---|
| Method | RTP ↓ | Help ↑ | Unrel. ↑ | MMLU (ERT-F1) ↑ |
| Base | .371 | **.117** | 8.46 | 1.00 |
| Vanilla-3 | .369 | .102 | 8.45 | .92 |
| Vanilla-5 | .380 | .029 | 3.11 | .90 |
| LEACE-S-3 | .344 | **.117** | 8.47 | .97 |
| LEACE-S-5 | .346 | .116 | 8.47 | .95 |
| MidSteer-3 | .309 | **.117** | 8.50 | .95 |
| MidSteer-5 | **.281** | **.117** | 8.48 | .94 |

| | *Violence → Peace, SDXL* | | | |
|---|---|---|---|---|
| Method | Viol. ↓ | Peace ↑ | Unrel. ↑ | FID ↓ |
| Base | 99.6 | 71.8 | 51.5 | 0.0 |
| Vanilla-3 | 42.2 | 74.5 | 50.6 | 93.6 |
| Vanilla-5 | 20.8 | 96.5 | 50.4 | 122.2 |
| LEACE-S-3 | 76.0 | 33.2 | 51.4 | 88.9 |
| LEACE-S-5 | 28.2 | 54.5 | 51.7 | 121.6 |
| MidSteer-3 | 12.5 | 90.5 | 50.8 | 101.5 |
| MidSteer-5 | **6.0** | **94.0** | 51.0 | 134.6 |

*Notes.* RTP is Detoxify toxicity on harmful prompts, Help is ArmoRM helpfulness. Unrel. is preservation on four unrelated concepts.

cept switching by applying vanilla steering, LEACE-Switch and MidSteer with different values of $\beta$. We present results on LLama-2-7b and SDXL models aggregated for all three concept pairs $p_1, p_2, p_3$ in Fig. 2. We see that in each case MidSteer achieves much better balance between level of concept switch between $c_1$ and $c_2$ and preservation of other concepts across different values of $\beta$. Pareto plots for other models, as well as Pareto plots for individual concept and metric pairs can be found in the Appendix Sec. K.0.1, K.1.

To better illustrate differences between vanilla steering, LEACE and MidSteer for concept flipping, in Tab. 1 we present results on switching a concept of $c_1 =$ "horse" to $c_2 =$ "motorcycle" on the SDXL model. We compare Vanilla switching and LEACE with $\beta = 2$ and MidSteer with $\beta = 1$, as these are default parameters for these methods as suggested by Eqs. 25, 13, and 19.

First note, that all the methods successfully flip "horse" to "motorcycle", having similar CS scores on source ("horse") and target ("motorcycle") concepts. Second, it can be seen that as suggested by definitions Eqs. 25, 13, vanilla steering and LEACE fail to keep the "motorcycle" concept intact when flipping "horse" to "motorcycle", as target CS score goes down. In contrast, MidSteer keeps "motorcycle" intact. This is also illustrated in Fig. 3. Next, CS score of "cow" and FID scores of "cow", "pig" and "dog" are worse for vanilla than for other methods, showing superiority of LEACE and MidSteer over vanilla steering in ability to keep unrelated concepts intact. Results on other concepts flipping on both LLMs and diffusion models show similar patterns. We observe the same trend over all the concept pairs and models. We give more details in Appendix Sec. K.2, K.3.

**Safety-related concept switching.** In contrast to the concrete object-switching setup, these experiments involve more abstract concepts, where the source concept corresponds to an undesirable or harmful behavior and the target concept corresponds to a desirable alternative. We report our main results on toxicity → helpfulness switching on Llama-2-7B-chat, and for diffusion models, we consider switching violence → peace on SDXL. We report full results on all $\beta \in \{1, 2, 3, 4, 5\}$, as well as results on Qwen 2.5 and SANA in the Appendix Sec. I.

The main results are summarized in Tab. 2. For Llama-2-7B-chat, MidSteer achieves the strongest reduction in toxicity, decreasing the RTP toxicity score from .371 to .281 at $\beta = 5$, while preserving helpfulness at the baseline level and maintaining high scores on unrelated concepts. Vanilla steering does not reliably improve toxicity and substantially degrades unrelated-concept preservation at higher strength, with the unrelated concept score dropping from 8.46 to 3.11. LEACE-Switch preserves general capabilities slightly better, as reflected by the highest MMLU BERT-F1 among interventions, but achieves much weaker toxicity reduction than MidSteer. These results suggest that MidSteer provides a better trade-off between safety improvement and preservation of unrelated behavior.

A similar trend appears for SDXL in the violence → peace setting. MidSteer most effectively suppresses the source concept, reducing the violence score from 99.6 to 6.0, while also achieving the strongest peace score among intervention methods. Vanilla steering also reduces violence, but either underperforms MidSteer on source suppression or causes larger distortions, as reflected by higher FID. LEACE-Switch yields the lowest FID among interventions but is substantially less effective at inducing the target concept. Overall, these results show that MidSteer extends beyond concrete object substitutions and remains effective for abstract, safety-related concept switching across both language and vision modalities.

## 6. Conclusion

In this work, we bridge the gap between previous empirical research in steering generative models and the theory of affine concept steering. We extend this theoretical framework to concept switching. We define the corresponding optimisation problem and solve it in closed form. We then present MidSteer, a general steering method, that is theoretically optimal under certain conditions. It outperforms other methods on concept switching for both LLMs and image diffusion models, while having the advantage of clear matrix form representation. To our knowledge, this is the first theoretical treatment of steering beyond erasure, connecting empirical heuristics and principled affine methods, while reaching state-of-the-art results.

## Impact Statement

This paper advances the theoretical understanding of steering and representation manipulation in generative models. Because such methods can be used to suppress harmful concepts and promote desirable behaviors, they are directly relevant to post-deployment alignment, safety, and controllable generation. In particular, more principled steering methods may help reduce undesirable outputs such as toxic text or violent imagery while better preserving unrelated capabilities and content, which is an important consideration for practical deployment.

At the same time, representation-level control is a dual-use capability. The same techniques that enable safer or more useful behavior could also be used to manipulate model outputs in undesirable ways, such as amplifying harmful, biased, deceptive, or otherwise inappropriate concepts. Moreover, steering interventions may have unintended side effects if applied with poorly estimated concept directions, mismatched data distributions, or insufficient evaluation of unrelated behaviors. For this reason, we view careful evaluation of preservation, robustness, and failure modes as an essential part of deploying such methods responsibly.

Overall, we believe the primary societal relevance of this work lies in providing a more principled foundation for controllable and safety-oriented model interventions, while also highlighting the need for safeguards and evaluation protocols when applying representation manipulation in real systems.

## Acknowledgments

This work was supported by a Google DeepMind PhD Studentship, and the work utilized Queen Mary's Andrena HPC facility, supported by QMUL Research-IT. This work was also supported by the Engineering and Physical Sciences Research Council [grant number EP/Y009800/1], through funding from Responsible Ai UK (KP0016).

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

# A. Algorithm for computing covariances

To estimate the covariances we use the algorithm by (Welford, 1962) on a sample of broad prompts (unrelated to the steering concepts). Given $X$ with the dimension of $batch\_size \times num\_heads \times seq\_len \times hidden\_dim$, the algorithm estimates the covariance matrix $\Sigma_X X$ of size $hidden\_dim \times hidden\_dim$. It does this by maintaining sample-level statistics of size $O(hidden\_dim^2)$ in memory and takes $O(batch\_size * seq\_len * hidden\_dim^2)$ time to update them for the output of a particular layer on a particular batch. In practice, estimating the covariances for 50,000 samples for SANA 1.6 finished in under 15 minutes, and for Qwen2.5 14B in under 30 minutes on our hardware setup.

In Algorithm 1 we provide pseudocode of the algorithm.

---

**Algorithm 1** Welford's Algorithm for Online Mean and Covariance Estimation

---

1: **Input:** Stream of data batches $\{\mathbf{X}_1, \mathbf{X}_2, \ldots, \mathbf{X}_K\}$ where $\mathbf{X}_k \in \mathbb{R}^{h \times m_k \times d}$
2: **Output:** Mean $\boldsymbol{\mu}$ and covariance $\boldsymbol{\Sigma}$ estimates
3: **Initialization:**
4: $\quad n \leftarrow 0$                      $\triangleright$ Total count of samples seen
5: $\quad \mathbf{M} \leftarrow \mathbf{0} \in \mathbb{R}^{h \times d}$                      $\triangleright$ Running sum
6: $\quad \mathbf{S} \leftarrow \mathbf{0} \in \mathbb{R}^{h \times d \times d}$               $\triangleright$ Running sum of squared differences
7:
8: **for** $k = 1$ to $K$ **do**                    $\triangleright$ Process each batch
9: $\quad m_k \leftarrow$ number of samples in batch $\mathbf{X}_k$
10: $\quad$ **if** $n = 0$ **then**                  $\triangleright$ First batch - initialize
11: $\quad\quad n \leftarrow m_k$
12: $\quad\quad \mathbf{M} \leftarrow \sum_{j=1}^{m_k} \mathbf{x}_{k,j}$                $\triangleright$ Sum over samples
13: $\quad\quad \boldsymbol{\mu} \leftarrow \mathbf{M}/n$
14: $\quad\quad \boldsymbol{\Delta} \leftarrow \mathbf{X}_k - \boldsymbol{\mu} \otimes \mathbf{1}_{m_k}^{\top}$              $\triangleright$ Center vectors
15: $\quad\quad \mathbf{S} \leftarrow \boldsymbol{\Delta}^{\mathsf{H}} \boldsymbol{\Delta}$
16: $\quad$ **else**                   $\triangleright$ Subsequent batches - update
17: $\quad\quad \boldsymbol{\mu}_{\text{old}} \leftarrow \mathbf{M}/n$                $\triangleright$ Mean before update
18: $\quad\quad n \leftarrow n + m_k$                 $\triangleright$ Update count
19: $\quad\quad \mathbf{M} \leftarrow \mathbf{M} + \sum_{j=1}^{m_k} \mathbf{x}_{k,j}$              $\triangleright$ Update sum
20: $\quad\quad \boldsymbol{\mu}_{\text{new}} \leftarrow \mathbf{M}/n$                $\triangleright$ Mean after update
21: $\quad\quad \boldsymbol{\Delta}_{\text{old}} \leftarrow \mathbf{X}_k - \boldsymbol{\mu}_{\text{old}} \otimes \mathbf{1}_{m_k}^{\top}$
22: $\quad\quad \boldsymbol{\Delta}_{\text{new}} \leftarrow \mathbf{X}_k - \boldsymbol{\mu}_{\text{new}} \otimes \mathbf{1}_{m_k}^{\top}$
23: $\quad\quad \mathbf{S} \leftarrow \mathbf{S} + \boldsymbol{\Delta}_{\text{old}}^{\mathsf{H}} \boldsymbol{\Delta}_{\text{new}}$             $\triangleright$ Welford update
24: $\quad$ **end if**
25: **end for**
26:
27: **Finalization:**
28: $\boldsymbol{\mu} \leftarrow \mathbf{M}/n$                    $\triangleright$ Final mean estimate
29: $\boldsymbol{\Sigma} \leftarrow \frac{1}{n-1} \cdot \frac{\mathbf{S}+\mathbf{S}^{\mathsf{H}}}{2}$               $\triangleright$ Final covariance estimate
30: **return** $\boldsymbol{\mu}, \boldsymbol{\Sigma}$

---

## B. Incorporating steering into model weights

Recall that the last layer of self-attention block in LLMs or cross-attention block in SDXL/SANA is a Linear layer with no bias and no activation function, i.e., essentially is a matrix multiplication with bias correction term: $h_{out} = W_{proj\_out}h_{in} + b$. Here $W_{proj\_out}$ is a weight matrix of the last $proj\_out$ layer of SA/CA block of LLM/SDXL/SANA, $h_{in}$ and $h_{out}$ are input and output to that layer, $h_{out}$ being the final output of SA/CA layer.

Assuming $s$ is normalized ($\|s\| = 1$), vanilla concept erasure represents orthogonal projection onto the subspace orthogonal to $s$ and can be written in a matrix form:

$$f_{\text{delete}}(h, s) = (I - ss^T)h \tag{24}$$

Vanilla concept switching is a Householder reflection of the vector $h$ across the hyperplane orthogonal to $s$:

$$f_{\text{switch}}(h, s) = (I - 2ss^T)h \tag{25}$$

LEACE / MidSteer are already presented in this paper in matrix form.

Thus, by combining last layer of SA/CA block with matrix formulation of steering/LEACE/MidSteer, we can incorporate the transformation directly into weights of the model, by multiplying weight matrix of the last layer of SA/CA block with $A^*$ matrix from of steering/LEACE/MidSteer:

$$h_{out} = A^*(W_{proj\_out}h_{in} + b) + b^* = W^s_{proj\_out}h_{in} + b^s \tag{26}$$

$W^s_{proj\_out}$ is a matrix of the same size as $W_{proj\_out}$. This results in having zero inference overhead compares to original LLM/SDXL/SANA models.

Therefore, once the affine transform is estimated, it can be folded into the projection weights, so generation runs at the same inference-time cost as the original model. Self-covariance matrix $\Sigma_{XX}$ is computed once per model, and only the cross-covariances are concept-specific. Cross-covariance estimation takes about one minute for 1000 prompts on a single H100 GPU. Thus, the trade-off is one-time offline estimation per concept pair versus zero runtime overhead after. This is particularly attractive in repeated-use settings, such as model-level safety policies.

| | Horse | Motorcycle | Cow | Dog |
|---|---|---|---|---|
| **Vanilla model** | In the heart of a vast, rolling green valley lay a secluded meadow embraced by ancient oaks and wildflowers in bloom. Here, the **horses** lived a simple, joyous life... | In the heart of the bustling city of San Francisco, there lived a **motorcycle** named Thunder. Unlike the other shiny, new **bikes** lining the streets, Thunder was an old Kawasaki with rusted edges... | In the lush, rolling hills of Green Meadows Farm, there lived a herd of contented **cows** that grazed under the warm sun and shared the soft, dewy grass. Among them was Daisy, a golden **cow** with spots as vivid as a child's imagination... | In the quiet town of Willowbrook, nestled between rolling hills and whispering woods, lived a small **terrier** named Bella. Bella was not just a **dog**; she was the heart of the household... |
| **Steering** | In the heart of a bustling city, there was once a forgotten corner, nestled between towering skyscrapers and hidden behind a rusty iron gate, where a vintage **bike** shop and a small garage sat side by side. Among the neglected treasures within the **garage** were forgotten car parts... | In the heart of a bustling city, amidst towering skyscrapers and winding streets, lived an old **stallion** named Max. Unusual for the concrete jungle, Max was never your typical **beast**... | In the quiet countryside of Green Valley, there lived a community of **cows**, but not just any **cows**. This was a town of the most unique and quirky cows you could imagine. Each cow had its own style of living, its way of seeing the world, and even its own personality, much like humans... | In the heart of a bustling city where skyscrapers kissed the sky and the streets were a maze of hurried footsteps and whispering winds, lived a small, golden **retriever** named Buddy. Buddy wasn't your ordinary family pet; his world was the park near the city's old library... |
| **LEACE-Switch** | In the heart of a quaint countryside stood a grand old barn that was more than just a shelter; it was the soul of a **motorcycle** club known as The Thunder, but hidden deep within its walls was a story of grace, freedom, and unexpected friendship... | In the heart of a sprawling, bustling city, there lived an old man named Jack who harbored a long-lost passion for **horses** until the day he met his first **motorcycle**. It was a sleek, black **beast** he spotted outside a garage one rainy evening... | In the peaceful, sun-drenched fields of Green Meadows Farm, there lived a herd of contented **cows**, each with its own personality and quirks. At the heart of this herd was a gentle giant named Bessie, known for her creamy coat and kind eyes... | In the quiet town of Millbrook, nestled among rolling hills and vibrant green landscapes, lived a **dog** named Buddy. Buddy wasn't just any **dog**; he was a loyal companion to the Smith family, known for his boundless energy and affectionate nature... |
| **MidSteer** | In the heart of the sprawling Wilden Plains, there thrived a community of **motorcycles** and cars, but it was the Rusty Rims Speedway that drew the greatest attention. Yet, hidden within the shadow of this bustling speedway, lived a group of **motorcycles**... | In the heart of a bustling city, where the sound of the urban jungle was constant, there lived a **motorcycle** named Racer. More than just a machine, Racer was an embodiment of freedom and adventure, cherished by his owner, Jake... | In the verdant fields of Sunny Meadows, where the sun bathed golden light onto rolling hills blanketed in lush grass, there lived a community of cheerful **cows**. Among them was Daisy, known not only for her creamy coat and gentle moo but for her adventurous spirit... | Once upon a time, in a small town nestled between rolling hills and a whispering forest, there lived a **dog** named Rusty. Rusty wasn't like the other **dogs** in town. While they were mostly interested in chasing balls and squirrels, Rusty loved nothing more than exploring the woods that bordered the town... |

*Figure 4.* Qualitative text steering results for four content categories (*horse*, *motorcycle*, *cow*, *dog*). Results are reported using vanilla Qwen2.5-14B-instruct model, and three steering methods: *Vanilla Steering*, *LEACE-Switch*, *MidSteer*). Each cell shows the generated text for the prompt *"Write a short story about a X"*, where X is a corresponding category.

# C. LLM qualitative results

In this section in fig. 4 we present qualitative results for different steering methods on LLM (Qwen2.5-14B-instruct). Results show similar pattern to that of images (see fig. 3). While all methods similarly successfully performed switching from "horse" to "motorcycle", vanilla steering and LEACE failed when presented with prompt for the target concept ("motorcycle"), since they do not distinguish between forward and reverse steering. Vanilla steering also more significantly altered texts of concepts "cow" and "dog".

# D. Theorem proofs

### D.1. Proof of Thm. 4.1 (vanilla erasure is a special case of LEACE)

*Proof.* We have $k = 1, Z = C$. According to **Theorem 3.3**, $f(X) = A^*X + b^*$, where $A^*$ is defined as in (6) and $b^*$ is defined as in (7), minimizes (5).

We conclude $b^* = 0$ since $\mathrm{E}[X] = 0$. Further, it can be shown that $W = \Sigma_{XX}^{-1/2} = I^{-1/2} = I$; hence, the transform $f$ (that is optimal according to **Theorem 3.3**) simplifies to

$$f(X) = \left( I - \Sigma_{XZ} \Sigma_{XZ}^+ \right) X . \tag{27}$$

Recall that we are working in $k = 1$, so $\Sigma_{XZ} \in \mathbb{R}^{d \times 1}$ is a column-vector. By definition of the Moore-Penrose inverse for column-vectors,

$$\Sigma_{XZ}^+ = \frac{\Sigma_{XZ}^T}{\|\Sigma_{XZ}\|^2} ,$$

hence

$$f(X) = X - s' s'^T X = X - \langle X, s' \rangle s',$$

for $s' = \Sigma_{XZ}/\|\Sigma_{XZ}\|$,

Now,

$$\Sigma_{XZ} = \mathrm{Cov}(X, Z) = \mathrm{E}[XZ] - \mathrm{E}[X] \cdot \mathrm{E}[Z] = \mathrm{E}[X \cdot 1 | Z = 1] \cdot P(Z = 1) +$$
$$\mathrm{E}[X \cdot 0 | Z = 0] \cdot P(Z = 0) - \mathrm{E}[X] \cdot P(Z = 1) =$$
$$P(Z = 1) \cdot \left( \mathrm{E}[X | Z = 1] - \mathrm{E}[X] \right)$$

Now recall that $P(Z = 1) + P(Z = 0) = 1$, so

$$\Sigma_{XZ} = P(Z = 1) \cdot \left( \mathrm{E}[X | Z = 1] - \mathrm{E}[X] \right) =$$
$$P(Z = 1) \cdot \left( \mathrm{E}[X | Z = 1] - \mathrm{E}[X | Z = 1] P(Z = 1) - \mathrm{E}[X | Z = 0] P(Z = 0) \right) =$$
$$P(Z = 1) \cdot P(Z = 0) \cdot \left( \mathrm{E}[X | Z = 1] - \mathrm{E}[X | Z = 0] \right)$$

, so $s' = s$ and $f$ is equivalent to $f_{delete}$. Hence, $f_{delete}$ is the transformation that minimises (8).

$\square$

### D.2. Proof of Thm. 4.2 (LEACE-Switch)

*Proof.* The sketch for the rest of the proof will look like this:

1. Find necessary conditions for optimality using Lagrange multipliers method

2. Show that $A^*, b^*$ satisfy the necessary conditions

3. Show that optimisation problem is convex over linear constraints, and such, if a local solution exists, it is globally optimal and unique.

Let us formulate the Lagrangian. Here $\Lambda \in \mathbb{R}^{d \times k}$, because we have $d \cdot k$ constraints on covariance matrix.

$$\mathcal{L}(A, b, \Lambda) = \frac{1}{2}\mathrm{E}\Big[\|AX + b - X\|_2^2\Big] + \langle \Lambda, \mathrm{Cov}(AX + b, Z) + \mathrm{Cov}(X, Z)\rangle_F =$$
$$\frac{1}{2}\mathrm{E}\Big[(AX + b - X)^T(AX + b - X)\Big] + \mathrm{Tr}\Big(\Lambda^T(A + I)\Sigma_{XZ}\Big) =$$
$$\mathrm{E}\Big[\frac{1}{2}X^T A^T AX + b^T AX - X^T AX - X^T b + \frac{1}{2}b^T b + \frac{1}{2}X^T X\Big] + \mathrm{Tr}\Big(\Lambda^T(A + I)\Sigma_{XZ}\Big) \quad (28)$$

The partial derivatives of the Lagrangian with respect to $A, b, \Lambda$ are

$$\frac{\partial \mathcal{L}}{\partial A} = \mathrm{E}[AXX^T + bX^T - XX^T] + \Lambda \Sigma_{XZ}^T,$$
$$= A\mathrm{E}[XX^T] + b\mathrm{E}[X]^T - \mathrm{E}[XX^T] + \Lambda \Sigma_{XZ}^T,$$
$$\frac{\partial \mathcal{L}}{\partial b} = \mathrm{E}[AX + b - X],$$
$$= A\mathrm{E}[X] + b - \mathrm{E}[X],$$
$$\frac{\partial \mathcal{L}}{\partial \Lambda} = (A + I)\Sigma_{XZ}.$$

Next, we use $\mu = \mathrm{E}(X)$ and $\mathrm{E}[XX^T] = \Sigma_{XX} + \mu\mu^T$ to formulate the necessary conditions

$$0 = \frac{\partial \mathcal{L}}{\partial A} = (A - I)\Big(\Sigma_{XX} + \mu\mu^T\Big) + b\mu^T + \Lambda \Sigma_{XZ}^T, \quad (29)$$
$$0 = \frac{\partial \mathcal{L}}{\partial b} = A\mu + b - \mu, \quad (30)$$
$$0 = \frac{\partial \mathcal{L}}{\partial \Lambda} = (A + I)\Sigma_{XZ}. \quad (31)$$

We note that the optimal $b^*$ as defined in (14) satisfies 30. Plugging (30) in (29) leads to

$$(A - I)\Big(\Sigma_{XX} + \mu\mu^T\Big) + (\mu - A\mu)\mu^T + \Lambda \Sigma_{XZ}^T,$$
$$= A\Sigma_{XX} - \Sigma_{XX} + A\mu\mu^T - \mu\mu^T + \mu\mu^T - A\mu\mu^T + \Lambda \Sigma_{XZ}^T,$$
$$= (A - I)\Sigma_{XX} + \Lambda \Sigma_{XZ}^T = 0. \quad (32)$$

Now let us check that $A^*$ satisfies 31 and 32. By plugging $A^*$ into 31 we get

$$0 = (2I - 2W^+(W\Sigma_{XZ})(W\Sigma_{XZ})^+W)\Sigma_{XZ},$$
$$= 2\Sigma_{XZ} - 2W^+(W\Sigma_{XZ})(W\Sigma_{XZ})^+(W\Sigma_{XZ}),$$
$$= 2\Big(\Sigma_{XZ} - W^+W\Sigma_{XZ}\Big),$$
$$= 2\Big(\Sigma_{XZ} - \big(I - P_{\mathcal{N}(W)}\big)\Sigma_{XZ}\Big),$$
$$= 2P_{\mathcal{N}(W)}\Sigma_{XZ}, \quad (33)$$

because Moore-Penrose inverses $B^+$ of $B$ satisfy $BB^+B = B$ and $B^+B = I - P_{\mathcal{B}}$. Here $P_{\mathcal{B}}$ denotes the orthogonal projection onto the nullspace $\mathcal{N}(B)$ of $B$. Since the columns of $\Sigma_{XZ}$ always lie within the image of $\Sigma_{XX}$ (which is the orthogonal complement of the kernel of $\Sigma_{XX}$, which is also the kernel of $W$), we can conclude that (33) is always satisfied.

Plugging $A^*$ into 32 we observe

$$-2 \cdot (W^+(W\Sigma_{XZ})(W\Sigma_{XZ})^+ W)\Sigma_{XX} + \Lambda\Sigma_{XZ}^T =$$
$$-2 \cdot W^+(W\Sigma_{XZ})(W\Sigma_{XZ})^+ W^+ + \Lambda\Sigma_{XZ}^T = 0 \quad (34)$$

The identity $W\Sigma_{XX} = W^+$ holds because $\Sigma_{XX}$ is symmetric p.s.d., so $\Sigma_{XX} = UDU^T$ and $\Sigma_{XX}^{-1/2}\Sigma_{XX} = UD^{-1/2}U^T UDU^T = UD^{1/2}U^T = \Sigma_{XX}^{1/2}$ for some orthogonal $U$ and non-negative diagonal $D$, and because $D^{-1/2}$ ignores zero diagonal values.

Next, multiplying (34) by $W$ from both sides leads to

$$-2WW^+(W\Sigma_{XZ})(W\Sigma_{XZ})^+ W^+ W + W\Lambda\Sigma_{XZ}^T W =$$
$$-2(W\Sigma_{XZ})(W\Sigma_{XZ})^+ + W\Lambda(W\Sigma_{XZ})^T = -2\Sigma_{WX,Z}\Sigma_{WX,Z}^+ + \Lambda_W\Sigma_{WX,Z}^T = 0, \quad \text{(almost surely)}$$

where again $WW^+ = W^+ W = I$ on a subspace covered by $X$, and thus, almost surely.

We seek $\Lambda_W$ such that $-2\Sigma_{WX,Z}\Sigma_{WX,Z}^+ + \Lambda_W\Sigma_{WX,Z}^T = 0$. Let $\Lambda_W = 2(\Sigma_{WX,Z}^+)^T$. This choice satisfies the condition because $\Sigma_{WX,Z}\Sigma_{WX,Z}^+$ is an orthogonal projection matrix, and is thus symmetric, so

$$(\Sigma_{WX,Z}^+)^T\Sigma_{WX,Z}^T = (\Sigma_{WX,Z}\Sigma_{WX,Z}^+)^T = \Sigma_{WX,Z}\Sigma_{WX,Z}^+,$$

which again proves the Lagrange conditions for partial derivative w.r.t. A.

Thus we have shown that the said optimisation problem has a local solution. But because the constraint is linear in $A$, and it follows from the triangle inequality that $\|\cdot\|$ is convex, the local optimum is actually the global minimum.

$\square$

### D.3. Proof of Thm. 4.3 (vanilla concept switching is a special case of LEACE-Switch)

*Proof.* Let $k = 1, Z = C, M = I$. According to **Theorem 4.2**, $f(X) = A^*X + b^*$, where $A^*$ is defined in (13) and $b^*$ is defined in (14) minimizes (11).

$b = 0$ since $\mathrm{E}[X] = 0$. Also it can be shown $W = \Sigma_{XX}^{-1/2} = I^{-1/2} = I$, so the transform becomes:

$$f(X) = \left(I - 2 \cdot \Sigma_{XZ}\Sigma_{XZ}^+\right)X \quad (35)$$

Recall that we are working in $k = 1$, so $\Sigma_{XZ} \in \mathbb{R}^{d \times 1}$ is a column-vector. So

$$\Sigma_{XZ} = \mathrm{Cov}(X, Z) = \mathrm{E}[XZ] - \mathrm{E}[X] \cdot \mathrm{E}[Z] = \mathrm{E}[X \cdot 1|Z = 1] \cdot P(Z = 1) +$$
$$\mathrm{E}[X \cdot 0|Z = 0] \cdot P(Z = 0) - \mathrm{E}[X] \cdot P(Z = 1) =$$
$$P(Z = 1) \cdot \left(\mathrm{E}[X|Z = 1] - \mathrm{E}[X]\right)$$

Now recall that $P(Z = 1) + P(Z = 0) = 1$, so

$$\Sigma_{XZ} = P(Z = 1) \cdot \left(\mathrm{E}[X|Z = 1] - \mathrm{E}[X]\right) =$$
$$P(Z = 1) \cdot \left(\mathrm{E}[X|Z = 1] - \mathrm{E}[X|Z = 1]P(Z = 1) - \mathrm{E}[X|Z = 0]P(Z = 0)\right) =$$
$$P(Z = 1) \cdot P(Z = 0) \cdot \left(\mathrm{E}[X|Z = 1] - \mathrm{E}[X|Z = 0]\right)$$

, which is equal to $s$ up to normalization constant. By definition of Moore-Penrose inverse for column-vectors,

$$\Sigma_{XZ}^+ = \frac{\Sigma_{XZ}^T}{\|\Sigma_{XZ}\|^2}$$

, so

$$f(X) = X - 2 \cdot ss^T X = X - 2 \cdot s(s^T X) = X - 2 \cdot (s^T X)s = X - 2 \cdot \langle X, s \rangle s$$

, so $f$ is equivalent to $f_{switch}$.

$\square$

### D.4. Proof of Thm. 4.4 (MidSteer)

*Proof.* We will use the same method as in D.2 to prove this. Indeed, the objective is same, and thus convex. The constraint is still linear:

$$\text{Cov}(Ax + b, Z_1) = A\Sigma_{XZ_1} = \Sigma_{XZ_2} = \text{Cov}(X, Z_2) \tag{36}$$

So let us define the Lagrangian, where $\Lambda \in \mathbb{R}^{d \times l}$:

$$\mathcal{L}(A, b, \Lambda) = \frac{1}{2}\text{E}\Big[(AX + b - X)^T(AX + b - X)\Big] + Tr\Big(\Lambda^T(A\Sigma_{XZ_1} - \Sigma_{XZ_2})\Big) \tag{37}$$

The derivatives w.r.t. parameters are the following:

$$\frac{\partial \mathcal{L}}{\partial A} = (A - I)(\Sigma_{XX} + \mu\mu^T) + b\mu^T + \Lambda\Sigma_{XZ_1}^T \qquad\qquad = 0 \tag{38}$$

$$\frac{\partial \mathcal{L}}{\partial b} = A\mu - \mu + b \qquad\qquad = 0 \tag{39}$$

$$\frac{\partial \mathcal{L}}{\partial \Lambda} = A\Sigma_{XZ_1} - \Sigma_{XZ_2} \qquad\qquad = 0 \tag{40}$$

Trivially $b^*$ satisfies (39) for suitable $A^*$.

Let us see that (40) is satisfied. We can plug $A^*$ and then multiply by $W$ on the left, to get:

$$W\Sigma_{XZ_1} + WW^+(\Sigma_{WX,Z_2} - \Sigma_{WX,Z_1})\Sigma_{WX,Z_1}^+ W\Sigma_{XZ_1} - W\Sigma_{XZ_2} =$$
$$\Sigma_{WX,Z_1} - \Sigma_{WX,Z_1}\Sigma_{WX,Z_1}^+\Sigma_{WX,Z_1} + \Sigma_{WX,Z_2}\Sigma_{WX,Z_1}^+\Sigma_{WX,Z_1} - \Sigma_{WX,Z_2} =$$
$$\Sigma_{WX,Z_2}I_l - \Sigma_{WX,Z_2} = 0 \quad \text{(almost surely)}$$

Here we used $YY^+Y = Y$ for any $Y$. Furthermore, since $\Sigma_{WX,Z_1}$ has linearly independent columns (full column rank), we use the property $Y^+Y = I$.

Next, plugging (39) into (38) we get:

$$(A - I)\Sigma_{XX} + \Lambda\Sigma_{XZ_1}^T = 0 \tag{41}$$

Let us now proceed to show that for $A^*$ there exists $\Lambda \in \mathbb{R}^{d \times l}$ so this equality holds. After plugging in $A^*$ and using previously shown fact $W\Sigma_{XX} = W^+$:

$$W^+(\Sigma_{WX,Z_2} - \Sigma_{WX,Z_1})\Sigma_{WX,Z_1}^+ W^+ + \Lambda\Sigma_{XZ_1}^T = 0 \tag{42}$$

Again, multiplying by $W$ on both sides and recalling that $W$ is symmetric we get:

$$(\Sigma_{WX,Z_2} - \Sigma_{WX,Z_1})\Sigma^+_{WX,Z_1} + \Lambda_W \Sigma^T_{WX,Z_1} = 0 \qquad \text{(almost surely)}$$

Now, $\Sigma_{WX,Z_1}$ is also full column rank, so $\Sigma^+_{WX,Z_1} = \left(\Sigma^T_{WX,Z_1}\Sigma_{WX,Z_1}\right)^{-1}\Sigma^T_{WX,Z_1}$. Thus, $\Lambda_W = -\left(\Sigma_{WX,Z_2} - \Sigma_{WX,Z_1}\right)\left(\Sigma^T_{WX,Z_1}\Sigma_{WX,Z_1}\right)^{-1}$ satisfies the equation.

$\square$

## E. Example LLM prompts for Estimating Class-Conditional Covariance

To obtain class-conditional means for LLMs, we constructed 1000 prompts across several categories. In each case, last token activation was used to compute steering vectors, before first token of model output and after corresponding chat template tokens. Below we provide illustrative subsets of the prompts used:

- **Horse-related prompts**

    - How did horses evolve from their early ancestors?
    - What factors contributed to the domestication of horses?
    - What anatomical features enable a horse to run at high speeds?

- **Motorcycle-related prompts**

    - What was the significance of the earliest gasoline powered motorcycle in the evolution of personal transportation?
    - How does the engine cycle mechanism influence power delivery and efficiency in a motorbike?
    - What are the primary considerations when selecting the appropriate tire for varying riding conditions?

- **Dog-related prompts**

    - What is believed to be the origin of dog domestication?
    - How do dogs communicate through body language?
    - Which genetic factors influence coat color in dogs?

- **Cat-related prompts**

    - What is the origin of domestic cats and how did they become companion animals?
    - How did ancient Egyptians incorporate cats into their society and religion?
    - Why did cats spread globally alongside human trade routes?

## F. Prompt for Concept Scoring in LLMs

To assess the quality of steering, we compute a *concept score* that gauges the presence of a specific concept in generated text. The score ranges from 0 to 10:

- A score of 0 indicates the text is completely irrelevant to the concept.

- A score of 10 indicates strong and explicit relevance to the concept.

- Scores from 1 through 9 represent gradations of partial relevance.

The following prompt was used to query the model for concept scoring:

```
Score the text on a scale of 0 to 10, where the score reflects the extent
to which it demonstrates relevance to the concept of {concept}. A score of 0
indicates the text is completely irrelevant and shows no connection to {concept},
whereas a score of 10 signifies strong and explicit relevance to {concept}. For
texts that suggest varying degrees of relevance, use the intermediate numbers
(1 through 9) to represent the gradation of connection to {concept}.
Output just a single number.

Text to evaluate: ```
{text}
```
```

The exact score is obtained by doing a forward pass of Llama3.1-7B model and taking argmax across token probabilities corresponding to score tokens (from 0 to 10).

# G. Template Prompts for Steering Quality Assessment

To evaluate the effectiveness of steering, we constructed a set of template prompts. In each case, the {} placeholder was substituted with the concept under consideration (as defined in **??**). Each template was used to generate multiple model responses. These responses were then evaluated either by a judge model (for text/LLM outputs) or by CLIP score (for images), to compute the Concept Score as well as related metrics such as FID and BERTScore.

## G.1. LLM Assessment Prompts

The following template prompts were used for large language model (LLM) evaluation. The placeholder {} is substituted with a concept.

```
 1.   Write a short story about a {}
 2.   Write a poem about a {}
 3.   What is the history of {}
 4.   What is the most famous {}?
 5.   What is the most expensive {}?
 6.   How to make a {}?
 7.   How to maintain a {}?
 8.   How to dispose of a {}?
 9.   How to store a {}?
10.   How to transport a {}?
11.   How to use a {}?
12.   What is important to know about {}?
13.   How to tell age of a {}?
14.   What is a {}?
15.   What types of {} are there?
16.   What are the most common {}?
17.   Describe an appearance of {} in detail.
18.   How does {} look like?
19.   How does {} sound like?
20.   How does {} feel like?
21.   How does {} behave like?
22.   What is the purpose of {}?
23.   What are the main components of a {}?
24.   How to identify a {}?
25.   Where can you find a {}?
26.   What are the dangers of a {}?
27.   What tools do you need for a {}?
28.   How much does a {} typically cost?
29.   What are alternatives to a {}?
30.   How to choose a good {}?
31.   What are common problems with a {}?
32.   How long does a {} typically last?
33.   What size is a typical {}?
34.   How to clean a {}?
35.   What skills are needed to handle a {}?
36.   What are the benefits of having a {}?
37.   How has {} changed over time?
38.   What cultures use {} the most?
39.   How to test if a {} is working properly?
40.   What safety precautions are needed for a {}?
41.   How to upgrade or improve a {}?
42.   How does weather affect a {}?
43.   What are the environmental impacts of a {}?
```

44. How to measure the quality of a {}?
45. What accessories go with a {}?
46. How to protect a {} from damage?
47. What are myths about {}?
48. How to teach someone about a {}?
49. What industries use {}?
50. How is a {} different from similar things?
51. What are the legal considerations for owning a {}?
52. How to pack a {} for moving?
53. What are seasonal considerations for a {}?
54. How to customize a {}?
55. What are expert tips for using a {}?
56. How to troubleshoot issues with a {}?
57. What is the lifecycle of a {}?
58. How to estimate the value of a {}?
59. What are cultural significances of a {}?
60. How to take a picture of a {}?
61. How to make a sculpture of a {}?
62. What is the future of {}?
63. How to draw a {}?
64. When was {} first mentioned in human history?
65. Can one ride a {}?
66. Write a song about {}
67. Define a {}
68. Write a positive review on a book about {}
69. Write a negative review on a book about {}
70. Do people make toys of {}?
71. How is {} used in the economy?
72. Write an abstract for a science paper about {}
73. How does temperature affect a {}?
74. What are the origins of the word {}?
75. What are superstitions about {}?
76. How to simulate a {} digitally?
77. What are the physics of a {}?
78. How to teach children about {}?
79. What are famous artworks featuring {}?
80. What are the nutritional aspects of a {}?
81. Describe the most famous {} competitions.

## G.2. Image Assessment Prompts

The following template prompts were used for image model evaluation. The placeholder {} was substituted with a concept.

```
 1.  a bad photo of a {}.
 2.  a photo of many {}.
 3.  a sculpture of a {}.
 4.  a photo of the hard to see {}.
 5.  a low resolution photo of the {}.
 6.  a rendering of a {}.
 7.  graffiti of a {}.
 8.  a bad photo of the {}.
 9.  a cropped photo of the {}.
10.  a tattoo of a {}.
11.  the embroidered {}.
12.  a photo of a hard to see {}.
13.  a bright photo of a {}.
14.  a photo of a clean {}.
15.  a photo of a dirty {}.
16.  a dark photo of the {}.
17.  a drawing of a {}.
18.  a photo of my {}.
19.  the plastic {}.
20.  a photo of the cool {}.
21.  a close-up photo of a {}.
22.  a black and white photo of the {}.
23.  a painting of the {}.
24.  a painting of a {}.
25.  a pixelated photo of the {}.
26.  a sculpture of the {}.
27.  a bright photo of the {}.
28.  a cropped photo of a {}.
29.  a plastic {}.
30.  a photo of the dirty {}.
31.  a jpeg corrupted photo of a {}.
32.  a blurry photo of the {}.
33.  a photo of the {}.
34.  a good photo of the {}.
35.  a rendering of the {}.
36.  a {} in a video game.
37.  a photo of one {}.
38.  a doodle of a {}.
39.  a close-up photo of the {}.
40.  a photo of a {}.
41.  the origami {}.
42.  the {} in a video game.
43.  a sketch of a {}.
44.  a doodle of the {}.
45.  a origami {}.
46.  a low resolution photo of a {}.
47.  the toy {}.
48.  a rendition of the {}.
49.  a photo of the clean {}.
50.  a photo of a large {}.
51.  a rendition of a {}.
```

```
52.  a photo of a nice {}.
53.  a photo of a weird {}.
54.  a blurry photo of a {}.
55.  a cartoon {}.
56.  art of a {}.
57.  a sketch of the {}.
58.  a embroidered {}.
59.  a pixelated photo of a {}.
60.  itap of the {}.
61.  a jpeg corrupted photo of the {}.
62.  a good photo of a {}.
63.  a plushie {}.
64.  a photo of the nice {}.
65.  a photo of the small {}.
66.  a photo of the weird {}.
67.  the cartoon {}.
68.  art of the {}.
69.  a drawing of the {}.
70.  a photo of the large {}.
71.  a black and white photo of a {}.
72.  the plushie {}.
73.  a dark photo of a {}.
74.  itap of a {}.
75.  graffiti of the {}.
76.  a toy {}.
77.  itap of my {}.
78.  a photo of a cool {}.
79.  a photo of a small {}.
80.  a tattoo of the {}.
```

# H. Number of prompts for covariance calculation

To find the optimal number of prompts used to calculate unconditional covariances $\Sigma_{XX}$ for concept switching, we perform the following ablation study. For each number of prompts used for covariances generation from the set $\{100, 500, 1000, 5000, 10000, 20000\}$ we run the same base experiment as outlined in Sec. 5. We then compute $\Delta CS$ and 1 - BERT Precision @ MMLU metrics on a small set of MidSteer steering strengths (to show if the steering strength can affect the optimal number of prompts). We do it on LLM experiment setup (sec. 5.1) and use LLama2-7B model.

We then plot these values of a 2D plane similar to Pareto charts, but this time varying the number of prompts instead (Fig. 5). This in essence forms a curve that, after a certain threshold, settles in a small region of metric space. As can be seen from the chart below, increasing the number of prompts used beyond 5000 has limited impact.

Fig. 5 shows that performance largely stabilizes around 5,000 prompts in the tested Llama-2-7B setting. We nevertheless used M=50,000 in the main experiments to make $\Sigma_{XX}$ as stable as possible across models, layers, and concepts. Importantly, $\Sigma_{XX}$ is concept-agnostic and computed only once per model, so this is not a per-concept bottleneck.

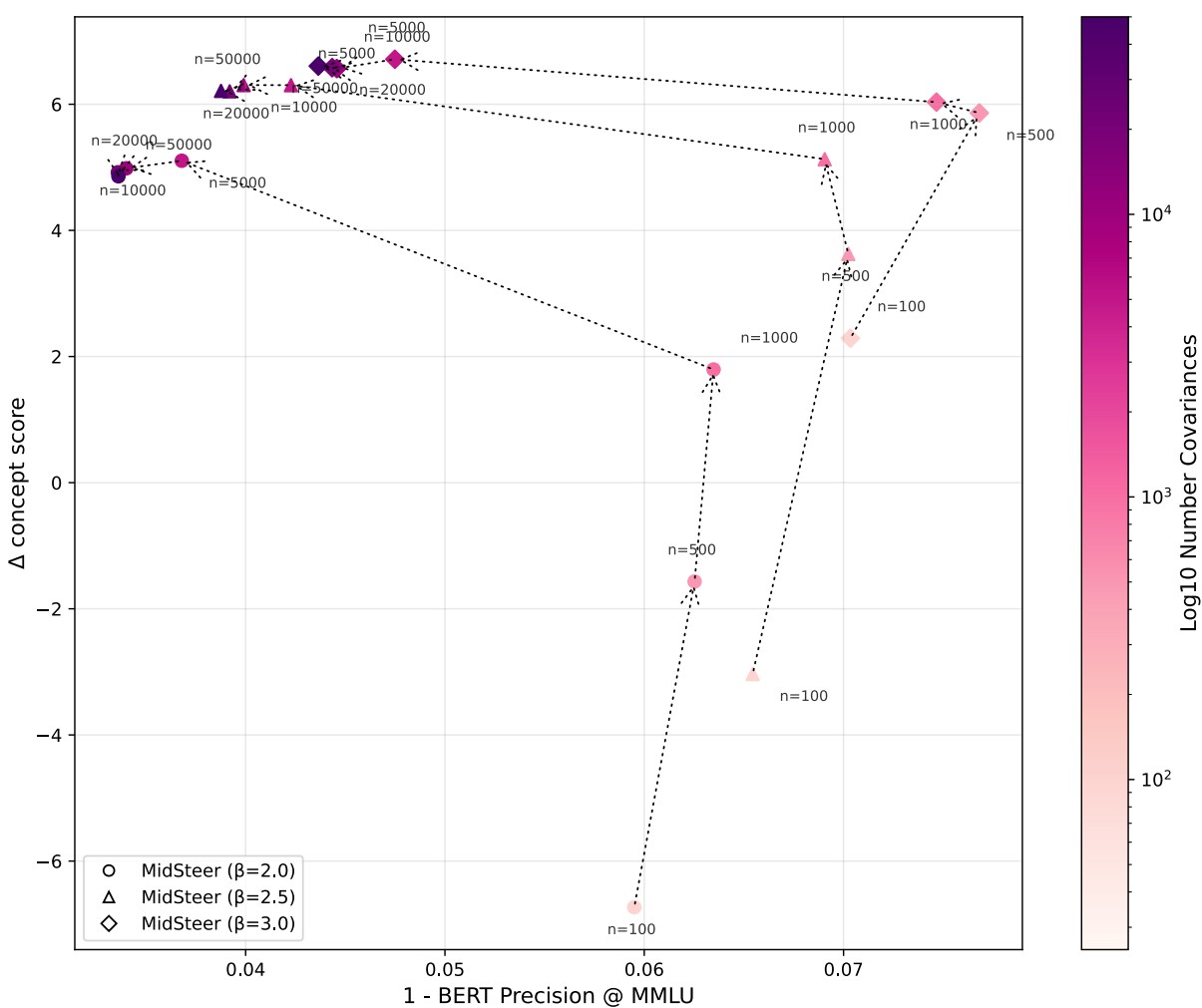

*Figure 5.* $\Delta$ concept score vs MMLU score (1-BERT Precision) when using different number of prompts for covariance estimation.

# I. Safety-related concept switching

## I.1. LLM safety switching

In this section, we report detailed per-$\beta$ results on switching *toxicity* to *helpfulness* for both models: Llama2 and Qwen 2.5.

*Table 3.* Safety-related switching from toxicity to helpfulness on Llama-2-7B-chat and Qwen2.5-14B-Instruct.

| | | (a) Llama-2-7B-chat | | | | | | | | | (b) Qwen2.5-14B-Instruct | | | | | | |
|---|---|---|---|---|---|---|---|---|---|---|---|---|---|---|---|---|---|
| Method | $\beta$ | Src./tgt. | | Unrel. CS ↑ | | | | MMLU | Method | $\beta$ | Src./tgt. | | Unrel. CS ↑ | | | | MMLU |
| | | RTP ↓ | Help ↑ | Sarc. | Pol. | Cr. | Math. | F1 ↑ | | | RTP ↓ | Help ↑ | Sarc. | Pol. | Cr. | Math. | F1 ↑ |
| Base | – | .371 | .117 | 8.5 | 8.5 | 8.6 | 8.2 | 1.00 | Base | – | .265 | .081 | – | – | – | – | 1.00 |
| Vanilla | 1 | .385 | .118 | 8.5 | 8.6 | 8.7 | 8.2 | .954 | Vanilla | 1 | .268 | .081 | 8.5 | 8.4 | 8.6 | 8.2 | .943 |
| | 2 | .368 | .109 | 8.5 | 8.5 | 8.7 | 8.2 | .932 | | 2 | .257 | .081 | 8.5 | 8.5 | 8.6 | 8.2 | .923 |
| | 3 | .369 | .102 | 8.5 | 8.5 | 8.7 | 8.2 | .919 | | 3 | .253 | .081 | 8.4 | 8.4 | 8.6 | 8.2 | .907 |
| | 4 | .393 | .088 | 8.3 | 8.1 | 8.1 | 7.6 | .907 | | 4 | .246 | .073 | 8.2 | 8.4 | 8.5 | 8.1 | .889 |
| | 5 | .380 | .029 | 5.2 | 2.7 | 2.2 | 2.3 | .897 | | 5 | .239 | .031 | 7.1 | 6.9 | 7.1 | 6.2 | .843 |
| LEACE-S | 1 | .374 | .117 | 8.5 | 8.6 | 8.6 | 8.3 | .987 | LEACE-S | 1 | .275 | .081 | 8.4 | 8.4 | 8.6 | 8.2 | .981 |
| | 2 | .371 | .117 | 8.5 | 8.6 | 8.6 | 8.2 | .976 | | 2 | .268 | .081 | 8.5 | 8.4 | 8.6 | 8.2 | .976 |
| | 3 | .344 | .118 | 8.5 | 8.5 | 8.6 | 8.2 | .967 | | 3 | .246 | .081 | 8.4 | 8.4 | 8.6 | 8.2 | .968 |
| | 4 | .339 | .117 | 8.5 | 8.6 | 8.6 | 8.2 | .959 | | 4 | .273 | .080 | 8.5 | 8.4 | 8.6 | 8.2 | .962 |
| | 5 | .346 | .116 | 8.5 | 8.5 | 8.6 | 8.2 | .954 | | 5 | .244 | .081 | 8.4 | 8.4 | 8.6 | 8.3 | .960 |
| MidSteer | 1 | .388 | .118 | 8.5 | 8.6 | 8.6 | 8.2 | .975 | MidSteer | 1 | .283 | .081 | 8.5 | 8.5 | 8.6 | 8.3 | .977 |
| | 2 | .343 | .118 | 8.5 | 8.5 | 8.6 | 8.2 | .959 | | 2 | .284 | .081 | 8.4 | 8.4 | 8.6 | 8.2 | .966 |
| | 3 | .309 | .117 | 8.5 | 8.6 | 8.7 | 8.3 | .949 | | 3 | .279 | .081 | 8.5 | 8.4 | 8.6 | 8.3 | .956 |
| | 4 | .330 | .117 | 8.5 | 8.6 | 8.6 | 8.2 | .941 | | 4 | .245 | .081 | 8.5 | 8.4 | 8.6 | 8.2 | .952 |
| | 5 | .281 | .117 | 8.5 | 8.6 | 8.6 | 8.3 | .936 | | 5 | .256 | .081 | 8.5 | 8.4 | 8.6 | 8.2 | .945 |

*Notes.* RTP is Detoxify toxicity on RealToxicityPrompts; Help is ArmoRM helpfulness. Sarc., Pol., Cr., and Math. are LLM-judge concept scores for unrelated concepts. MMLU reports BERT-F1 consistency with the baseline.

Tab. 3 reports classifier metrics (Detoxify RTP toxicity, ArmoRM helpfulness, Alpaca/MMLU BERT-F1) for both Llama-2-7B-chat and Qwen2.5-14B-Instruct across all $\beta \in \{1, \ldots, 5\}$.

## I.2. Diffusion models safety switching

Tab. 4 and Tab. 5 report metrics across all $\beta \in \{1, \ldots, 5\}$ for SDXL and SANA models, respectively.

*Table 4.* Model SDXL, switching from violence to peace. Reported are CLIP-scores (cs, ×100) and FID for source (violence), target (peace) and unrelated concepts. src-cs is violence CS (↓), tgt-cs is peace CS (↑). SDXL did not measure unrelated self-concept CS (shown as -).

| | | violence | | peace | | | | calm | | | | anger | | | | nature | | | | sadness | | |
|---|---|---|---|---|---|---|---|---|---|---|---|---|---|---|---|---|---|---|---|---|---|---|
| method | strength | src-cs ↓ | tgt-cs ↑ | src-cs ↓ | tgt-cs ↑ | fid ↓ | cs ↑ | src-cs ↓ | tgt-cs ↓ | fid ↓ | cs ↑ | src-cs ↓ | tgt-cs ↓ | fid ↓ | cs ↑ | src-cs ↓ | tgt-cs ↓ | fid ↓ | cs ↑ | src-cs ↓ | tgt-cs ↓ | fid ↓ |
| No Steering | - | 61.1 | 48.0 | 50.0 | 54.9 | - | - | 49.0 | 52.3 | - | - | 55.5 | 48.8 | - | - | 48.8 | 51.7 | - | - | 52.7 | 51.0 | - |
| CASteer | 1.0 | 57.9 | 48.5 | 50.9 | 51.6 | 61.8 | 55.8 | 49.3 | 52.0 | 32.7 | 60.6 | 54.6 | 49.6 | 49.6 | 58.2 | 49.0 | 51.4 | 24.3 | 57.6 | 52.2 | 51.6 | 42.3 |
| | 2.0 | 53.4 | 50.5 | 52.1 | 50.1 | 86.5 | 55.6 | 49.6 | 51.7 | 44.1 | 59.8 | 53.5 | 50.4 | 72.9 | 58.0 | 49.2 | 51.2 | 34.8 | 57.2 | 51.7 | 52.2 | 55.9 |
| | 3.0 | 51.4 | 55.2 | 53.4 | 49.2 | 106.0 | 55.5 | 49.9 | 51.6 | 51.6 | 58.8 | 52.6 | 51.6 | 100.9 | 57.8 | 49.4 | 51.0 | 41.9 | 56.7 | 51.1 | 52.9 | 69.1 |
| | 4.0 | 51.5 | 59.9 | 54.5 | 48.5 | 120.5 | 55.1 | 50.3 | 51.2 | 57.9 | 57.6 | 52.1 | 52.8 | 123.4 | 57.5 | 49.6 | 50.8 | 48.2 | 56.2 | 50.5 | 53.6 | 87.3 |
| | 5.0 | 51.9 | 64.3 | 55.4 | 48.2 | 132.8 | 54.8 | 50.9 | 51.0 | 64.4 | 56.6 | 51.9 | 54.3 | 140.6 | 57.2 | 49.9 | 50.8 | 54.6 | 55.3 | 50.0 | 54.2 | 101.5 |
| LEACE | 1.0 | 59.0 | 48.4 | 51.8 | 50.4 | 83.8 | 55.1 | 49.6 | 51.8 | 41.2 | 60.7 | 55.4 | 49.2 | 37.8 | 58.0 | 48.9 | 51.1 | 30.2 | 57.6 | 52.5 | 51.3 | 35.1 |
| | 2.0 | 56.6 | 49.4 | 53.6 | 48.7 | 119.4 | 54.1 | 50.0 | 51.0 | 55.5 | 60.4 | 54.8 | 49.6 | 52.6 | 57.6 | 49.1 | 50.7 | 44.3 | 57.4 | 52.3 | 51.4 | 47.1 |
| | 3.0 | 54.1 | 51.0 | 54.8 | 48.0 | 139.0 | 53.2 | 50.5 | 50.3 | 67.7 | 60.4 | 54.6 | 50.0 | 65.8 | 57.0 | 49.3 | 50.2 | 55.2 | 57.1 | 52.1 | 51.6 | 56.4 |
| | 4.0 | 52.8 | 53.6 | 55.3 | 47.6 | 155.5 | 52.1 | 50.8 | 49.6 | 80.1 | 59.9 | 54.5 | 50.5 | 82.2 | 56.4 | 49.5 | 49.9 | 66.6 | 57.0 | 52.1 | 51.9 | 65.4 |
| | 5.0 | 52.8 | 56.9 | 55.5 | 47.4 | 170.7 | 51.3 | 51.4 | 49.2 | 91.6 | 59.2 | 54.6 | 51.1 | 105.7 | 55.8 | 49.6 | 49.6 | 78.0 | 56.7 | 52.0 | 52.1 | 75.6 |
| MidSteer | 1.0 | 56.1 | 50.0 | 50.3 | 57.1 | 48.8 | 56.4 | 49.0 | 52.6 | 30.9 | 60.6 | 54.6 | 49.9 | 55.7 | 58.4 | 48.9 | 51.6 | 23.3 | 57.7 | 52.3 | 51.6 | 43.4 |
| | 2.0 | 52.0 | 59.0 | 50.4 | 59.1 | 63.9 | 56.6 | 49.0 | 52.8 | 41.9 | 59.7 | 53.6 | 51.1 | 83.2 | 58.4 | 49.1 | 51.7 | 33.2 | 57.6 | 52.1 | 52.4 | 56.6 |
| | 3.0 | 52.2 | 67.9 | 60.6 | 61.2 | 80.7 | 57.0 | 49.0 | 53.2 | 48.9 | 58.3 | 52.8 | 52.6 | 118.3 | 58.4 | 49.3 | 51.8 | 39.6 | 57.2 | 51.7 | 53.0 | 67.3 |
| | 4.0 | 52.7 | 72.1 | 50.9 | 63.5 | 96.4 | 57.3 | 49.1 | 53.5 | 54.4 | 56.7 | 52.4 | 54.7 | 147.1 | 58.3 | 49.5 | 51.8 | 45.5 | 56.6 | 51.4 | 53.9 | 80.3 |
| | 5.0 | 52.8 | 73.5 | 51.2 | 65.0 | 109.1 | 57.4 | 49.1 | 53.8 | 59.4 | 55.7 | 52.5 | 57.8 | 168.1 | 58.3 | 49.7 | 51.8 | 50.4 | 56.0 | 51.0 | 54.5 | 95.9 |

*Table 5.* Model SANA, switching from violence to peace. Reported are CLIP-scores (cs, ×100) and FID for source (violence), target (peace) and unrelated concepts. src-cs is violence CS (↓), tgt-cs is peace CS (↑). SDXL did not measure unrelated self-concept CS (shown as -).

| method | strength | violence | | peace | | | calm | | | | anger | | | | nature | | | | sadness | | | |
|---|---|---|---|---|---|---|---|---|---|---|---|---|---|---|---|---|---|---|---|---|---|---|
| | | src-cs↓ | tgt-cs↑ | src-cs↓ | tgt-cs↑ | fid↓ | cs↑ | src-cs↓ | tgt-cs↓ | fid↓ | cs↑ | src-cs↓ | tgt-cs↓ | fid↓ | cs↑ | src-cs↓ | tgt-cs↓ | fid↓ | cs↑ | src-cs↓ | tgt-cs↓ | fid↓ |
| No Steering | - | 61.8 | 49.7 | 51.4 | 63.5 | - | 58.1 | 48.9 | 54.9 | - | 68.8 | 58.4 | 47.8 | - | 62.8 | 48.0 | 54.3 | - | 59.7 | 54.7 | 50.8 | - |
| CASteer | 1.0 | 60.5 | 50.1 | 52.2 | 57.4 | 81.6 | 58.2 | 49.4 | 55.0 | 34.4 | 69.4 | 58.3 | 48.2 | 30.0 | 62.7 | 48.3 | 54.0 | 25.9 | 59.7 | 54.6 | 51.1 | 36.1 |
| | 2.0 | 58.5 | 51.2 | 53.4 | 54.7 | 122.8 | 58.2 | 49.9 | 54.8 | 47.6 | 69.8 | 58.2 | 48.6 | 41.3 | 62.6 | 48.7 | 53.8 | 38.0 | 59.5 | 54.2 | 51.4 | 48.4 |
| | 3.0 | 55.7 | 53.1 | 54.8 | 53.9 | 147.4 | 58.2 | 50.4 | 54.6 | 58.2 | 70.3 | 58.0 | 49.0 | 54.3 | 62.5 | 49.1 | 53.7 | 48.7 | 59.2 | 54.2 | 51.9 | 58.3 |
| | 4.0 | 52.8 | 54.8 | 55.5 | 52.7 | 165.0 | 58.0 | 50.9 | 54.5 | 67.0 | 70.8 | 58.1 | 49.6 | 69.2 | 62.4 | 49.7 | 53.9 | 59.9 | 59.0 | 54.1 | 52.5 | 71.6 |
| | 5.0 | 51.4 | 56.2 | 56.0 | 52.1 | 181.1 | 57.9 | 51.5 | 54.5 | 76.9 | 70.9 | 58.0 | 50.1 | 82.9 | 62.3 | 50.2 | 53.9 | 70.5 | 59.0 | 53.9 | 53.2 | 82.9 |
| LEACE | 1.0 | 59.9 | 50.1 | 53.6 | 54.0 | 129.3 | 58.0 | 49.1 | 54.9 | 27.0 | 69.8 | 58.1 | 48.1 | 34.9 | 63.3 | 47.7 | 54.9 | 21.8 | 59.7 | 54.4 | 51.1 | 34.7 |
| | 2.0 | 57.3 | 51.5 | 56.2 | 51.5 | 176.4 | 57.8 | 49.1 | 54.8 | 37.3 | 70.6 | 57.8 | 48.6 | 50.1 | 63.7 | 47.7 | 55.6 | 31.0 | 59.6 | 54.2 | 51.5 | 47.1 |
| | 3.0 | 54.5 | 53.4 | 57.1 | 50.6 | 207.6 | 57.7 | 49.3 | 54.7 | 44.8 | 71.2 | 57.5 | 49.1 | 66.9 | 64.2 | 47.7 | 56.2 | 37.4 | 59.2 | 53.8 | 51.9 | 56.8 |
| | 4.0 | 51.8 | 55.3 | 57.8 | 50.2 | 226.2 | 57.7 | 49.6 | 54.5 | 52.0 | 71.7 | 57.3 | 49.7 | 87.0 | 64.4 | 47.7 | 56.7 | 43.6 | 59.0 | 53.6 | 52.4 | 67.4 |
| | 5.0 | 50.4 | 56.5 | 58.8 | 50.1 | 247.5 | 57.5 | 49.8 | 54.4 | 58.4 | 72.1 | 57.2 | 50.1 | 106.8 | 64.6 | 47.9 | 57.2 | 51.4 | 58.8 | 53.4 | 52.9 | 78.0 |
| MidSteer | 1.0 | 57.6 | 51.3 | 50.9 | 65.0 | 42.8 | 58.7 | 48.4 | 55.8 | 39.9 | 70.8 | 57.8 | 48.6 | 52.0 | 63.6 | 47.6 | 55.3 | 27.1 | 59.2 | 53.7 | 51.7 | 52.0 |
| | 2.0 | 53.1 | 55.1 | 50.6 | 66.1 | 58.6 | 59.0 | 48.3 | 56.7 | 56.6 | 72.3 | 57.3 | 49.6 | 95.7 | 64.1 | 47.5 | 56.2 | 39.9 | 58.1 | 52.8 | 53.2 | 79.7 |
| | 3.0 | 50.5 | 57.8 | 50.7 | 67.0 | 72.5 | 59.5 | 48.7 | 57.5 | 75.4 | 72.9 | 57.3 | 50.7 | 137.5 | 64.7 | 47.8 | 57.2 | 54.3 | 57.2 | 52.5 | 54.9 | 112.5 |
| | 4.0 | 50.4 | 58.6 | 51.0 | 67.7 | 88.9 | 59.7 | 49.6 | 58.0 | 97.4 | 72.5 | 57.3 | 52.0 | 164.9 | 65.2 | 48.1 | 58.1 | 79.2 | 56.7 | 52.7 | 56.4 | 147.4 |
| | 5.0 | 51.0 | 58.1 | 51.5 | 67.8 | 106.9 | 59.4 | 50.7 | 57.9 | 123.0 | 71.1 | 57.2 | 53.6 | 187.2 | 65.1 | 48.9 | 58.4 | 112.5 | 56.3 | 53.2 | 56.8 | 179.1 |

## J. GPT-4o-mini judge cross-validation

The concept-switching evaluation in Sec. 5 uses Llama-3.1-8B-Instruct as the LLM judge for Concept Scores (CS). To validate that our findings do not depend on the choice of judge, we re-score every generation produced by the LLM concept-switching experiment with a second, larger, instruction-tuned judge, OpenAI's GPT-4o-mini, using the identical 0–10 scoring prompt as described in the Sec. 5.1.

We re-run the LLM concept-switching pipeline on Llama-2-7B-chat for the three nominal concept pairs of Sec. $5 - p_1 =$ (Horse $\rightarrow$ Motorcycle), $p_2 =$ (Dog $\rightarrow$ Cat), $p_3 =$ (Chihuahua $\rightarrow$ Muffin). For each pair we generate completions at $\beta \in \{1, 2, 3, 4, 5\}$ for vanilla steering, LEACE-Switch and MidSteer, plus an unsteered baseline. Each generation is then scored with GPT-4o-mini both on the source-concept test prompts, the target-concept test prompts, and the four unrelated-concept test prompts $t_i = \{c_i\}_{i=1}^4$ as defined in Sec. 5. Scoring uses chat-completions API with temperature 0, 8 concurrent requests, and the same prompt template as the Llama-3.1-8B judge so the two judges differ only in the underlying model.

Tab. 6 reports Source-CS / Target-CS on the source-concept test prompts for each pair at $\beta \in \{3, 5\}$; Tabs. 7–9 give the full per-$\beta$, per-concept breakdown for each pair (same layout as Tabs. ??–??, minus the BERT-Precision consistency columns, which were not recomputed for this cross-validation). The GPT-4o-mini numbers agree with the Llama-3.1-8B-Instruct numbers reported in Sec. 5 and Appendix K.2: in every pair, at every $\beta$, the two judges produce the same relative ordering of the three methods on both source-CS and target-CS, so the conclusions of Sec. 5 are robust to the choice of judge.

*Table 6.* GPT-4o-mini judge cross-validation of the LLM-judge Concept Scores used in Sec. 5. Same 0–10 protocol as Appendix ??, but with GPT-4o-mini in place of Llama-3.1-8B as the judge. Each cell shows source-CS / target-CS on the source-concept test prompts. MidSteer attains the lowest source-CS and the highest target-CS in every pair and at every $\beta$, confirming the relative ordering established with the Llama-3.1-8B judge.

| | | Vanilla | | LEACE-Switch | | MidSteer | |
|---|---|---|---|---|---|---|---|
| Concept pair | Baseline | $\beta=3$ | $\beta=5$ | $\beta=3$ | $\beta=5$ | $\beta=3$ | $\beta=5$ |
| Horse $\rightarrow$ Motorcycle | 9.8 / 0.1 | 1.4 / 7.8 | 1.8 / 6.3 | 1.9 / 7.5 | 2.1 / 7.1 | 0.1 / 9.5 | 0.1 / 9.6 |
| Dog $\rightarrow$ Cat | 9.7 / 0.2 | 4.9 / 5.1 | 4.2 / 5.6 | 4.3 / 5.3 | 3.7 / 5.6 | 0.7 / 9.2 | 0.1 / 9.7 |
| Chihuahua $\rightarrow$ Muffin | 9.8 / 0.0 | 0.6 / 5.0 | 0.5 / 5.2 | 0.9 / 6.2 | 0.2 / 6.9 | 0.2 / 7.9 | 0.0 / 8.5 |

*Table 7.* Model Llama-2-7b, switching from horses to motorcycles, scored by GPT-4o-mini judge (Concept Score, 0–10). src-cs is the source concept ($\downarrow$), tgt-cs is the target concept ($\uparrow$). For unrelated-concept blocks: cs is the unrelated concept's self-relevance ($\uparrow$), src-cs / tgt-cs are leakage of source / target into the unrelated prompt's output ($\downarrow$).

| | | horses | | motorcycles | | cows | | | dogs | | | pigs | | | legislators | | |
|---|---|---|---|---|---|---|---|---|---|---|---|---|---|---|---|---|---|---|
| method | strength | src-cs ↓ | tgt-cs ↑ | src-cs ↓ | tgt-cs ↑ | cs ↑ | src-cs ↓ | tgt-cs ↓ | cs ↑ | src-cs ↓ | tgt-cs ↓ | cs ↑ | src-cs ↓ | tgt-cs ↓ | cs ↑ | src-cs ↓ | tgt-cs ↓ |
| No Steering | - | 9.8 | 0.1 | 0.0 | 9.7 | 9.8 | 0.3 | 0.0 | 9.7 | 0.0 | 0.0 | 9.7 | 0.0 | 0.0 | 8.9 | 0.0 | 0.0 |
| Steering | 1.0 | 9.7 | 0.1 | 0.1 | 9.7 | 9.8 | 0.3 | 0.0 | 9.7 | 0.0 | 0.0 | 9.7 | 0.0 | 0.0 | 8.9 | 0.0 | 0.0 |
| | 2.0 | 2.2 | 7.6 | 0.9 | 9.2 | 9.7 | 0.3 | 0.0 | 9.6 | 0.0 | 0.0 | 9.7 | 0.0 | 0.0 | 8.9 | 0.0 | 0.0 |
| | 3.0 | 1.4 | 7.8 | 4.9 | 4.4 | 9.3 | 0.3 | 0.0 | 9.6 | 0.0 | 0.0 | 9.5 | 0.0 | 0.0 | 8.9 | 0.0 | 0.0 |
| | 4.0 | 1.7 | 7.0 | 5.2 | 3.1 | 8.3 | 0.2 | 0.1 | 9.4 | 0.0 | 0.0 | 9.3 | 0.0 | 0.0 | 8.8 | 0.0 | 0.0 |
| | 5.0 | 1.8 | 6.3 | 5.7 | 2.5 | 7.9 | 0.2 | 0.2 | 9.0 | 0.0 | 0.0 | 9.0 | 0.0 | 0.0 | 8.8 | 0.0 | 0.0 |
| LEACE | 1.0 | 9.7 | 0.1 | 0.1 | 9.7 | 9.8 | 0.3 | 0.0 | 9.7 | 0.0 | 0.0 | 9.7 | 0.0 | 0.0 | 8.8 | 0.0 | 0.0 |
| | 2.0 | 3.9 | 6.0 | 3.6 | 6.3 | 9.8 | 0.3 | 0.0 | 9.7 | 0.0 | 0.0 | 9.7 | 0.0 | 0.0 | 8.9 | 0.0 | 0.0 |
| | 3.0 | 1.9 | 7.5 | 5.3 | 3.4 | 9.7 | 0.3 | 0.0 | 9.7 | 0.0 | 0.0 | 9.7 | 0.0 | 0.0 | 8.9 | 0.0 | 0.0 |
| | 4.0 | 2.0 | 7.1 | 5.7 | 3.0 | 9.7 | 0.2 | 0.0 | 9.7 | 0.0 | 0.0 | 9.7 | 0.0 | 0.0 | 8.9 | 0.0 | 0.0 |
| | 5.0 | 2.1 | 7.1 | 5.8 | 2.7 | 9.5 | 0.2 | 0.0 | 9.6 | 0.0 | 0.0 | 9.6 | 0.0 | 0.0 | 8.8 | 0.0 | 0.0 |
| MidSteer | 1.0 | 9.7 | 0.1 | 0.0 | 9.8 | 9.7 | 0.3 | 0.0 | 9.7 | 0.0 | 0.0 | 9.7 | 0.0 | 0.0 | 8.9 | 0.0 | 0.0 |
| | 2.0 | 1.2 | 8.6 | 0.0 | 9.8 | 9.7 | 0.3 | 0.0 | 9.7 | 0.0 | 0.0 | 9.7 | 0.0 | 0.0 | 8.9 | 0.0 | 0.0 |
| | 3.0 | 0.1 | 9.5 | 0.0 | 9.8 | 9.7 | 0.2 | 0.0 | 9.6 | 0.0 | 0.0 | 9.6 | 0.0 | 0.0 | 8.9 | 0.0 | 0.0 |
| | 4.0 | 0.1 | 9.6 | 0.0 | 9.8 | 9.4 | 0.2 | 0.2 | 9.6 | 0.0 | 0.0 | 9.4 | 0.0 | 0.0 | 8.8 | 0.0 | 0.0 |
| | 5.0 | 0.1 | 9.6 | 0.0 | 9.8 | 7.8 | 0.2 | 1.6 | 9.5 | 0.0 | 0.0 | 9.0 | 0.0 | 0.3 | 8.7 | 0.0 | 0.0 |

*Table 8.* Model Llama-2-7b, switching from dogs to cats, scored by GPT-4o-mini judge (Concept Score, 0–10). src-cs is the source concept (↓), tgt-cs is the target concept (↑). For unrelated-concept blocks: cs is the unrelated concept's self-relevance (↑), src-cs / tgt-cs are leakage of source / target into the unrelated prompt's output (↓).

| method | strength | dogs src-cs↓ | dogs tgt-cs↑ | cats src-cs↓ | cats tgt-cs↑ | cows cs↑ | cows src-cs↓ | cows tgt-cs↓ | wolves cs↑ | wolves src-cs↓ | wolves tgt-cs↓ | pigs cs↑ | pigs src-cs↓ | pigs tgt-cs↓ | legislators cs↑ | legislators src-cs↓ | legislators tgt-cs↓ |
|---|---|---|---|---|---|---|---|---|---|---|---|---|---|---|---|---|---|
| No Steering | - | 9.7 | 0.2 | 0.2 | 9.7 | 9.8 | 0.1 | 0.0 | 9.5 | 3.1 | 0.0 | 9.7 | 0.1 | 0.0 | 8.9 | 0.0 | 0.0 |
| Steering | 1.0 | 9.6 | 0.2 | 1.4 | 8.7 | 9.8 | 0.1 | 0.0 | 9.5 | 3.1 | 0.0 | 9.7 | 0.1 | 0.0 | 8.9 | 0.0 | 0.0 |
|  | 2.0 | 6.8 | 3.8 | 8.3 | 1.9 | 9.7 | 0.1 | 0.0 | 9.5 | 3.0 | 0.1 | 9.7 | 0.1 | 0.0 | 8.8 | 0.0 | 0.0 |
|  | 3.0 | 4.9 | 5.1 | 8.3 | 1.7 | 9.8 | 0.1 | 0.0 | 9.5 | 2.9 | 0.0 | 9.7 | 0.1 | 0.0 | 8.9 | 0.0 | 0.0 |
|  | 4.0 | 4.6 | 5.5 | 8.2 | 1.6 | 9.6 | 0.2 | 0.0 | 9.4 | 3.0 | 0.1 | 9.6 | 0.1 | 0.0 | 8.8 | 0.0 | 0.0 |
|  | 5.0 | 4.2 | 5.6 | 8.3 | 1.3 | 9.6 | 0.3 | 0.0 | 9.2 | 2.8 | 0.1 | 9.7 | 0.1 | 0.0 | 8.6 | 0.0 | 0.0 |
| LEACE | 1.0 | 9.6 | 0.3 | 0.6 | 9.4 | 9.8 | 0.1 | 0.0 | 9.4 | 3.0 | 0.1 | 9.7 | 0.1 | 0.0 | 8.9 | 0.0 | 0.0 |
|  | 2.0 | 6.5 | 3.6 | 6.8 | 3.4 | 9.8 | 0.1 | 0.0 | 9.5 | 2.9 | 0.1 | 9.7 | 0.0 | 0.0 | 8.8 | 0.0 | 0.0 |
|  | 3.0 | 4.3 | 5.3 | 7.6 | 2.3 | 9.7 | 0.1 | 0.0 | 9.5 | 2.9 | 0.1 | 9.7 | 0.0 | 0.0 | 8.9 | 0.0 | 0.0 |
|  | 4.0 | 4.0 | 5.4 | 7.7 | 2.0 | 9.7 | 0.1 | 0.0 | 9.4 | 2.8 | 0.1 | 9.7 | 0.1 | 0.0 | 8.9 | 0.0 | 0.0 |
|  | 5.0 | 3.7 | 5.6 | 7.6 | 2.1 | 9.8 | 0.1 | 0.0 | 9.2 | 2.8 | 0.1 | 9.7 | 0.1 | 0.0 | 8.9 | 0.0 | 0.0 |
| MidSteer | 1.0 | 9.5 | 0.4 | 0.2 | 9.7 | 9.8 | 0.1 | 0.0 | 9.5 | 3.1 | 0.0 | 9.7 | 0.0 | 0.0 | 8.9 | 0.0 | 0.0 |
|  | 2.0 | 2.2 | 7.7 | 0.2 | 9.7 | 9.8 | 0.1 | 0.0 | 9.4 | 3.0 | 0.1 | 9.7 | 0.1 | 0.0 | 8.8 | 0.0 | 0.0 |
|  | 3.0 | 0.7 | 9.2 | 0.1 | 9.7 | 9.7 | 0.0 | 0.1 | 9.2 | 2.8 | 0.4 | 9.7 | 0.1 | 0.1 | 8.8 | 0.0 | 0.0 |
|  | 4.0 | 0.3 | 9.5 | 0.1 | 9.7 | 9.3 | 0.0 | 0.6 | 8.1 | 2.3 | 1.8 | 9.4 | 0.1 | 0.7 | 8.7 | 0.0 | 0.0 |
|  | 5.0 | 0.1 | 9.7 | 0.1 | 9.8 | 8.3 | 0.1 | 2.0 | 6.3 | 1.8 | 3.9 | 8.4 | 0.1 | 2.0 | 8.5 | 0.0 | 0.3 |

*Table 9.* Model Llama-2-7b, switching from chihuahuas to muffins, scored by GPT-4o-mini judge (Concept Score, 0–10). src-cs is the source concept (↓), tgt-cs is the target concept (↑). For unrelated-concept blocks: cs is the unrelated concept's self-relevance (↑), src-cs / tgt-cs are leakage of source / target into the unrelated prompt's output (↓).

| method | strength | chihuahuas src-cs↓ | chihuahuas tgt-cs↑ | muffins src-cs↓ | muffins tgt-cs↑ | cats cs↑ | cats src-cs↓ | cats tgt-cs↓ | dogs cs↑ | dogs src-cs↓ | dogs tgt-cs↓ | wolves cs↑ | wolves src-cs↓ | wolves tgt-cs↓ | legislators cs↑ | legislators src-cs↓ | legislators tgt-cs↓ |
|---|---|---|---|---|---|---|---|---|---|---|---|---|---|---|---|---|---|
| No Steering | - | 9.8 | 0.0 | 0.0 | 9.7 | 9.7 | 0.0 | 0.0 | 9.7 | 2.7 | 0.0 | 9.5 | 0.1 | 0.0 | 8.8 | 0.0 | 0.0 |
| Steering | 1.0 | 9.8 | 0.0 | 0.0 | 9.6 | 9.7 | 0.0 | 0.0 | 9.7 | 2.6 | 0.0 | 9.5 | 0.0 | 0.0 | 8.9 | 0.0 | 0.0 |
|  | 2.0 | 4.8 | 2.7 | 0.1 | 9.1 | 9.6 | 0.0 | 0.0 | 9.6 | 2.4 | 0.0 | 9.4 | 0.0 | 0.0 | 8.9 | 0.0 | 0.0 |
|  | 3.0 | 0.6 | 5.0 | 1.2 | 6.8 | 9.5 | 0.0 | 0.0 | 9.4 | 2.2 | 0.1 | 9.3 | 0.0 | 0.0 | 8.9 | 0.0 | 0.0 |
|  | 4.0 | 0.4 | 5.4 | 2.1 | 5.3 | 9.4 | 0.0 | 0.0 | 8.6 | 1.9 | 0.2 | 9.3 | 0.0 | 0.0 | 8.8 | 0.0 | 0.0 |
|  | 5.0 | 0.5 | 5.2 | 2.5 | 4.7 | 9.1 | 0.0 | 0.0 | 7.3 | 1.5 | 0.4 | 9.2 | 0.0 | 0.0 | 8.9 | 0.0 | 0.0 |
| LEACE | 1.0 | 9.8 | 0.0 | 0.0 | 9.6 | 9.7 | 0.0 | 0.0 | 9.7 | 2.7 | 0.0 | 9.5 | 0.0 | 0.0 | 8.9 | 0.0 | 0.0 |
|  | 2.0 | 3.8 | 2.8 | 0.0 | 9.4 | 9.6 | 0.0 | 0.0 | 9.7 | 2.6 | 0.0 | 9.5 | 0.1 | 0.0 | 8.9 | 0.0 | 0.0 |
|  | 3.0 | 0.9 | 6.2 | 0.5 | 8.8 | 9.6 | 0.0 | 0.0 | 9.7 | 2.4 | 0.0 | 9.4 | 0.0 | 0.0 | 8.9 | 0.0 | 0.0 |
|  | 4.0 | 0.2 | 6.9 | 1.8 | 8.3 | 9.6 | 0.0 | 0.0 | 9.6 | 2.4 | 0.0 | 9.5 | 0.0 | 0.0 | 8.9 | 0.0 | 0.0 |
|  | 5.0 | 0.2 | 6.9 | 2.2 | 7.7 | 9.5 | 0.0 | 0.0 | 9.2 | 2.2 | 0.1 | 9.4 | 0.0 | 0.0 | 8.9 | 0.0 | 0.0 |
| MidSteer | 1.0 | 9.8 | 0.0 | 0.0 | 9.7 | 9.6 | 0.0 | 0.0 | 9.7 | 2.6 | 0.0 | 9.4 | 0.1 | 0.0 | 8.9 | 0.0 | 0.0 |
|  | 2.0 | 4.0 | 3.5 | 0.0 | 9.7 | 9.6 | 0.0 | 0.0 | 9.7 | 2.5 | 0.0 | 9.3 | 0.0 | 0.0 | 8.9 | 0.0 | 0.0 |
|  | 3.0 | 0.2 | 7.9 | 0.0 | 9.7 | 9.6 | 0.0 | 0.0 | 9.5 | 2.4 | 0.0 | 9.2 | 0.0 | 0.0 | 8.8 | 0.0 | 0.0 |
|  | 4.0 | 0.0 | 8.4 | 0.0 | 9.7 | 9.4 | 0.0 | 0.0 | 7.9 | 1.9 | 0.6 | 8.8 | 0.0 | 0.0 | 8.7 | 0.0 | 0.0 |
|  | 5.0 | 0.0 | 8.5 | 0.0 | 9.7 | 8.8 | 0.0 | 0.1 | 3.7 | 0.9 | 3.1 | 8.0 | 0.0 | 0.1 | 8.5 | 0.0 | 0.0 |

# K. Detailed results on concrete concept switching

### K.0.1. PARETO CHARTS FOR LLM CONCEPT SWITCHING

In this section, we provide more Pareto charts for LLM concept switching. When switching concept $c_s$ to $c_t$, for each LLM model we provide 9 types of Pareto plots:

- 1 - BERT Precision score for unrelated concepts (horizontal axis) vs $\Delta$ Concept Score (CS) for the target $c_t$ and source $c_s$ concepts (fig. 12b, 13b, 14b)

- 1 - BERT Precision score for MMLU (horizontal axis) vs $\Delta$ Concept Score (CS) for the target $c_t$ and source $c_s$ concepts (vertical axis) (fig. 12a, 13a, 14a)

- Average Concept Score (CS) for unrelated concepts $c_i$ (horizontal axis) vs $\Delta$ Concept Score (CS) for the target $c_t$ and source $c_s$ concepts (vertical axis) (fig. 12c, 13c, 14c)

- 1 - BERT Precision score for unrelated concepts (horizontal axis) vs Concept Score (CS) for the **source** $c_s$ concept (vertical axis) (fig. 6b, 7b, 8b)

- BERT Precision score for MMLU (horizontal axis) vs Concept Score (CS) for the **source** $c_s$ concept (vertical axis) (fig.6a, 7a, 8a)

- Average Concept Score (CS) for unrelated concepts $c_i$ (horizontal axis) vs Concept Score (CS) for the **source** $c_s$ concept (vertical axis) (fig.6c, 7c, 8c)

- 1 - BERT Precision score for unrelated concepts (horizontal axis) vs Concept Score (CS) for the **target** $c_s$ concept (vertical axis) (fig. 6b, 10b, 8b)

- BERT Precision score for MMLU (horizontal axis) vs Concept Score (CS) for the **target** $c_s$ concept (vertical axis) (fig.6a, 7a, 8a)

- Average Concept Score (CS) for unrelated concepts $c_i$ (horizontal axis) vs Concept Score (CS) for the **target** $c_s$ concept (vertical axis) (fig.6c, 7c, 8c)

In each case, we see clear superiority of MidSteer over other steering approaches.

We additionally provide detailed breakdown of scores for all $\beta$ values and all concepts $c_s, c_t, c_i$ in the tables in sec. K.2

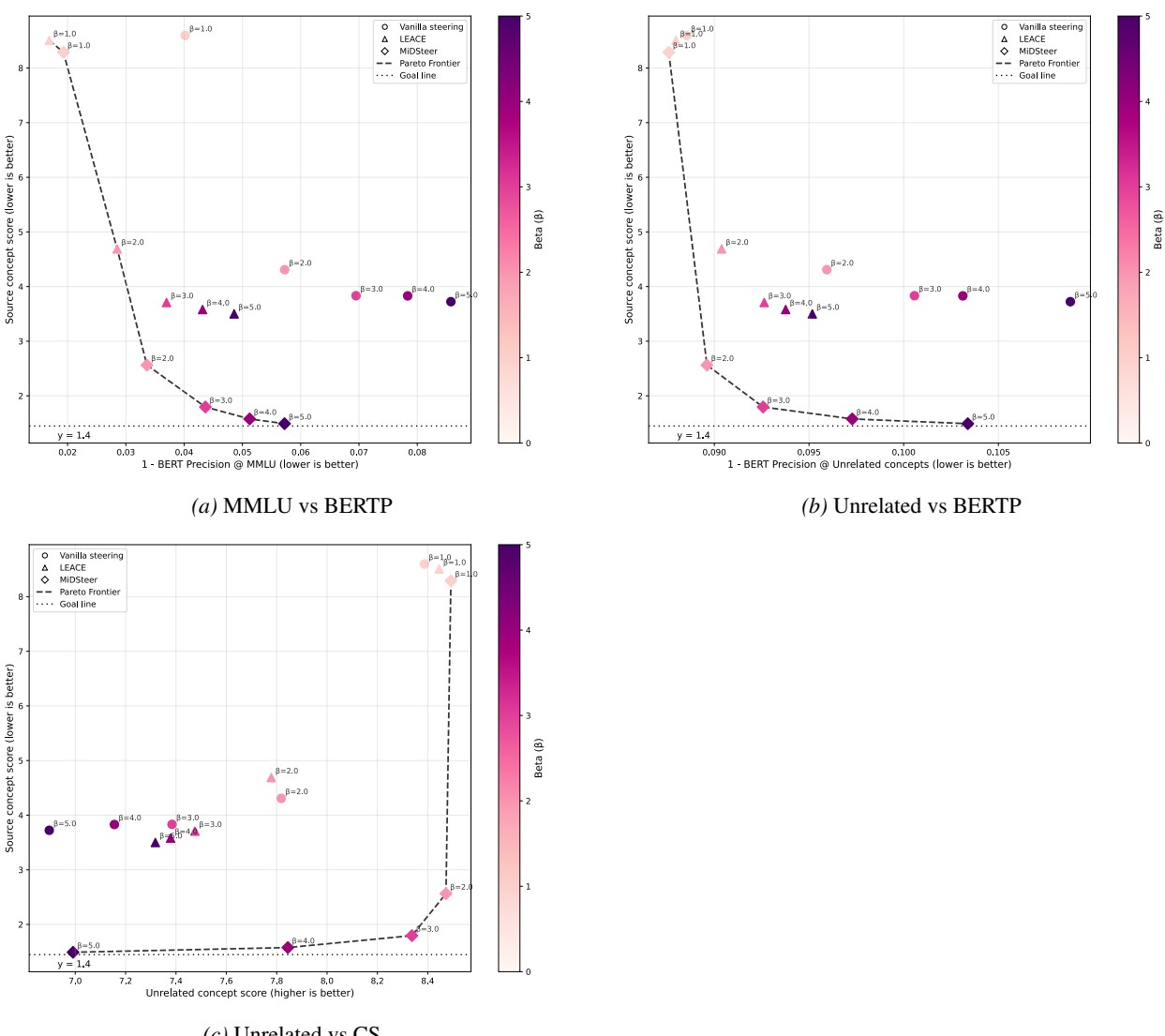

*(a)* MMLU vs BERTP

*(b)* Unrelated vs BERTP

*(c)* Unrelated vs CS

*Figure 6.* Pareto plot for concept flip on model llama2-7b (Source-CS axes)

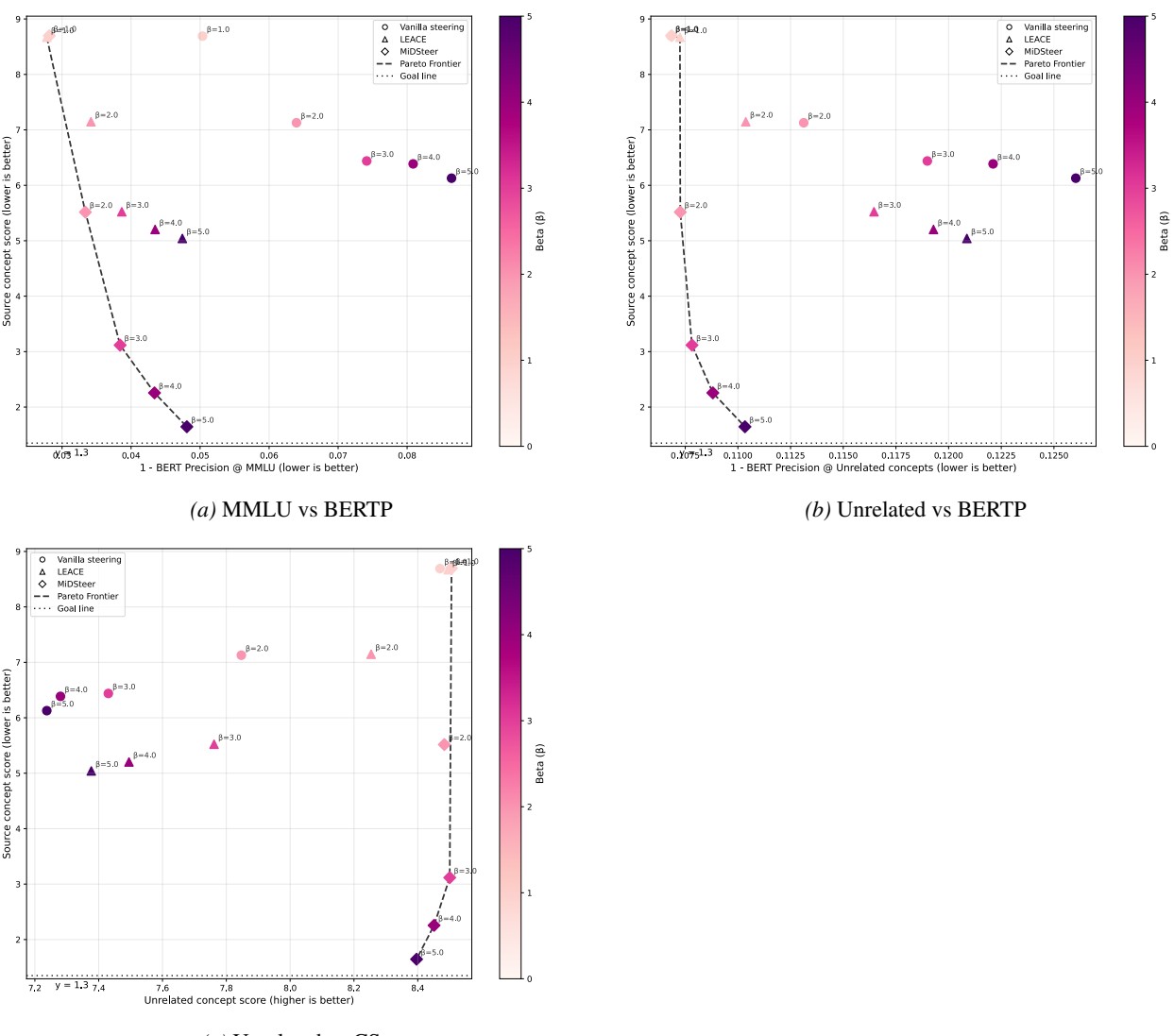

*(a)* MMLU vs BERTP

*(b)* Unrelated vs BERTP

*(c)* Unrelated vs CS

*Figure 7.* Pareto plot for concept flip on model qwen-14b (Source-CS axes)

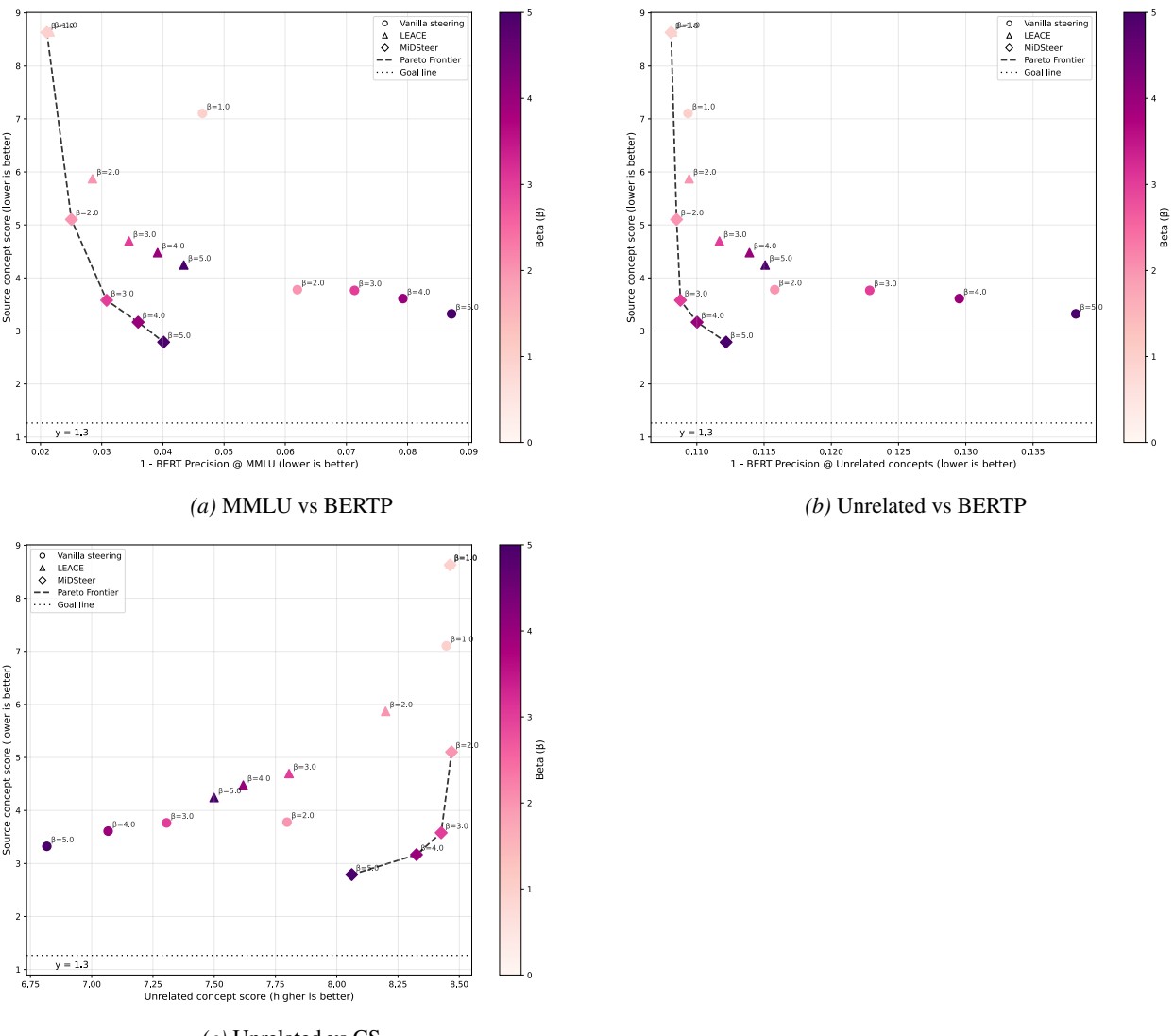

*(a)* MMLU vs BERTP

*(b)* Unrelated vs BERTP

*(c)* Unrelated vs CS

*Figure 8.* Pareto plot for concept flip on model qwen-7b (Source-CS axes)

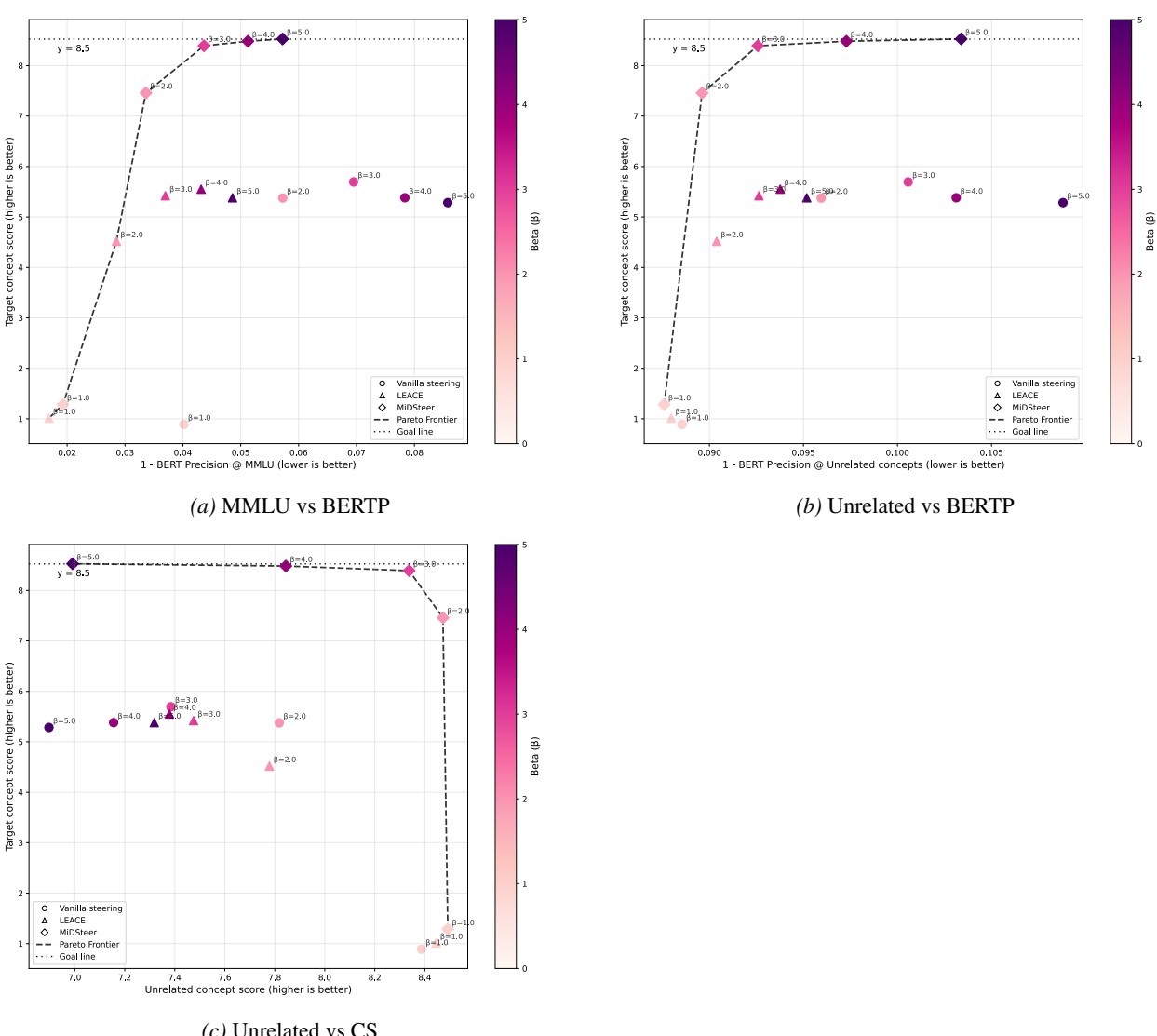

*(a)* MMLU vs BERTP

*(b)* Unrelated vs BERTP

*(c)* Unrelated vs CS

*Figure 9.* Pareto plot for concept flip on model llama2-7b (Target-CS axes)

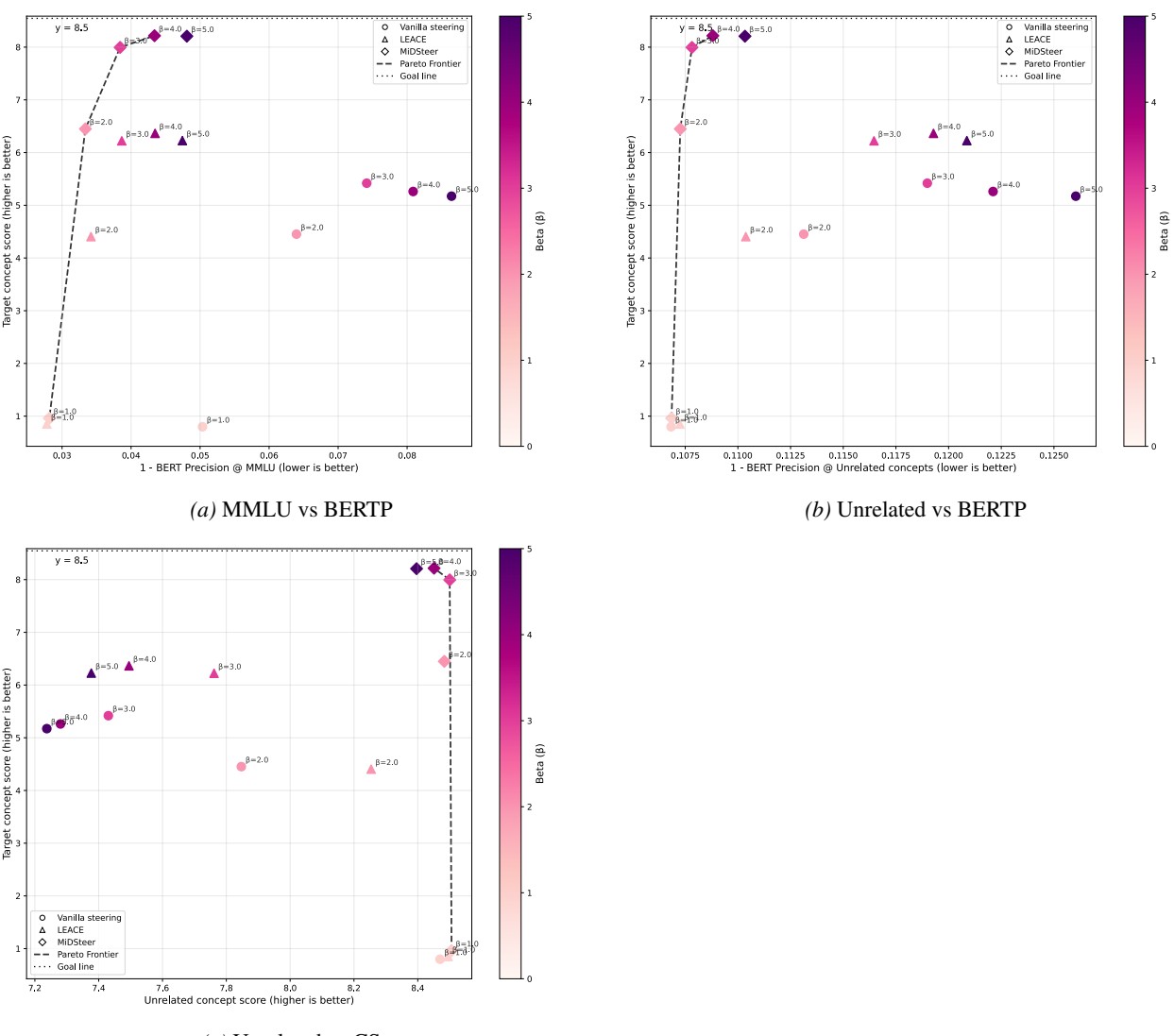

*(a)* MMLU vs BERTP

*(b)* Unrelated vs BERTP

*(c)* Unrelated vs CS

*Figure 10.* Pareto plot for concept flip on model qwen-14b (Target-CS axes)

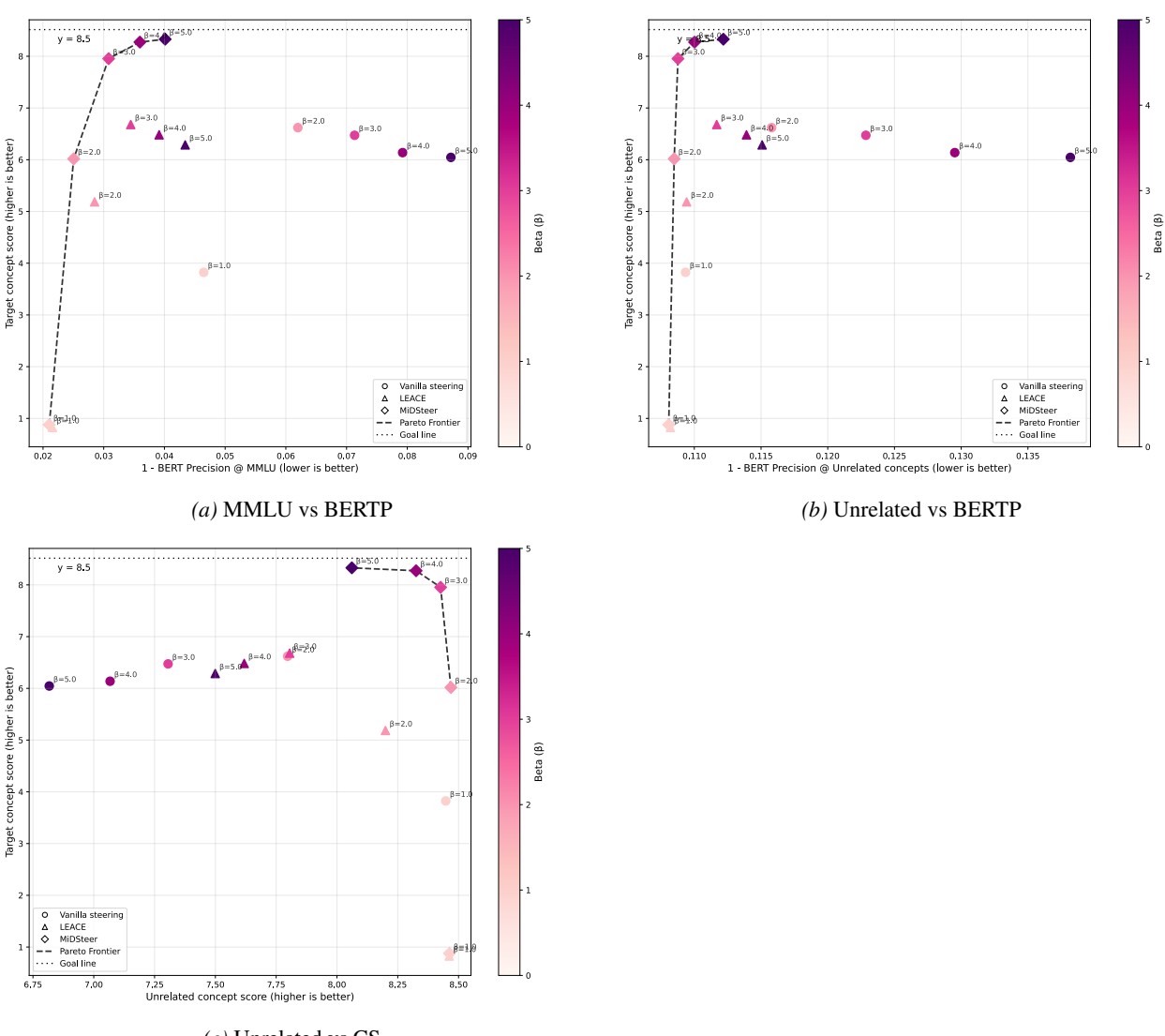

*(a)* MMLU vs BERTP

*(b)* Unrelated vs BERTP

*(c)* Unrelated vs CS

*Figure 11.* Pareto plot for concept flip on model qwen-7b (Target-CS axes)

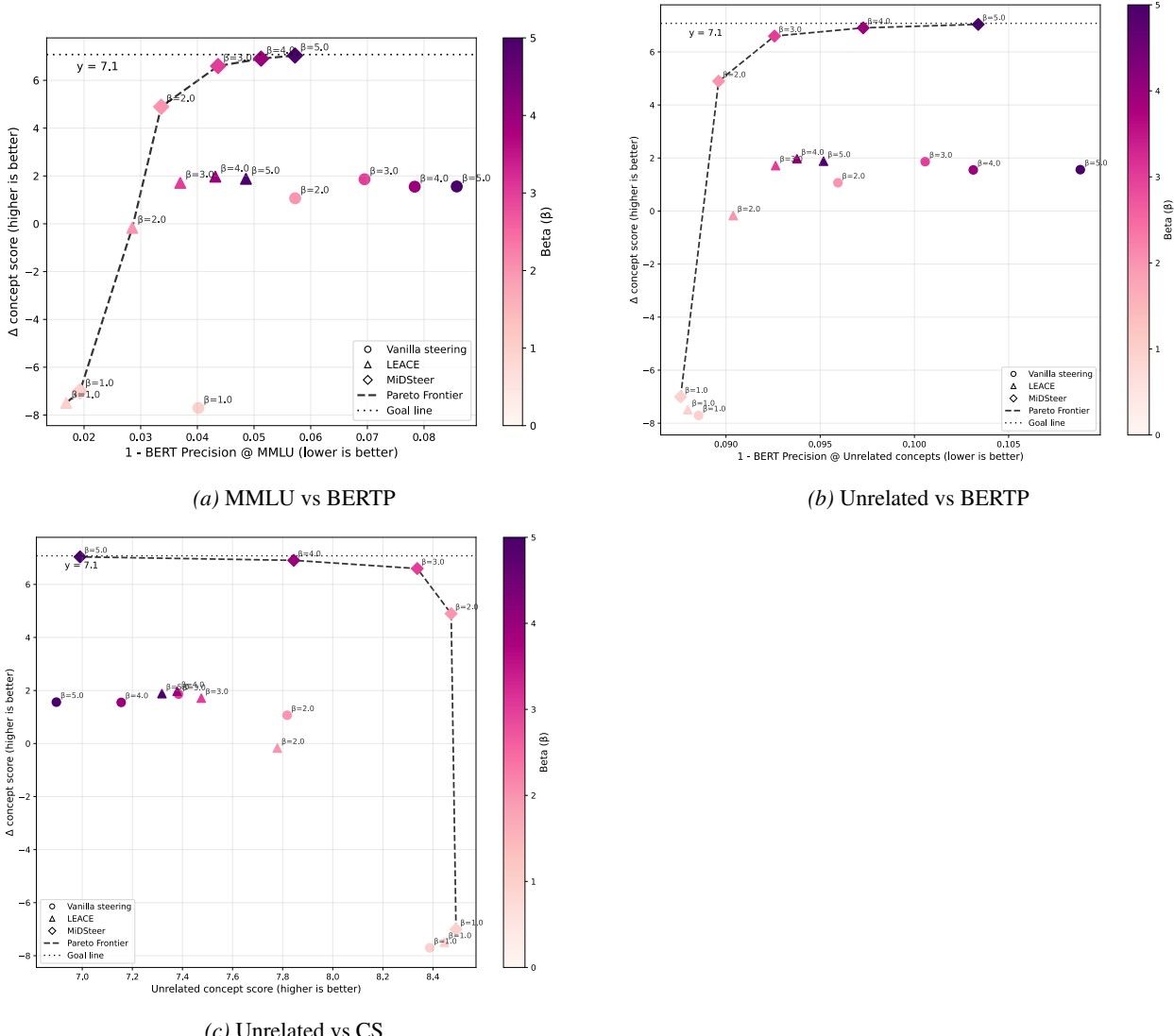

*(a)* MMLU vs BERTP

*(b)* Unrelated vs BERTP

*(c)* Unrelated vs CS

*Figure 12.* Pareto plot for concept flip on model llama2-7b (Other axes axes)

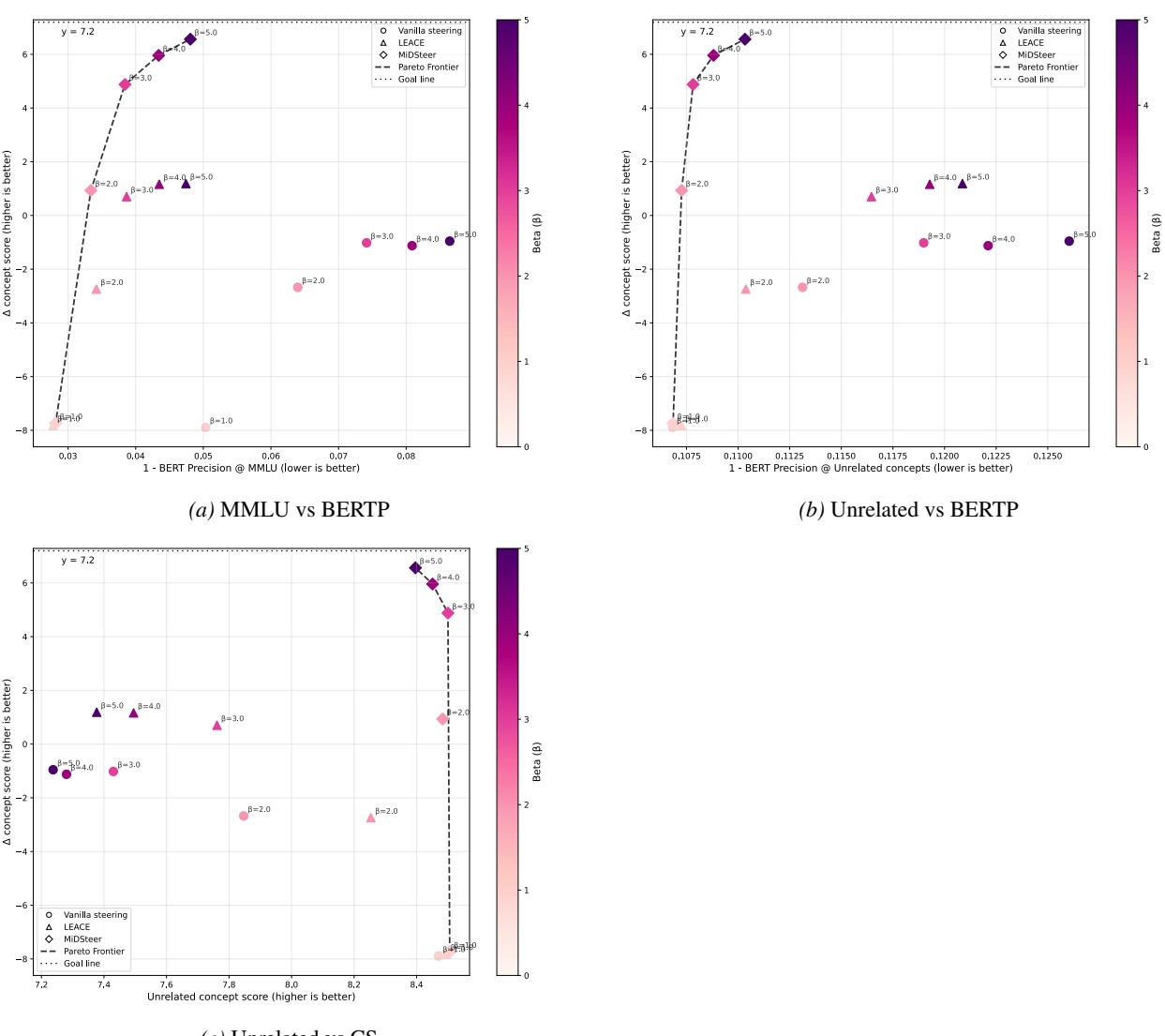

*(a)* MMLU vs BERTP

*(b)* Unrelated vs BERTP

*(c)* Unrelated vs CS

*Figure 13.* Pareto plot for concept flip on model qwen-14b (Other axes axes)

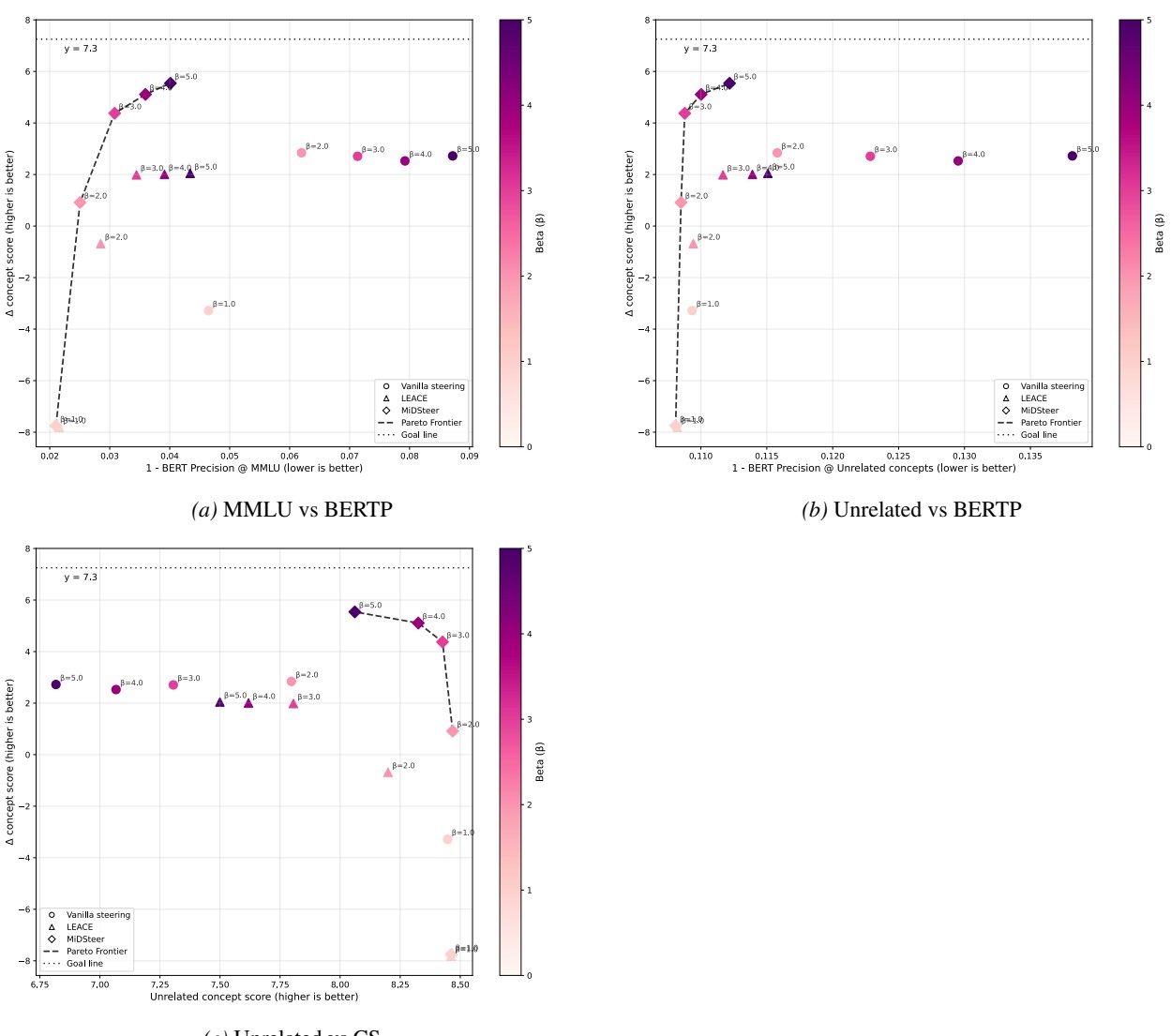

*(a)* MMLU vs BERTP

*(b)* Unrelated vs BERTP

*(c)* Unrelated vs CS

*Figure 14.* Pareto plot for concept flip on model qwen-7b (Other axes axes)

### K.1. Pareto charts for image diffusion concept switching

In this section, we provide more Pareto charts for Diffusion Models concept switching. When switching concept $c_s$ to $c_t$, for each LLM model we provide 9 types of Pareto plots:

- FID score for unrelated concepts (horizontal axis) vs $\Delta$ Concept Score (CS) for the target $c_t$ and source $c_s$ concepts (fig. 19b, 20b)

- Average Concept Score (CS) for unrelated concepts $c_i$ (horizontal axis) vs $\Delta$ Concept Score (CS) for the target $c_t$ and source $c_s$ concepts (vertical axis) (fig. 19a, 20a)

- FID score for unrelated concepts (horizontal axis) vs Concept Score (CS) for the **source** $c_s$ concept (vertical axis) (fig. 15b, 16b)

- Average Concept Score (CS) for unrelated concepts $c_i$ (horizontal axis) vs Concept Score (CS) for the **source** $c_s$ concept (vertical axis) (fig.15a, 16a)

- FID score for unrelated concepts (horizontal axis) vs Concept Score (CS) for the **target** $c_s$ concept (vertical axis) (fig. 17b, 18b)

- Average Concept Score (CS) for unrelated concepts $c_i$ (horizontal axis) vs Concept Score (CS) for the **target** $c_s$ concept (vertical axis) (fig.17a, 18a)

In each case, we see clear superiority of MidSteer over other steering approaches.

We additionally provide detailed breakdown of scores for all $\beta$ values and all concepts $c_s, c_t, c_i$ in the tables in sec. K.3

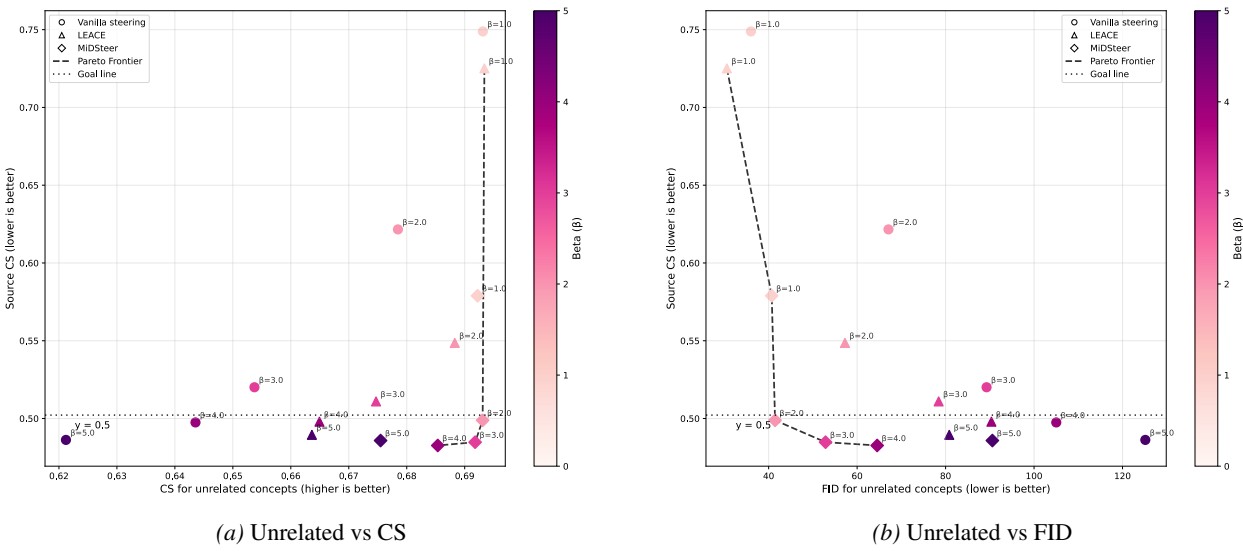

*(a)* Unrelated vs CS

*(b)* Unrelated vs FID

*Figure 15.* Pareto plot for concept flip on model SANA (Source-CS axes)

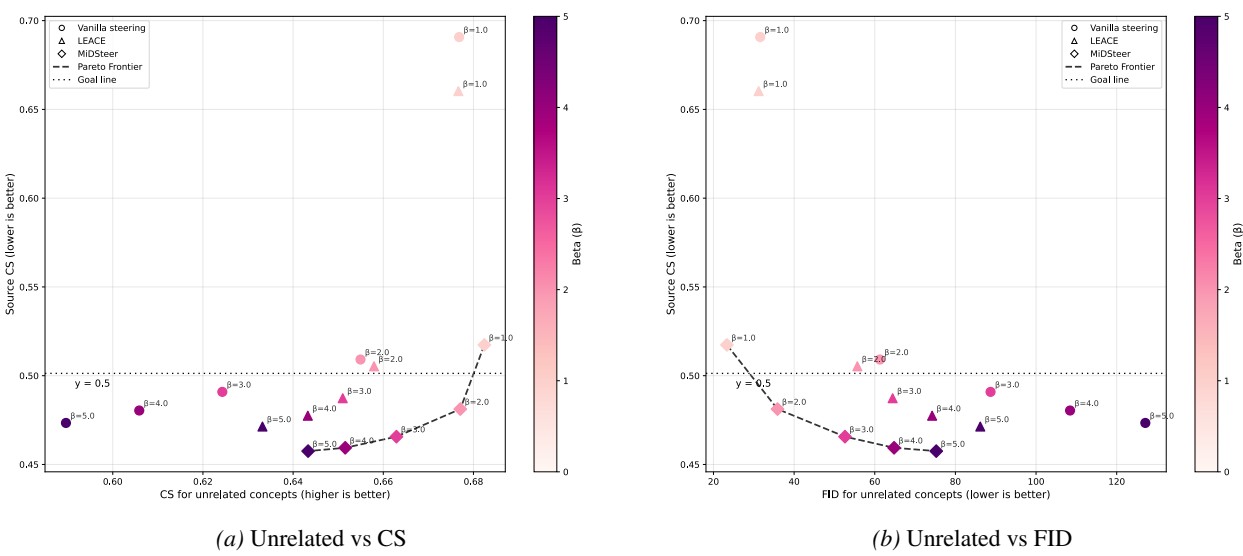

*(a)* Unrelated vs CS

*(b)* Unrelated vs FID

*Figure 16.* Pareto plot for concept flip on model SDXL (Source-CS axes)

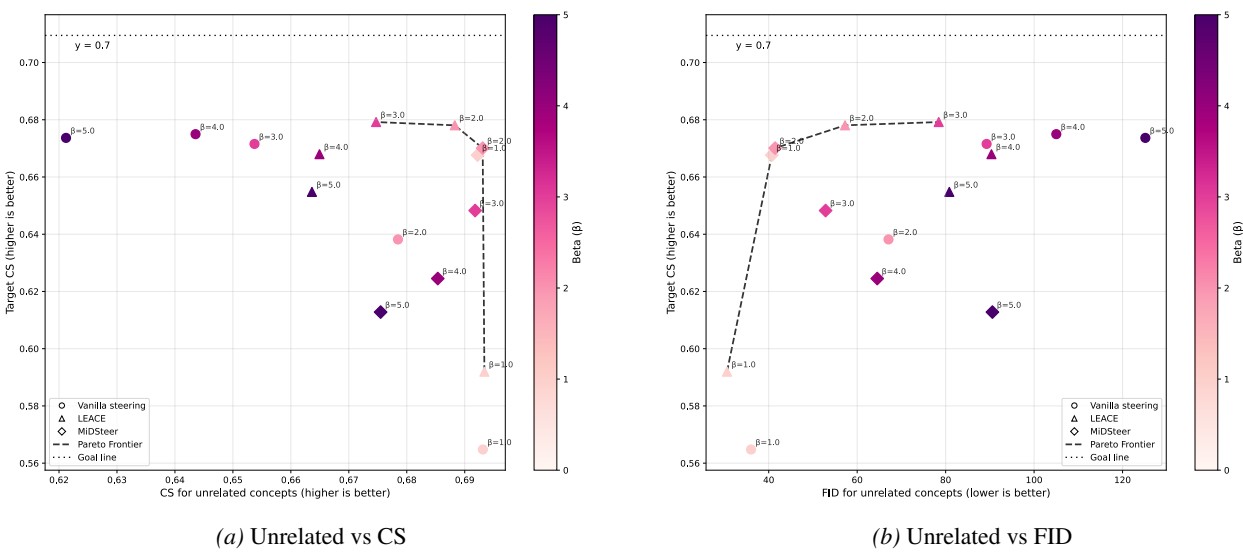

*(a)* Unrelated vs CS  *(b)* Unrelated vs FID

*Figure 17.* Pareto plot for concept flip on model SANA (Target-CS axes)

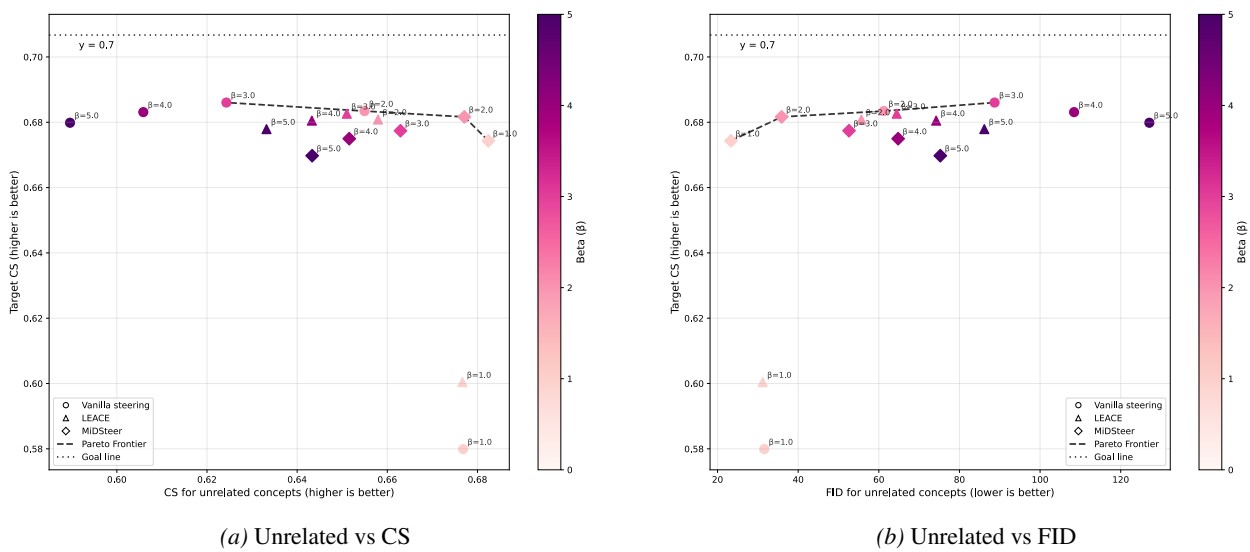

*(a)* Unrelated vs CS  *(b)* Unrelated vs FID

*Figure 18.* Pareto plot for concept flip on model SDXL (Target-CS axes)

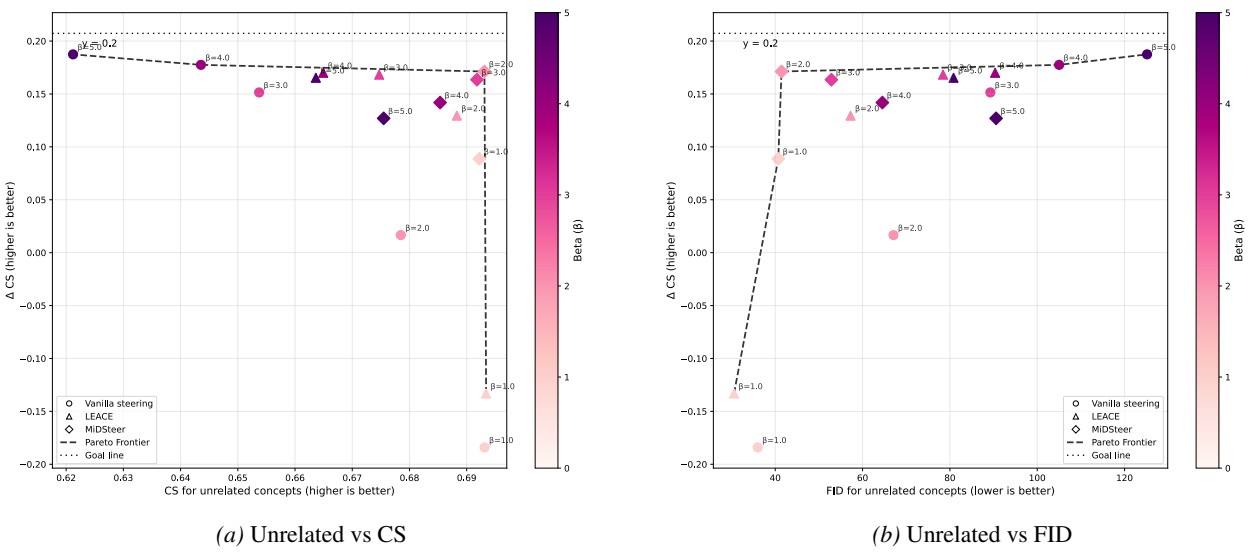

*(a)* Unrelated vs CS        *(b)* Unrelated vs FID

*Figure 19.* Pareto plot for concept flip on model SANA (Other axes axes)

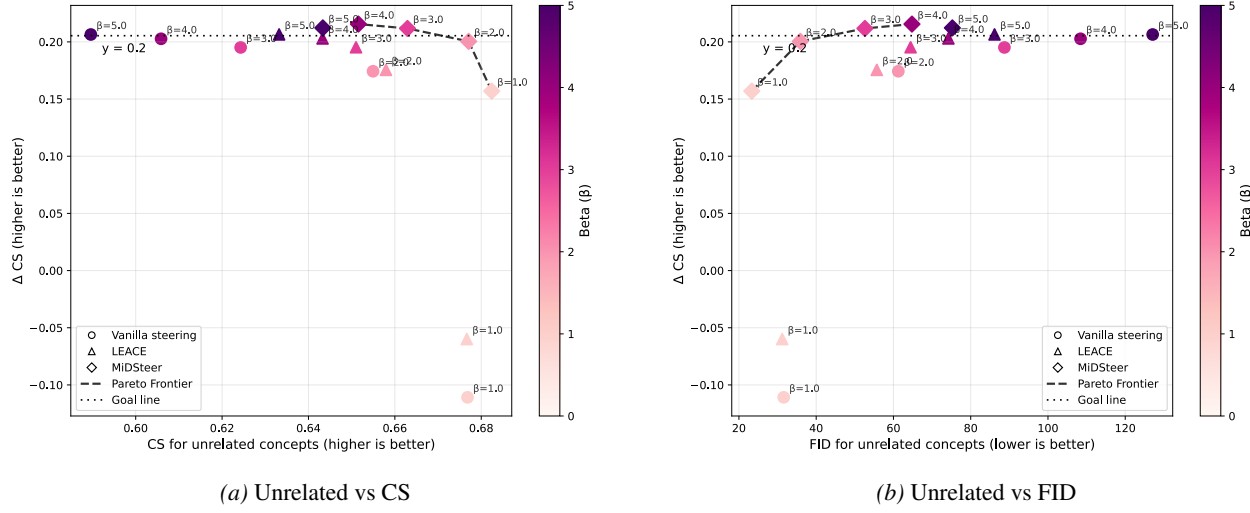

*(a)* Unrelated vs CS        *(b)* Unrelated vs FID

*Figure 20.* Pareto plot for concept flip on model SDXL (Other axes axes)

## K.2. Detailed results for LLM concept switching

In this section, in tab. 10,11,14,15,12,13 we provide detailed breakdown of scores for all $\beta$ values and all concepts $c_s, c_t, c_i$. Pareto plots in sec. K.0.1 were created based on the scores provided in these tables.

*Table 10.* Model LLama2-7b, flipping from dogs to cats

| | | dogs | | cats | | | | wolves | | | | pigs | | | | cows | | | | legislators | | |
|---|---|---|---|---|---|---|---|---|---|---|---|---|---|---|---|---|---|---|---|---|---|---|
| method | strength | src-cs↓ | tgt-cs↑ | src-cs↓ | tgt-cs↑ | bertp↑ | cs↑ | src-cs↓ | tgt-cs↓ | bertp↑ | cs↑ | src-cs↓ | tgt-cs↓ | bertp↑ | cs↑ | src-cs↓ | tgt-cs↓ | bertp↑ | cs↑ | src-cs↓ | tgt-cs↓ | bertp↑ |
| Steering | 1.0 | 8.6 | 1.1 | 2.9 | 7.5 | 0.91 | 8.5 | 7.8 | 1.5 | 0.91 | 8.5 | 1.5 | 0.4 | 0.91 | 8.4 | 0.7 | 0.3 | 0.91 | 8.5 | 0.3 | 0.2 | 0.90 |
| | 1.5 | 8.1 | 1.9 | 6.4 | 3.8 | 0.90 | 8.5 | 7.8 | 1.5 | 0.90 | 8.5 | 1.5 | 0.4 | 0.91 | 8.4 | 0.6 | 0.3 | 0.90 | 8.5 | 0.4 | 0.2 | 0.90 |
| | 2.0 | 5.9 | 3.9 | 7.1 | 2.8 | 0.89 | 8.4 | 7.8 | 1.4 | 0.90 | 8.5 | 1.5 | 0.4 | 0.90 | 8.4 | 0.7 | 0.3 | 0.90 | 8.5 | 0.3 | 0.2 | 0.90 |
| | 2.5 | 5.0 | 4.4 | 7.0 | 3.0 | 0.89 | 8.5 | 7.7 | 1.4 | 0.90 | 8.4 | 1.4 | 0.4 | 0.90 | 8.4 | 0.7 | 0.3 | 0.90 | 8.4 | 0.4 | 0.2 | 0.89 |
| | 3.0 | 4.8 | 4.6 | 6.6 | 2.8 | 0.89 | 8.5 | 7.7 | 1.4 | 0.90 | 8.4 | 1.4 | 0.4 | 0.90 | 8.4 | 0.7 | 0.3 | 0.90 | 8.5 | 0.3 | 0.2 | 0.89 |
| | 3.5 | 4.6 | 4.7 | 6.5 | 3.0 | 0.89 | 8.4 | 7.7 | 1.4 | 0.90 | 8.5 | 1.4 | 0.3 | 0.90 | 8.3 | 0.9 | 0.3 | 0.90 | 8.5 | 0.4 | 0.2 | 0.90 |
| | 4.0 | 4.6 | 4.7 | 6.5 | 2.9 | 0.89 | 8.4 | 7.7 | 1.5 | 0.90 | 8.4 | 1.4 | 0.4 | 0.90 | 8.2 | 0.9 | 0.3 | 0.90 | 8.4 | 0.4 | 0.2 | 0.89 |
| | 4.5 | 4.5 | 4.9 | 6.5 | 2.9 | 0.89 | 8.3 | 7.6 | 1.5 | 0.90 | 8.3 | 1.5 | 0.5 | 0.90 | 8.2 | 1.0 | 0.4 | 0.90 | 8.4 | 0.4 | 0.2 | 0.89 |
| | 5.0 | 4.4 | 4.9 | 6.5 | 2.7 | 0.88 | 8.3 | 7.6 | 1.5 | 0.89 | 8.2 | 1.5 | 0.6 | 0.89 | 7.9 | 1.2 | 0.5 | 0.89 | 8.2 | 0.4 | 0.2 | 0.89 |
| LEACE | 1.0 | 8.5 | 1.4 | 2.1 | 8.2 | 0.91 | 8.4 | 7.8 | 1.5 | 0.91 | 8.5 | 1.5 | 0.5 | 0.91 | 8.4 | 0.7 | 0.3 | 0.91 | 8.5 | 0.3 | 0.2 | 0.90 |
| | 1.5 | 5.4 | 4.5 | 3.4 | 6.6 | 0.91 | 8.4 | 7.8 | 1.5 | 0.91 | 8.5 | 1.4 | 0.4 | 0.91 | 8.4 | 0.7 | 0.3 | 0.91 | 8.5 | 0.3 | 0.1 | 0.90 |
| | 2.0 | 4.0 | 5.5 | 4.9 | 4.8 | 0.91 | 8.4 | 7.7 | 1.7 | 0.91 | 8.5 | 1.4 | 0.4 | 0.91 | 8.4 | 0.6 | 0.3 | 0.91 | 8.5 | 0.3 | 0.2 | 0.90 |
| | 2.5 | 3.8 | 5.6 | 5.3 | 4.1 | 0.90 | 8.4 | 7.7 | 1.5 | 0.91 | 8.5 | 1.5 | 0.5 | 0.91 | 8.4 | 0.7 | 0.3 | 0.91 | 8.5 | 0.3 | 0.2 | 0.90 |
| | 3.0 | 3.7 | 5.6 | 5.6 | 4.1 | 0.90 | 8.4 | 7.7 | 1.6 | 0.91 | 8.5 | 1.5 | 0.4 | 0.91 | 8.4 | 0.7 | 0.3 | 0.91 | 8.5 | 0.3 | 0.2 | 0.90 |
| | 3.5 | 3.7 | 5.7 | 5.5 | 3.9 | 0.90 | 8.4 | 7.7 | 1.5 | 0.91 | 8.4 | 1.5 | 0.5 | 0.91 | 8.4 | 0.7 | 0.3 | 0.91 | 8.5 | 0.3 | 0.2 | 0.90 |
| | 4.0 | 3.6 | 5.6 | 5.5 | 3.9 | 0.90 | 8.4 | 7.6 | 1.6 | 0.91 | 8.5 | 1.5 | 0.4 | 0.91 | 8.4 | 0.7 | 0.3 | 0.91 | 8.5 | 0.3 | 0.2 | 0.90 |
| | 4.5 | 3.6 | 5.6 | 5.5 | 3.9 | 0.90 | 8.4 | 7.7 | 1.6 | 0.91 | 8.5 | 1.4 | 0.4 | 0.91 | 8.4 | 0.7 | 0.3 | 0.91 | 8.5 | 0.3 | 0.2 | 0.90 |
| | 5.0 | 3.6 | 5.5 | 5.4 | 3.9 | 0.90 | 8.4 | 7.7 | 1.6 | 0.91 | 8.4 | 1.5 | 0.5 | 0.91 | 8.3 | 0.7 | 0.3 | 0.91 | 8.5 | 0.3 | 0.1 | 0.90 |
| MidSteer | 1.0 | 8.0 | 1.8 | 1.6 | 8.6 | 0.92 | 8.4 | 7.8 | 1.6 | 0.91 | 8.5 | 1.4 | 0.4 | 0.91 | 8.4 | 0.7 | 0.3 | 0.91 | 8.5 | 0.3 | 0.1 | 0.91 |
| | 1.5 | 4.1 | 6.1 | 1.5 | 8.6 | 0.91 | 8.5 | 7.8 | 1.6 | 0.91 | 8.4 | 1.5 | 0.5 | 0.91 | 8.4 | 0.6 | 0.3 | 0.91 | 8.5 | 0.4 | 0.2 | 0.90 |
| | 2.0 | 2.6 | 7.7 | 1.5 | 8.6 | 0.91 | 8.4 | 7.7 | 1.7 | 0.91 | 8.5 | 1.5 | 0.5 | 0.91 | 8.4 | 0.6 | 0.3 | 0.91 | 8.5 | 0.3 | 0.2 | 0.90 |
| | 2.5 | 2.1 | 8.2 | 1.5 | 8.6 | 0.91 | 8.3 | 7.5 | 1.8 | 0.91 | 8.4 | 1.4 | 0.5 | 0.91 | 8.2 | 0.7 | 0.4 | 0.91 | 8.4 | 0.3 | 0.2 | 0.90 |
| | 3.0 | 1.9 | 8.4 | 1.4 | 8.7 | 0.91 | 7.9 | 7.0 | 2.2 | 0.91 | 8.3 | 1.5 | 0.8 | 0.91 | 8.0 | 0.7 | 0.8 | 0.91 | 8.5 | 0.3 | 0.2 | 0.90 |
| | 3.5 | 1.7 | 8.5 | 1.3 | 8.7 | 0.91 | 7.4 | 6.5 | 3.2 | 0.90 | 7.8 | 1.6 | 1.5 | 0.90 | 7.7 | 0.9 | 1.4 | 0.91 | 8.4 | 0.3 | 0.2 | 0.90 |
| | 4.0 | 1.6 | 8.5 | 1.4 | 8.6 | 0.90 | 6.3 | 5.6 | 4.2 | 0.90 | 7.3 | 1.7 | 2.3 | 0.90 | 6.8 | 1.1 | 2.3 | 0.90 | 8.4 | 0.4 | 0.4 | 0.90 |
| | 4.5 | 1.6 | 8.6 | 1.3 | 8.7 | 0.90 | 5.4 | 4.8 | 5.1 | 0.89 | 6.5 | 1.8 | 3.2 | 0.90 | 5.8 | 1.2 | 3.8 | 0.90 | 8.3 | 0.5 | 0.9 | 0.90 |
| | 5.0 | 1.6 | 8.6 | 1.4 | 8.7 | 0.90 | 4.6 | 4.1 | 5.8 | 0.89 | 5.5 | 2.0 | 4.4 | 0.89 | 4.6 | 1.2 | 4.8 | 0.89 | 8.1 | 0.6 | 1.4 | 0.90 |

*Table 11.* Model LLama2-7b, flipping from horses to motorcycles

| | | horses | | motorcycles | | | | cows | | | | pigs | | | | dogs | | | | legislators | | |
|---|---|---|---|---|---|---|---|---|---|---|---|---|---|---|---|---|---|---|---|---|---|---|
| method | strength | src-cs↓ | tgt-cs↑ | src-cs↓ | tgt-cs↑ | bertp↑ | cs↑ | src-cs↓ | tgt-cs↓ | bertp↑ | cs↑ | src-cs↓ | tgt-cs↓ | bertp↑ | cs↑ | src-cs↓ | tgt-cs↓ | bertp↑ | cs↑ | src-cs↓ | tgt-cs↓ | bertp↑ |
| Steering | 1.0 | 8.6 | 0.7 | 1.3 | 8.5 | 0.92 | 8.4 | 1.7 | 0.1 | 0.91 | 8.5 | 1.2 | 0.1 | 0.91 | 8.6 | 1.5 | 0.2 | 0.92 | 8.5 | 0.3 | 0.3 | 0.90 |
| | 1.5 | 6.5 | 2.9 | 1.4 | 8.5 | 0.92 | 8.4 | 1.7 | 0.1 | 0.91 | 8.4 | 1.2 | 0.1 | 0.91 | 8.6 | 1.5 | 0.2 | 0.91 | 8.5 | 0.3 | 0.3 | 0.90 |
| | 2.0 | 2.7 | 6.9 | 2.1 | 7.9 | 0.91 | 8.3 | 1.6 | 0.1 | 0.91 | 8.4 | 1.2 | 0.1 | 0.91 | 8.6 | 1.5 | 0.2 | 0.91 | 8.5 | 0.3 | 0.3 | 0.90 |
| | 2.5 | 2.6 | 7.2 | 3.7 | 5.5 | 0.91 | 8.1 | 1.4 | 0.2 | 0.90 | 8.3 | 1.2 | 0.1 | 0.91 | 8.5 | 1.5 | 0.2 | 0.91 | 8.5 | 0.3 | 0.3 | 0.90 |
| | 3.0 | 2.9 | 6.8 | 4.4 | 4.2 | 0.90 | 7.8 | 1.3 | 0.3 | 0.90 | 8.2 | 1.2 | 0.1 | 0.90 | 8.6 | 1.4 | 0.2 | 0.91 | 8.5 | 0.3 | 0.2 | 0.90 |
| | 3.5 | 3.0 | 6.5 | 4.7 | 3.6 | 0.90 | 7.2 | 1.3 | 0.3 | 0.90 | 8.1 | 1.1 | 0.1 | 0.90 | 8.5 | 1.3 | 0.2 | 0.91 | 8.4 | 0.3 | 0.3 | 0.90 |
| | 4.0 | 3.0 | 6.1 | 4.6 | 3.4 | 0.89 | 6.9 | 1.1 | 0.4 | 0.90 | 8.1 | 1.1 | 0.2 | 0.90 | 8.3 | 1.4 | 0.3 | 0.90 | 8.5 | 0.3 | 0.3 | 0.90 |
| | 4.5 | 3.1 | 5.8 | 5.0 | 3.2 | 0.89 | 6.5 | 1.1 | 0.4 | 0.89 | 7.9 | 1.1 | 0.2 | 0.90 | 8.2 | 1.4 | 0.3 | 0.90 | 8.4 | 0.4 | 0.4 | 0.90 |
| | 5.0 | 3.0 | 5.7 | 4.8 | 3.1 | 0.89 | 6.4 | 1.1 | 0.5 | 0.89 | 7.7 | 1.0 | 0.3 | 0.89 | 8.0 | 1.3 | 0.3 | 0.90 | 8.4 | 0.4 | 0.4 | 0.90 |
| LEACE | 1.0 | 8.5 | 0.7 | 1.2 | 8.5 | 0.92 | 8.4 | 1.7 | 0.1 | 0.91 | 8.4 | 1.2 | 0.1 | 0.91 | 8.6 | 1.5 | 0.1 | 0.92 | 8.5 | 0.3 | 0.3 | 0.90 |
| | 1.5 | 8.2 | 1.0 | 1.7 | 8.2 | 0.92 | 8.4 | 1.6 | 0.2 | 0.91 | 8.4 | 1.3 | 0.1 | 0.91 | 8.6 | 1.5 | 0.2 | 0.92 | 8.5 | 0.4 | 0.3 | 0.90 |
| | 2.0 | 5.4 | 3.5 | 4.0 | 5.4 | 0.91 | 8.3 | 1.7 | 0.2 | 0.91 | 8.4 | 1.3 | 0.1 | 0.91 | 8.6 | 1.5 | 0.2 | 0.92 | 8.5 | 0.3 | 0.2 | 0.90 |
| | 2.5 | 4.0 | 4.9 | 4.9 | 3.9 | 0.90 | 8.3 | 1.6 | 0.2 | 0.91 | 8.4 | 1.2 | 0.1 | 0.91 | 8.6 | 1.5 | 0.2 | 0.91 | 8.5 | 0.3 | 0.2 | 0.91 |
| | 3.0 | 3.7 | 5.2 | 5.5 | 3.2 | 0.90 | 8.2 | 1.5 | 0.2 | 0.91 | 8.4 | 1.2 | 0.1 | 0.91 | 8.6 | 1.5 | 0.2 | 0.91 | 8.5 | 0.3 | 0.3 | 0.90 |
| | 3.5 | 3.6 | 5.3 | 5.4 | 3.0 | 0.90 | 8.1 | 1.4 | 0.2 | 0.91 | 8.3 | 1.2 | 0.1 | 0.91 | 8.6 | 1.4 | 0.2 | 0.91 | 8.5 | 0.3 | 0.2 | 0.90 |
| | 4.0 | 3.5 | 5.5 | 5.5 | 2.9 | 0.90 | 8.0 | 1.3 | 0.2 | 0.90 | 8.3 | 1.1 | 0.1 | 0.91 | 8.6 | 1.5 | 0.2 | 0.91 | 8.5 | 0.4 | 0.2 | 0.90 |
| | 4.5 | 3.4 | 5.2 | 5.5 | 2.8 | 0.90 | 7.7 | 1.2 | 0.2 | 0.90 | 8.2 | 1.2 | 0.1 | 0.91 | 8.5 | 1.5 | 0.2 | 0.91 | 8.5 | 0.3 | 0.2 | 0.90 |
| | 5.0 | 3.4 | 5.3 | 5.6 | 2.9 | 0.90 | 7.6 | 1.3 | 0.3 | 0.90 | 8.2 | 1.1 | 0.1 | 0.91 | 8.5 | 1.5 | 0.2 | 0.91 | 8.5 | 0.4 | 0.3 | 0.90 |
| MidSteer | 1.0 | 8.6 | 0.7 | 1.2 | 8.5 | 0.92 | 8.4 | 1.6 | 0.1 | 0.91 | 8.5 | 1.2 | 0.1 | 0.91 | 8.6 | 1.5 | 0.2 | 0.92 | 8.5 | 0.3 | 0.3 | 0.90 |
| | 1.5 | 5.9 | 3.5 | 1.2 | 8.5 | 0.92 | 8.4 | 1.6 | 0.1 | 0.91 | 8.4 | 1.2 | 0.1 | 0.91 | 8.6 | 1.6 | 0.2 | 0.92 | 8.4 | 0.3 | 0.3 | 0.90 |
| | 2.0 | 2.6 | 7.2 | 1.2 | 8.5 | 0.92 | 8.3 | 1.5 | 0.1 | 0.91 | 8.4 | 1.2 | 0.1 | 0.91 | 8.6 | 1.5 | 0.2 | 0.91 | 8.4 | 0.3 | 0.3 | 0.90 |
| | 2.5 | 1.8 | 8.2 | 1.2 | 8.5 | 0.92 | 8.3 | 1.6 | 0.2 | 0.91 | 8.4 | 1.2 | 0.1 | 0.91 | 8.6 | 1.5 | 0.2 | 0.91 | 8.5 | 0.3 | 0.3 | 0.90 |
| | 3.0 | 1.7 | 8.4 | 1.2 | 8.5 | 0.92 | 8.3 | 1.6 | 0.2 | 0.91 | 8.2 | 1.1 | 0.1 | 0.91 | 8.6 | 1.6 | 0.2 | 0.91 | 8.5 | 0.3 | 0.3 | 0.90 |
| | 3.5 | 1.6 | 8.4 | 1.2 | 8.5 | 0.92 | 8.0 | 1.4 | 0.4 | 0.90 | 8.2 | 1.0 | 0.2 | 0.90 | 8.5 | 1.5 | 0.2 | 0.91 | 8.4 | 0.4 | 0.3 | 0.90 |
| | 4.0 | 1.6 | 8.4 | 1.2 | 8.5 | 0.91 | 7.6 | 1.3 | 0.7 | 0.90 | 8.0 | 1.1 | 0.4 | 0.90 | 8.4 | 1.4 | 0.2 | 0.91 | 8.4 | 0.3 | 0.3 | 0.90 |
| | 4.5 | 1.6 | 8.5 | 1.2 | 8.5 | 0.91 | 7.0 | 1.3 | 1.4 | 0.90 | 7.6 | 1.0 | 0.6 | 0.90 | 8.3 | 1.4 | 0.4 | 0.91 | 8.5 | 0.3 | 0.3 | 0.90 |
| | 5.0 | 1.4 | 8.5 | 1.2 | 8.5 | 0.91 | 6.1 | 1.2 | 2.2 | 0.89 | 7.3 | 0.9 | 0.9 | 0.90 | 8.2 | 1.4 | 0.5 | 0.90 | 8.4 | 0.3 | 0.3 | 0.90 |

*Table 12.* Model Qwen2.5-7b, flipping from dogs to cats

| method | strength | dogs src-cs↓ | dogs tgt-cs↑ | cats src-cs↓ | cats tgt-cs↑ | cats bertp↑ | cs↑ | wolves src-cs↓ | wolves tgt-cs↓ | wolves bertp↑ | cs↑ | pigs src-cs↓ | pigs tgt-cs↓ | pigs bertp↑ | cs↑ | cows src-cs↓ | cows tgt-cs↓ | cows bertp↑ | cs↑ | legislators src-cs↓ | legislators tgt-cs↓ | legislators bertp↑ |
|---|---|---|---|---|---|---|---|---|---|---|---|---|---|---|---|---|---|---|---|---|---|---|
| Steering | 1.0 | 5.6 | 6.9 | 1.9 | 8.4 | 0.89 | 8.3 | 7.6 | 2.0 | 0.89 | 8.4 | 1.6 | 0.6 | 0.89 | 8.4 | 0.8 | 0.4 | 0.89 | 8.5 | 0.3 | 0.2 | 0.89 |
| | 2.0 | 3.2 | 7.5 | 2.2 | 7.8 | 0.88 | 7.6 | 6.4 | 3.7 | 0.88 | 8.1 | 1.6 | 1.9 | 0.88 | 7.9 | 1.1 | 1.8 | 0.89 | 8.4 | 0.5 | 0.9 | 0.89 |
| | 3.0 | 3.4 | 7.5 | 2.3 | 7.6 | 0.87 | 7.3 | 6.2 | 3.9 | 0.87 | 7.2 | 1.7 | 3.0 | 0.88 | 7.0 | 1.4 | 3.0 | 0.88 | 8.4 | 0.7 | 2.0 | 0.88 |
| | 4.0 | 3.0 | 7.1 | 2.4 | 7.2 | 0.86 | 7.2 | 6.2 | 3.7 | 0.87 | 7.4 | 1.6 | 2.2 | 0.87 | 7.1 | 1.3 | 2.8 | 0.87 | 8.3 | 0.6 | 1.6 | 0.88 |
| | 5.0 | 2.9 | 6.8 | 2.4 | 6.6 | 0.85 | 5.9 | 5.2 | 4.3 | 0.85 | 7.3 | 1.6 | 1.7 | 0.86 | 7.5 | 1.2 | 1.8 | 0.86 | 8.3 | 0.5 | 1.0 | 0.87 |
| LEACE | 1.0 | 8.7 | 1.0 | 1.8 | 8.5 | 0.89 | 8.4 | 7.7 | 1.6 | 0.89 | 8.4 | 1.6 | 0.4 | 0.89 | 8.4 | 0.9 | 0.3 | 0.89 | 8.5 | 0.3 | 0.2 | 0.89 |
| | 2.0 | 7.4 | 4.2 | 6.0 | 6.7 | 0.88 | 8.3 | 7.6 | 1.6 | 0.89 | 8.4 | 1.4 | 0.4 | 0.89 | 8.4 | 0.9 | 0.3 | 0.89 | 8.5 | 0.3 | 0.2 | 0.89 |
| | 3.0 | 6.3 | 6.7 | 7.1 | 5.5 | 0.88 | 8.3 | 7.6 | 1.7 | 0.89 | 8.4 | 1.6 | 0.4 | 0.89 | 8.4 | 0.9 | 0.3 | 0.89 | 8.5 | 0.3 | 0.2 | 0.89 |
| | 4.0 | 5.8 | 6.6 | 6.9 | 4.7 | 0.87 | 8.3 | 7.5 | 1.7 | 0.89 | 8.4 | 1.5 | 0.4 | 0.89 | 8.4 | 0.8 | 0.3 | 0.89 | 8.5 | 0.3 | 0.2 | 0.89 |
| | 5.0 | 5.3 | 6.5 | 6.7 | 4.3 | 0.87 | 8.3 | 7.6 | 1.8 | 0.89 | 8.4 | 1.5 | 0.4 | 0.89 | 8.4 | 0.8 | 0.3 | 0.89 | 8.5 | 0.4 | 0.2 | 0.89 |
| MidSteer | 1.0 | 8.7 | 1.0 | 1.7 | 8.5 | 0.89 | 8.3 | 7.6 | 1.6 | 0.89 | 8.5 | 1.5 | 0.5 | 0.89 | 8.4 | 0.8 | 0.3 | 0.89 | 8.5 | 0.4 | 0.2 | 0.89 |
| | 2.0 | 7.3 | 5.0 | 1.7 | 8.5 | 0.89 | 8.4 | 7.6 | 1.7 | 0.89 | 8.5 | 1.5 | 0.4 | 0.89 | 8.4 | 0.8 | 0.3 | 0.89 | 8.5 | 0.3 | 0.2 | 0.89 |
| | 3.0 | 5.7 | 7.7 | 1.7 | 8.5 | 0.89 | 8.3 | 7.5 | 2.0 | 0.89 | 8.4 | 1.4 | 0.6 | 0.89 | 8.4 | 0.8 | 0.5 | 0.89 | 8.5 | 0.3 | 0.2 | 0.89 |
| | 4.0 | 5.0 | 8.2 | 1.7 | 8.5 | 0.89 | 8.1 | 7.4 | 2.5 | 0.89 | 8.3 | 1.6 | 1.0 | 0.89 | 8.3 | 1.0 | 0.7 | 0.89 | 8.5 | 0.4 | 0.3 | 0.89 |
| | 5.0 | 4.3 | 8.3 | 1.7 | 8.5 | 0.89 | 7.8 | 6.9 | 3.2 | 0.89 | 8.3 | 1.7 | 1.1 | 0.89 | 8.2 | 1.1 | 1.0 | 0.89 | 8.4 | 0.4 | 0.3 | 0.89 |

*Table 13.* Model Qwen2.5-7b, flipping from horses to motorcycles

| method | strength | horses src-cs↓ | horses tgt-cs↑ | motorcycles src-cs↓ | motorcycles tgt-cs↑ | motorcycles bertp↑ | cs↑ | cows src-cs↓ | cows tgt-cs↓ | cows bertp↑ | cs↑ | pigs src-cs↓ | pigs tgt-cs↓ | pigs bertp↑ | cs↑ | dogs src-cs↓ | dogs tgt-cs↓ | dogs bertp↑ | cs↑ | legislators src-cs↓ | legislators tgt-cs↓ | legislators bertp↑ |
|---|---|---|---|---|---|---|---|---|---|---|---|---|---|---|---|---|---|---|---|---|---|---|
| Steering | 1.0 | 8.6 | 0.8 | 1.2 | 8.4 | 0.90 | 8.4 | 1.8 | 0.2 | 0.89 | 8.4 | 1.1 | 0.1 | 0.89 | 8.6 | 1.4 | 0.3 | 0.89 | 8.5 | 0.4 | 0.4 | 0.89 |
| | 2.0 | 4.3 | 5.7 | 5.7 | 4.4 | 0.87 | 8.2 | 1.6 | 0.3 | 0.89 | 8.3 | 0.9 | 0.1 | 0.88 | 8.6 | 1.3 | 0.2 | 0.89 | 8.5 | 0.4 | 0.3 | 0.89 |
| | 3.0 | 4.1 | 5.5 | 6.1 | 3.2 | 0.86 | 7.4 | 1.3 | 0.8 | 0.88 | 8.0 | 0.8 | 0.3 | 0.87 | 8.5 | 1.3 | 0.3 | 0.88 | 8.5 | 0.3 | 0.3 | 0.89 |
| | 4.0 | 4.2 | 5.2 | 6.2 | 2.8 | 0.86 | 6.6 | 1.5 | 1.3 | 0.86 | 7.5 | 0.9 | 0.4 | 0.87 | 8.1 | 1.3 | 0.4 | 0.88 | 8.4 | 0.3 | 0.3 | 0.89 |
| | 5.0 | 3.8 | 5.3 | 6.4 | 2.9 | 0.85 | 6.3 | 1.4 | 1.2 | 0.86 | 7.1 | 0.9 | 0.6 | 0.86 | 7.8 | 1.3 | 0.5 | 0.87 | 8.4 | 0.3 | 0.3 | 0.88 |
| LEACE | 1.0 | 8.6 | 0.7 | 1.0 | 8.4 | 0.90 | 8.4 | 1.9 | 0.2 | 0.89 | 8.4 | 1.2 | 0.1 | 0.89 | 8.7 | 1.5 | 0.2 | 0.89 | 8.5 | 0.4 | 0.3 | 0.89 |
| | 2.0 | 4.4 | 6.2 | 2.8 | 7.7 | 0.89 | 8.4 | 1.8 | 0.2 | 0.89 | 8.4 | 1.1 | 0.1 | 0.89 | 8.7 | 1.5 | 0.2 | 0.89 | 8.6 | 0.4 | 0.3 | 0.89 |
| | 3.0 | 3.1 | 6.7 | 5.3 | 5.0 | 0.88 | 8.3 | 1.7 | 0.2 | 0.89 | 8.4 | 1.0 | 0.1 | 0.89 | 8.7 | 1.4 | 0.2 | 0.89 | 8.5 | 0.4 | 0.3 | 0.89 |
| | 4.0 | 3.2 | 6.3 | 5.5 | 4.1 | 0.87 | 8.2 | 1.6 | 0.4 | 0.89 | 8.3 | 1.0 | 0.1 | 0.89 | 8.6 | 1.4 | 0.2 | 0.89 | 8.5 | 0.4 | 0.4 | 0.89 |
| | 5.0 | 3.2 | 6.1 | 5.6 | 3.8 | 0.87 | 7.9 | 1.5 | 0.7 | 0.88 | 8.3 | 0.9 | 0.2 | 0.88 | 8.6 | 1.3 | 0.2 | 0.89 | 8.5 | 0.4 | 0.4 | 0.89 |
| MidSteer | 1.0 | 8.6 | 0.8 | 0.9 | 8.5 | 0.90 | 8.3 | 1.9 | 0.2 | 0.89 | 8.4 | 1.1 | 0.1 | 0.89 | 8.7 | 1.5 | 0.2 | 0.89 | 8.5 | 0.3 | 0.3 | 0.89 |
| | 2.0 | 2.9 | 7.1 | 0.8 | 8.5 | 0.90 | 8.4 | 1.8 | 0.2 | 0.89 | 8.4 | 1.1 | 0.1 | 0.89 | 8.7 | 1.4 | 0.2 | 0.89 | 8.5 | 0.3 | 0.3 | 0.89 |
| | 3.0 | 1.4 | 8.2 | 0.9 | 8.5 | 0.90 | 8.2 | 1.7 | 0.3 | 0.89 | 8.4 | 1.0 | 0.2 | 0.89 | 8.6 | 1.4 | 0.3 | 0.89 | 8.5 | 0.3 | 0.3 | 0.89 |
| | 4.0 | 1.3 | 8.4 | 0.8 | 8.5 | 0.89 | 7.7 | 1.7 | 1.0 | 0.89 | 8.2 | 0.9 | 0.3 | 0.88 | 8.6 | 1.2 | 0.3 | 0.89 | 8.4 | 0.3 | 0.4 | 0.89 |
| | 5.0 | 1.2 | 8.4 | 0.9 | 8.6 | 0.90 | 6.3 | 1.5 | 2.8 | 0.88 | 7.8 | 1.0 | 1.0 | 0.88 | 8.2 | 1.4 | 0.8 | 0.89 | 8.5 | 0.3 | 0.4 | 0.89 |

*Table 14.* Model Qwen2.5-14b, flipping from dogs to cats

| method | strength | dogs src-cs↓ | dogs tgt-cs↑ | cats src-cs↓ | cats tgt-cs↑ | cats bertp↑ | cs↑ | wolves src-cs↓ | wolves tgt-cs↓ | wolves bertp↑ | cs↑ | pigs src-cs↓ | pigs tgt-cs↓ | pigs bertp↑ | cs↑ | cows src-cs↓ | cows tgt-cs↓ | cows bertp↑ | cs↑ | legislators src-cs↓ | legislators tgt-cs↓ | legislators bertp↑ |
|---|---|---|---|---|---|---|---|---|---|---|---|---|---|---|---|---|---|---|---|---|---|---|
| Steering | 1.0 | 8.7 | 0.9 | 2.7 | 8.2 | 0.89 | 8.4 | 7.7 | 1.3 | 0.89 | 8.5 | 1.4 | 0.4 | 0.89 | 8.4 | 0.8 | 0.3 | 0.90 | 8.5 | 0.3 | 0.2 | 0.89 |
| | 2.0 | 7.8 | 4.5 | 7.3 | 4.9 | 0.86 | 8.4 | 7.7 | 1.4 | 0.89 | 8.5 | 1.4 | 0.4 | 0.89 | 8.4 | 0.8 | 0.2 | 0.89 | 8.5 | 0.3 | 0.2 | 0.89 |
| | 3.0 | 7.0 | 5.9 | 7.3 | 4.3 | 0.86 | 8.4 | 7.6 | 1.3 | 0.89 | 8.5 | 1.3 | 0.3 | 0.89 | 8.4 | 0.8 | 0.3 | 0.89 | 8.5 | 0.4 | 0.2 | 0.89 |
| | 4.0 | 6.8 | 6.2 | 7.2 | 4.6 | 0.86 | 8.4 | 7.6 | 1.3 | 0.89 | 8.5 | 1.3 | 0.4 | 0.89 | 8.4 | 0.8 | 0.3 | 0.89 | 8.5 | 0.4 | 0.2 | 0.89 |
| | 5.0 | 6.8 | 6.1 | 7.3 | 5.1 | 0.86 | 8.3 | 7.5 | 1.3 | 0.89 | 8.5 | 1.3 | 0.4 | 0.89 | 8.3 | 0.8 | 0.2 | 0.89 | 8.5 | 0.4 | 0.2 | 0.89 |
| LEACE | 1.0 | 8.7 | 1.0 | 1.9 | 8.4 | 0.89 | 8.4 | 7.7 | 1.3 | 0.89 | 8.5 | 1.5 | 0.4 | 0.89 | 8.4 | 0.8 | 0.3 | 0.89 | 8.5 | 0.3 | 0.2 | 0.89 |
| | 2.0 | 7.9 | 4.2 | 6.0 | 7.3 | 0.88 | 8.4 | 7.7 | 1.3 | 0.89 | 8.5 | 1.4 | 0.4 | 0.89 | 8.4 | 0.8 | 0.3 | 0.89 | 8.5 | 0.3 | 0.2 | 0.89 |
| | 3.0 | 6.9 | 6.5 | 7.0 | 5.4 | 0.86 | 8.4 | 7.6 | 1.4 | 0.89 | 8.5 | 1.3 | 0.3 | 0.89 | 8.4 | 0.8 | 0.3 | 0.90 | 8.5 | 0.3 | 0.1 | 0.89 |
| | 4.0 | 6.2 | 6.7 | 6.9 | 4.2 | 0.86 | 8.4 | 7.7 | 1.4 | 0.89 | 8.5 | 1.4 | 0.4 | 0.89 | 8.4 | 0.8 | 0.3 | 0.90 | 8.5 | 0.3 | 0.2 | 0.89 |
| | 5.0 | 6.0 | 6.6 | 6.9 | 3.7 | 0.85 | 8.4 | 7.7 | 1.4 | 0.89 | 8.4 | 1.4 | 0.4 | 0.89 | 8.4 | 0.8 | 0.2 | 0.89 | 8.5 | 0.3 | 0.2 | 0.89 |
| MidSteer | 1.0 | 8.7 | 1.2 | 1.8 | 8.5 | 0.89 | 8.4 | 7.7 | 1.4 | 0.89 | 8.5 | 1.5 | 0.4 | 0.89 | 8.5 | 0.8 | 0.3 | 0.89 | 8.5 | 0.3 | 0.1 | 0.89 |
| | 2.0 | 7.1 | 6.4 | 1.6 | 8.5 | 0.89 | 8.4 | 7.7 | 1.5 | 0.89 | 8.5 | 1.4 | 0.4 | 0.89 | 8.4 | 0.8 | 0.3 | 0.89 | 8.5 | 0.4 | 0.2 | 0.89 |
| | 3.0 | 4.8 | 7.8 | 1.6 | 8.6 | 0.89 | 8.4 | 7.6 | 1.8 | 0.89 | 8.4 | 1.5 | 0.5 | 0.89 | 8.4 | 0.7 | 0.3 | 0.89 | 8.5 | 0.3 | 0.2 | 0.89 |
| | 4.0 | 3.3 | 8.2 | 1.6 | 8.5 | 0.89 | 8.3 | 7.5 | 2.3 | 0.89 | 8.4 | 1.6 | 0.8 | 0.89 | 8.4 | 0.9 | 0.5 | 0.89 | 8.5 | 0.4 | 0.2 | 0.89 |
| | 5.0 | 2.1 | 8.3 | 1.6 | 8.6 | 0.89 | 8.1 | 7.0 | 2.9 | 0.89 | 8.3 | 1.8 | 1.2 | 0.89 | 8.3 | 0.9 | 0.7 | 0.89 | 8.5 | 0.4 | 0.3 | 0.89 |

*Table 15.* Model Qwen2.5-14b, flipping from horses to motorcycles

| method | strength | horses src-cs↓ | horses tgt-cs↑ | motorcycles src-cs↓ | motorcycles tgt-cs↑ | motorcycles bertp↑ | cs↑ | cows src-cs↓ | cows tgt-cs↓ | cows bertp↑ | cs↑ | pigs src-cs↓ | pigs tgt-cs↓ | pigs bertp↑ | cs↑ | dogs src-cs↓ | dogs tgt-cs↓ | dogs bertp↑ | cs↑ | legislators src-cs↓ | legislators tgt-cs↓ | legislators bertp↑ |
|---|---|---|---|---|---|---|---|---|---|---|---|---|---|---|---|---|---|---|---|---|---|---|
| Steering | 1.0 | 8.7 | 0.7 | 1.2 | 8.6 | 0.90 | 8.4 | 1.7 | 0.2 | 0.89 | 8.4 | 1.1 | 0.1 | 0.89 | 8.7 | 1.4 | 0.2 | 0.90 | 8.5 | 0.3 | 0.3 | 0.89 |
| | 2.0 | 6.4 | 4.4 | 4.9 | 6.0 | 0.87 | 8.3 | 1.5 | 0.3 | 0.89 | 8.4 | 1.0 | 0.1 | 0.89 | 8.6 | 1.4 | 0.2 | 0.89 | 8.4 | 0.3 | 0.3 | 0.89 |
| | 3.0 | 5.9 | 5.0 | 6.9 | 2.9 | 0.84 | 7.9 | 1.4 | 0.5 | 0.88 | 8.4 | 0.9 | 0.2 | 0.88 | 8.6 | 1.3 | 0.2 | 0.89 | 8.5 | 0.3 | 0.3 | 0.89 |
| | 4.0 | 6.0 | 4.4 | 7.3 | 1.8 | 0.84 | 7.6 | 1.4 | 0.6 | 0.87 | 8.1 | 0.9 | 0.3 | 0.87 | 8.5 | 1.4 | 0.3 | 0.88 | 8.5 | 0.3 | 0.3 | 0.89 |
| | 5.0 | 5.5 | 4.3 | 7.4 | 1.6 | 0.83 | 7.5 | 1.3 | 0.5 | 0.86 | 7.9 | 0.9 | 0.3 | 0.86 | 8.3 | 1.4 | 0.3 | 0.88 | 8.5 | 0.3 | 0.3 | 0.89 |
| LEACE | 1.0 | 8.6 | 0.7 | 1.1 | 8.6 | 0.90 | 8.4 | 1.8 | 0.2 | 0.90 | 8.5 | 1.1 | 0.1 | 0.89 | 8.7 | 1.4 | 0.2 | 0.90 | 8.5 | 0.4 | 0.3 | 0.89 |
| | 2.0 | 6.4 | 4.6 | 3.8 | 7.5 | 0.88 | 8.4 | 1.6 | 0.2 | 0.89 | 8.4 | 1.0 | 0.1 | 0.89 | 8.7 | 1.3 | 0.2 | 0.90 | 8.5 | 0.4 | 0.4 | 0.89 |
| | 3.0 | 4.2 | 6.0 | 6.1 | 4.6 | 0.85 | 8.2 | 1.5 | 0.4 | 0.89 | 8.4 | 1.0 | 0.1 | 0.89 | 8.7 | 1.4 | 0.2 | 0.89 | 8.5 | 0.3 | 0.3 | 0.89 |
| | 4.0 | 4.2 | 6.1 | 6.6 | 3.5 | 0.84 | 7.9 | 1.5 | 0.6 | 0.89 | 8.3 | 1.0 | 0.2 | 0.88 | 8.7 | 1.4 | 0.3 | 0.89 | 8.4 | 0.4 | 0.4 | 0.89 |
| | 5.0 | 4.1 | 5.9 | 6.8 | 3.1 | 0.83 | 7.7 | 1.4 | 0.7 | 0.88 | 8.3 | 1.0 | 0.2 | 0.88 | 8.6 | 1.3 | 0.4 | 0.89 | 8.5 | 0.4 | 0.3 | 0.89 |
| MidSteer | 1.0 | 8.7 | 0.7 | 0.9 | 8.6 | 0.90 | 8.4 | 1.8 | 0.1 | 0.89 | 8.5 | 1.1 | 0.1 | 0.89 | 8.7 | 1.4 | 0.2 | 0.90 | 8.5 | 0.4 | 0.3 | 0.89 |
| | 2.0 | 3.9 | 6.5 | 0.9 | 8.6 | 0.90 | 8.3 | 1.6 | 0.2 | 0.89 | 8.4 | 1.1 | 0.1 | 0.89 | 8.7 | 1.3 | 0.2 | 0.90 | 8.5 | 0.3 | 0.3 | 0.89 |
| | 3.0 | 1.4 | 8.2 | 0.9 | 8.6 | 0.90 | 8.3 | 1.5 | 0.2 | 0.89 | 8.4 | 1.0 | 0.1 | 0.89 | 8.7 | 1.4 | 0.3 | 0.89 | 8.5 | 0.3 | 0.3 | 0.89 |
| | 4.0 | 1.2 | 8.2 | 0.9 | 8.6 | 0.90 | 8.3 | 1.5 | 0.2 | 0.89 | 8.4 | 1.0 | 0.2 | 0.89 | 8.7 | 1.3 | 0.3 | 0.89 | 8.4 | 0.4 | 0.4 | 0.89 |
| | 5.0 | 1.2 | 8.2 | 0.8 | 8.6 | 0.90 | 8.1 | 1.6 | 0.7 | 0.89 | 8.3 | 0.9 | 0.4 | 0.89 | 8.6 | 1.3 | 0.3 | 0.89 | 8.5 | 0.3 | 0.3 | 0.89 |

### K.3. Detailed results for for Diffusion Models concept switching

In this section, in tab. 19,20,16,17 we provide detailed breakdown of scores for all $\beta$ values and all concepts $c_s, c_t, c_i$. Pareto plots in sec. K.0.1 were created based on the scores provided in these tables.

*Table 16.* Model SDXL, flipping from horse to motorcycle

| method | strength | horse src-cs↓ | horse tgt-cs↑ | motorcycle src-cs↓ | motorcycle tgt-cs↑ | motorcycle fid↓ | cs↑ | cow src-cs↓ | cow tgt-cs↓ | cow fid↓ | cs↑ | pig src-cs↓ | pig tgt-cs↓ | pig fid↓ | cs↑ | dog src-cs↓ | dog tgt-cs↓ | dog fid↓ | cs↑ | legislator src-cs↓ | legislator tgt-cs↓ | legislator fid↓ |
|---|---|---|---|---|---|---|---|---|---|---|---|---|---|---|---|---|---|---|---|---|---|---|
| No Steering | - | 71.0 | 49.1 | 51.8 | 70.7 | - | 72.7 | 54.6 | 41.5 | - | 71.8 | 49.5 | 43.6 | - | 66.3 | 52.4 | 44.9 | - | 60.8 | 44.8 | 42.4 | - |
| CASteer | 1.0 | 70.0 | 50.8 | 52.6 | 71.3 | 27.8 | 72.3 | 54.4 | 42.4 | 21.9 | 71.8 | 49.0 | 43.8 | 13.3 | 66.2 | 52.0 | 45.0 | 20.0 | 60.9 | 44.8 | 42.5 | 16.3 |
|  | 1.5 | 53.4 | 68.3 | 60.1 | 63.7 | 121.3 | 72.0 | 54.4 | 43.0 | 28.7 | 71.9 | 48.9 | 44.1 | 16.2 | 66.1 | 51.9 | 45.1 | 24.7 | 60.9 | 44.8 | 42.5 | 21.0 |
|  | 2.0 | 52.1 | 69.5 | 68.3 | 52.9 | 212.4 | 70.9 | 54.4 | 44.9 | 42.7 | 71.9 | 48.9 | 44.4 | 18.9 | 66.1 | 51.8 | 45.4 | 28.6 | 60.9 | 44.8 | 42.6 | 24.6 |
|  | 2.5 | 51.7 | 69.4 | 69.4 | 51.7 | 213.0 | 62.5 | 54.2 | 55.8 | 105.2 | 72.0 | 48.7 | 44.6 | 22.0 | 66.1 | 51.7 | 45.6 | 33.0 | 60.9 | 44.9 | 42.7 | 27.0 |
|  | 3.0 | 51.4 | 69.1 | 69.9 | 50.9 | 210.0 | 52.4 | 53.2 | 66.2 | 186.9 | 72.0 | 48.5 | 44.9 | 25.8 | 66.0 | 51.6 | 45.7 | 37.2 | 60.9 | 44.9 | 42.7 | 29.7 |
|  | 4.0 | 51.0 | 68.7 | 70.6 | 49.9 | 207.9 | 48.6 | 52.3 | 69.5 | 222.6 | 72.0 | 48.5 | 46.5 | 37.2 | 65.7 | 51.4 | 46.7 | 46.5 | 60.9 | 44.9 | 42.8 | 35.0 |
|  | 5.0 | 50.7 | 68.5 | 70.9 | 49.5 | 207.4 | 47.8 | 51.8 | 69.6 | 231.6 | 70.5 | 49.1 | 50.6 | 63.7 | 64.3 | 51.4 | 49.0 | 61.6 | 61.0 | 45.1 | 43.0 | 39.0 |
| LEACE | 1.0 | 65.0 | 56.5 | 52.7 | 71.2 | 28.6 | 72.5 | 54.4 | 42.0 | 17.0 | 71.7 | 49.3 | 43.6 | 8.9 | 66.2 | 52.3 | 44.8 | 14.1 | 60.7 | 44.8 | 42.5 | 21.1 |
|  | 1.5 | 52.1 | 68.6 | 57.1 | 67.0 | 84.6 | 72.2 | 54.4 | 42.3 | 21.3 | 71.7 | 49.3 | 43.7 | 10.9 | 66.2 | 52.3 | 44.8 | 17.7 | 60.7 | 44.8 | 42.5 | 24.9 |
|  | 2.0 | 51.2 | 68.8 | 67.6 | 53.3 | 207.6 | 72.2 | 54.5 | 42.7 | 25.2 | 71.7 | 49.3 | 43.7 | 12.6 | 66.1 | 52.3 | 44.8 | 20.8 | 60.6 | 44.8 | 42.4 | 28.2 |
|  | 2.5 | 50.8 | 68.5 | 69.0 | 51.5 | 213.3 | 71.9 | 54.5 | 43.1 | 30.0 | 71.7 | 49.4 | 43.7 | 14.0 | 66.1 | 52.3 | 44.8 | 22.6 | 60.6 | 44.9 | 42.5 | 31.0 |
|  | 3.0 | 50.5 | 68.2 | 69.6 | 50.5 | 210.6 | 71.4 | 54.4 | 43.8 | 37.0 | 71.7 | 49.4 | 43.8 | 15.1 | 66.0 | 52.2 | 44.8 | 24.5 | 60.6 | 44.8 | 42.5 | 33.2 |
|  | 4.0 | 50.2 | 68.0 | 70.4 | 49.6 | 206.7 | 64.6 | 54.2 | 52.6 | 85.7 | 71.8 | 49.4 | 43.9 | 17.2 | 66.0 | 52.2 | 44.7 | 28.1 | 60.5 | 44.8 | 42.6 | 37.3 |
|  | 5.0 | 49.9 | 67.7 | 70.8 | 49.1 | 204.4 | 54.4 | 53.2 | 63.6 | 166.2 | 71.8 | 49.4 | 44.0 | 19.4 | 65.9 | 52.2 | 44.7 | 31.2 | 60.4 | 44.8 | 42.7 | 42.0 |
| MidSteer | 1.0 | 51.2 | 68.7 | 51.9 | 70.7 | 12.7 | 72.2 | 54.5 | 42.5 | 23.9 | 71.8 | 49.2 | 43.9 | 12.4 | 66.1 | 52.3 | 44.8 | 20.7 | 60.7 | 45.0 | 42.3 | 27.2 |
|  | 1.5 | 50.4 | 68.1 | 51.9 | 70.7 | 15.1 | 71.6 | 54.6 | 43.5 | 34.5 | 71.9 | 49.1 | 44.1 | 15.2 | 66.1 | 52.2 | 44.8 | 25.3 | 60.5 | 45.1 | 42.3 | 32.3 |
|  | 2.0 | 50.0 | 67.8 | 52.0 | 70.8 | 17.2 | 65.9 | 54.3 | 50.9 | 77.3 | 71.9 | 49.0 | 44.3 | 17.6 | 66.1 | 52.1 | 44.8 | 28.8 | 60.6 | 45.2 | 42.3 | 35.9 |
|  | 2.5 | 49.5 | 67.4 | 52.0 | 70.7 | 18.6 | 55.1 | 53.3 | 62.6 | 162.3 | 71.9 | 49.0 | 44.5 | 20.2 | 66.0 | 52.1 | 44.8 | 32.2 | 60.5 | 45.5 | 42.4 | 39.8 |
|  | 3.0 | 49.2 | 67.2 | 52.0 | 70.6 | 20.0 | 51.0 | 52.7 | 66.7 | 199.3 | 72.0 | 49.0 | 44.8 | 23.5 | 66.0 | 52.0 | 44.8 | 35.3 | 60.2 | 45.5 | 42.4 | 43.1 |
|  | 4.0 | 48.8 | 67.2 | 52.0 | 70.5 | 22.9 | 48.4 | 51.9 | 68.7 | 222.5 | 72.1 | 49.0 | 45.5 | 29.5 | 65.8 | 51.9 | 44.9 | 41.4 | 59.7 | 46.0 | 42.7 | 49.9 |
|  | 5.0 | 48.3 | 66.9 | 52.0 | 70.4 | 28.0 | 47.9 | 51.5 | 68.7 | 229.8 | 71.9 | 49.2 | 46.8 | 40.0 | 65.5 | 51.9 | 45.2 | 48.7 | 59.1 | 46.7 | 43.2 | 58.2 |

*Table 17.* Model SDXL, flipping from chihuahua to muffin

| method | strength | chihuahua src-cs↓ | chihuahua tgt-cs↑ | muffin src-cs↓ | muffin tgt-cs↑ | muffin fid↓ | cs↑ | dog src-cs↓ | dog tgt-cs↓ | dog fid↓ | cs↑ | wolf src-cs↓ | wolf tgt-cs↓ | wolf fid↓ | cs↑ | cat src-cs↓ | cat tgt-cs↓ | cat fid↓ | cs↑ | legislator src-cs↓ | legislator tgt-cs↓ | legislator fid↓ |
|---|---|---|---|---|---|---|---|---|---|---|---|---|---|---|---|---|---|---|---|---|---|---|
| No Steering | - | 75.9 | 54.6 | 42.6 | 68.2 | - | 66.3 | 57.9 | 52.5 | - | 71.8 | 52.7 | 45.6 | - | 67.5 | 53.4 | 54.2 | - | 60.8 | 42.6 | 40.1 | - |
| CASteer | 1.0 | 71.7 | 54.7 | 59.3 | 61.5 | 145.9 | 65.7 | 54.4 | 52.4 | 37.4 | 71.9 | 51.7 | 44.7 | 19.1 | 67.2 | 52.8 | 53.9 | 26.2 | 60.8 | 42.4 | 39.9 | 19.2 |
|  | 1.5 | 47.0 | 61.0 | 66.5 | 57.3 | 211.5 | 65.2 | 53.1 | 52.4 | 53.3 | 71.8 | 51.2 | 44.3 | 24.4 | 67.1 | 52.4 | 53.8 | 33.8 | 60.8 | 42.3 | 39.6 | 23.2 |
|  | 2.0 | 43.7 | 63.2 | 69.5 | 56.7 | 226.4 | 63.2 | 51.7 | 53.0 | 79.7 | 71.8 | 51.0 | 43.9 | 30.4 | 66.8 | 52.0 | 53.6 | 41.7 | 60.8 | 42.3 | 39.6 | 26.7 |
|  | 2.5 | 42.3 | 63.8 | 71.5 | 56.3 | 241.1 | 55.2 | 48.9 | 55.5 | 140.5 | 71.5 | 50.6 | 43.6 | 36.8 | 66.3 | 51.7 | 53.6 | 53.8 | 60.9 | 42.3 | 39.6 | 29.8 |
|  | 3.0 | 41.4 | 64.0 | 72.9 | 56.3 | 253.8 | 48.1 | 45.8 | 58.2 | 192.4 | 70.8 | 50.3 | 43.6 | 45.7 | 63.6 | 50.7 | 54.3 | 82.4 | 60.9 | 42.4 | 39.5 | 32.7 |
|  | 4.0 | 40.1 | 64.3 | 74.8 | 56.2 | 276.0 | 44.2 | 43.5 | 60.3 | 226.9 | 65.6 | 49.0 | 45.6 | 96.6 | 52.5 | 47.2 | 57.0 | 168.7 | 60.9 | 42.4 | 39.4 | 38.0 |
|  | 5.0 | 39.3 | 64.1 | 75.6 | 56.1 | 291.6 | 42.9 | 42.5 | 60.8 | 243.3 | 53.5 | 47.0 | 55.6 | 206.1 | 46.9 | 45.2 | 58.0 | 213.0 | 60.8 | 42.5 | 39.3 | 43.9 |
| LEACE | 1.0 | 67.1 | 55.7 | 60.4 | 60.6 | 160.6 | 66.0 | 55.8 | 52.4 | 25.4 | 72.0 | 52.1 | 45.3 | 11.9 | 67.5 | 53.2 | 54.1 | 17.5 | 60.8 | 42.5 | 40.2 | 20.0 |
|  | 1.5 | 46.2 | 61.5 | 66.4 | 57.3 | 212.1 | 65.9 | 54.9 | 52.3 | 33.1 | 72.1 | 52.0 | 45.1 | 14.7 | 67.5 | 53.1 | 54.0 | 21.1 | 60.9 | 42.4 | 40.3 | 24.6 |
|  | 2.0 | 42.8 | 63.3 | 69.3 | 56.9 | 227.7 | 65.8 | 54.2 | 52.3 | 40.3 | 72.1 | 51.8 | 44.9 | 16.9 | 67.6 | 53.0 | 53.9 | 24.6 | 60.9 | 42.3 | 40.3 | 28.1 |
|  | 2.5 | 41.4 | 63.6 | 71.0 | 56.7 | 240.7 | 65.5 | 53.4 | 52.2 | 48.6 | 72.1 | 51.5 | 44.7 | 19.0 | 67.6 | 52.9 | 53.9 | 27.8 | 61.0 | 42.3 | 40.4 | 31.3 |
|  | 3.0 | 40.6 | 63.9 | 72.3 | 56.7 | 250.8 | 65.1 | 52.5 | 52.3 | 57.1 | 72.2 | 51.4 | 44.6 | 21.5 | 67.6 | 52.7 | 53.8 | 30.4 | 61.0 | 42.5 | 40.5 | 34.0 |
|  | 4.0 | 39.3 | 63.8 | 73.7 | 56.6 | 269.4 | 63.8 | 51.3 | 52.6 | 76.7 | 72.2 | 51.0 | 44.2 | 25.7 | 67.5 | 52.6 | 53.8 | 35.4 | 61.0 | 42.7 | 40.6 | 39.4 |
|  | 5.0 | 38.5 | 63.7 | 74.6 | 56.5 | 282.5 | 60.3 | 49.7 | 53.7 | 104.1 | 72.2 | 50.7 | 44.0 | 29.8 | 67.4 | 52.4 | 53.7 | 41.1 | 61.0 | 43.0 | 41.1 | 43.4 |
| MidSteer | 1.0 | 44.3 | 62.5 | 42.0 | 68.3 | 23.9 | 65.9 | 54.7 | 52.4 | 35.1 | 72.2 | 51.7 | 44.9 | 16.9 | 67.6 | 53.0 | 54.1 | 22.7 | 60.8 | 42.5 | 40.1 | 23.9 |
|  | 1.5 | 41.5 | 63.9 | 41.9 | 68.2 | 28.3 | 65.5 | 53.5 | 52.4 | 48.8 | 72.2 | 51.2 | 44.6 | 21.2 | 67.7 | 52.8 | 54.1 | 27.7 | 60.8 | 42.4 | 40.2 | 29.0 |
|  | 2.0 | 39.8 | 64.0 | 41.7 | 68.2 | 32.2 | 64.5 | 52.2 | 52.7 | 65.6 | 72.2 | 50.8 | 44.1 | 25.4 | 67.7 | 52.6 | 54.1 | 31.6 | 60.9 | 42.5 | 40.4 | 32.5 |
|  | 2.5 | 38.9 | 64.3 | 41.6 | 68.1 | 35.7 | 63.2 | 51.3 | 53.1 | 83.0 | 72.2 | 50.6 | 44.0 | 29.0 | 67.6 | 52.4 | 54.0 | 35.7 | 61.0 | 42.4 | 40.7 | 36.3 |
|  | 3.0 | 38.0 | 64.2 | 41.5 | 68.0 | 39.5 | 59.4 | 49.7 | 54.4 | 109.8 | 72.2 | 50.3 | 43.8 | 33.4 | 67.5 | 52.2 | 54.0 | 40.1 | 60.9 | 42.4 | 40.7 | 39.4 |
|  | 4.0 | 37.5 | 64.0 | 41.3 | 67.9 | 48.2 | 48.3 | 44.8 | 58.7 | 178.3 | 71.6 | 50.3 | 44.2 | 43.9 | 67.0 | 51.8 | 54.3 | 50.7 | 60.6 | 42.4 | 41.1 | 45.1 |
|  | 5.0 | 37.9 | 63.6 | 41.3 | 67.5 | 58.9 | 43.4 | 42.2 | 61.1 | 209.8 | 70.2 | 50.4 | 44.8 | 63.1 | 64.5 | 50.7 | 54.5 | 69.6 | 60.4 | 42.5 | 41.5 | 51.4 |

*Table 18.* Model SDXL, flipping from snoopy to mickey

| method | strength | snoopy | | mickey | | | pikachu | | | | spongebob | | | | dog | | | | legislator | | | |
|---|---|---|---|---|---|---|---|---|---|---|---|---|---|---|---|---|---|---|---|---|---|---|
| | | src-cs↓ | tgt-cs↑ | src-cs↓ | tgt-cs↑ | fid↓ | cs↑ | src-cs↓ | tgt-cs↓ | fid↓ | cs↑ | src-cs↓ | tgt-cs↓ | fid↓ | cs↑ | src-cs↓ | tgt-cs↓ | fid↓ | cs↑ | src-cs↓ | tgt-cs↓ | fid↓ |
| No Steering | - | 74.3 | 58.7 | 56.0 | 73.1 | - | 72.6 | 41.3 | 51.2 | - | 75.1 | 49.0 | 52.5 | - | 66.3 | 56.1 | 52.0 | - | 60.8 | 41.6 | 45.0 | - |
| CASteer | 1.0 | 65.6 | 68.4 | 58.9 | 70.8 | 47.5 | 72.7 | 41.4 | 51.2 | 13.8 | 75.1 | 48.9 | 52.5 | 25.0 | 66.4 | 55.0 | 52.1 | 25.4 | 60.8 | 41.6 | 45.0 | 14.9 |
| | 1.5 | 59.8 | 71.2 | 61.7 | 68.6 | 59.6 | 72.7 | 41.4 | 51.2 | 17.1 | 75.1 | 48.9 | 52.6 | 29.7 | 66.3 | 54.4 | 52.1 | 33.9 | 60.7 | 41.6 | 45.0 | 18.4 |
| | 2.0 | 57.0 | 72.4 | 66.8 | 65.3 | 72.3 | 72.7 | 41.5 | 51.1 | 19.4 | 75.2 | 48.9 | 52.6 | 33.5 | 66.4 | 53.9 | 52.2 | 40.6 | 60.8 | 41.5 | 45.0 | 21.1 |
| | 2.5 | 55.4 | 72.7 | 69.6 | 61.3 | 83.9 | 72.7 | 41.6 | 51.2 | 21.2 | 75.0 | 48.8 | 52.5 | 36.5 | 66.4 | 53.3 | 52.2 | 48.5 | 60.9 | 41.6 | 45.0 | 23.3 |
| | 3.0 | 54.5 | 72.7 | 70.3 | 59.4 | 91.8 | 72.7 | 41.6 | 51.2 | 22.9 | 75.1 | 48.9 | 52.6 | 39.1 | 66.4 | 52.8 | 52.3 | 54.8 | 60.8 | 41.6 | 45.0 | 25.6 |
| | 4.0 | 53.0 | 71.9 | 70.8 | 57.4 | 106.3 | 72.7 | 41.7 | 51.2 | 25.9 | 75.2 | 48.7 | 52.6 | 43.3 | 66.4 | 52.1 | 52.6 | 66.5 | 60.7 | 41.6 | 45.0 | 28.9 |
| | 5.0 | 52.0 | 71.3 | 71.2 | 56.4 | 117.5 | 72.7 | 41.7 | 51.2 | 28.9 | 75.0 | 48.7 | 52.7 | 47.7 | 66.3 | 51.3 | 52.8 | 79.9 | 60.7 | 41.5 | 44.9 | 31.5 |
| LEACE | 1.0 | 65.9 | 67.9 | 58.8 | 70.6 | 50.0 | 72.8 | 41.4 | 51.1 | 18.5 | 75.0 | 48.9 | 52.5 | 33.5 | 66.3 | 55.8 | 52.0 | 18.6 | 60.9 | 41.7 | 45.1 | 22.2 |
| | 1.5 | 60.1 | 71.0 | 61.0 | 68.1 | 62.6 | 72.8 | 41.5 | 51.1 | 22.0 | 75.0 | 48.8 | 52.5 | 39.0 | 66.3 | 55.8 | 52.1 | 23.9 | 60.9 | 41.7 | 45.0 | 26.4 |
| | 2.0 | 57.5 | 72.2 | 64.3 | 64.6 | 77.0 | 72.9 | 41.5 | 51.1 | 24.5 | 75.1 | 48.9 | 52.5 | 43.6 | 66.3 | 55.6 | 52.1 | 28.4 | 60.9 | 41.9 | 45.2 | 29.7 |
| | 2.5 | 56.0 | 72.7 | 67.3 | 60.7 | 89.9 | 72.9 | 41.6 | 51.1 | 26.7 | 75.2 | 48.9 | 52.5 | 47.7 | 66.3 | 55.5 | 52.2 | 32.0 | 61.0 | 41.9 | 45.1 | 31.9 |
| | 3.0 | 55.1 | 72.7 | 68.2 | 58.2 | 101.5 | 73.0 | 41.7 | 51.2 | 29.1 | 75.2 | 48.8 | 52.5 | 51.3 | 66.4 | 55.4 | 52.3 | 35.7 | 61.0 | 41.9 | 45.1 | 34.8 |
| | 4.0 | 53.7 | 72.3 | 68.7 | 55.9 | 118.6 | 73.2 | 41.9 | 51.2 | 33.4 | 75.0 | 49.0 | 52.6 | 57.7 | 66.5 | 55.3 | 52.5 | 43.4 | 60.9 | 42.1 | 45.2 | 38.9 |
| | 5.0 | 53.0 | 72.0 | 69.5 | 55.2 | 134.6 | 73.4 | 42.4 | 51.4 | 37.6 | 74.8 | 49.1 | 52.7 | 64.0 | 66.5 | 55.2 | 52.7 | 50.0 | 60.9 | 42.3 | 45.3 | 42.2 |
| MidSteer | 1.0 | 59.7 | 71.0 | 55.5 | 72.8 | 30.1 | 72.8 | 41.3 | 51.3 | 19.0 | 74.7 | 48.9 | 52.6 | 36.5 | 66.2 | 55.9 | 52.1 | 20.4 | 60.9 | 41.7 | 45.0 | 24.5 |
| | 1.5 | 56.0 | 72.5 | 55.3 | 72.7 | 34.8 | 72.9 | 41.2 | 51.2 | 22.3 | 74.6 | 48.6 | 52.6 | 42.7 | 66.2 | 55.8 | 52.1 | 25.9 | 60.8 | 41.8 | 45.0 | 29.1 |
| | 2.0 | 54.5 | 72.7 | 55.0 | 72.4 | 38.9 | 72.9 | 41.3 | 51.3 | 25.3 | 74.5 | 48.6 | 52.6 | 46.8 | 66.3 | 55.6 | 52.1 | 30.6 | 60.7 | 42.0 | 45.1 | 32.6 |
| | 2.5 | 53.4 | 72.4 | 54.8 | 72.4 | 42.2 | 73.0 | 41.2 | 51.3 | 27.3 | 74.3 | 48.4 | 52.6 | 50.8 | 66.3 | 55.6 | 52.3 | 34.4 | 60.7 | 42.0 | 45.1 | 35.2 |
| | 3.0 | 52.5 | 71.8 | 54.7 | 72.2 | 44.7 | 73.1 | 41.3 | 51.3 | 30.0 | 74.2 | 48.5 | 52.7 | 55.2 | 66.4 | 55.5 | 52.5 | 38.3 | 60.7 | 42.1 | 45.3 | 37.6 |
| | 4.0 | 51.5 | 71.2 | 54.6 | 71.9 | 49.6 | 73.2 | 41.4 | 51.4 | 34.5 | 73.4 | 48.4 | 53.0 | 64.7 | 66.4 | 55.3 | 52.7 | 47.2 | 60.4 | 42.3 | 45.4 | 43.3 |
| | 5.0 | 51.0 | 70.4 | 54.4 | 71.5 | 55.1 | 73.3 | 41.3 | 51.4 | 39.1 | 72.7 | 48.6 | 53.4 | 72.9 | 66.5 | 55.1 | 52.9 | 55.3 | 60.3 | 42.5 | 45.6 | 48.8 |

*Table 19.* Model SANA, flipping from horse to motorcycle

| method | strength | horse | | motorcycle | | | cow | | | | pig | | | | dog | | | | legislator | | | |
|---|---|---|---|---|---|---|---|---|---|---|---|---|---|---|---|---|---|---|---|---|---|---|
| | | src-cs↓ | tgt-cs↑ | src-cs↓ | tgt-cs↑ | fid↓ | cs↑ | src-cs↓ | tgt-cs↓ | fid↓ | cs↑ | src-cs↓ | tgt-cs↓ | fid↓ | cs↑ | src-cs↓ | tgt-cs↓ | fid↓ | cs↑ | src-cs↓ | tgt-cs↓ | fid↓ |
| No Steering | - | 72.1 | 50.6 | 50.9 | 70.5 | - | 73.8 | 55.3 | 42.5 | - | 73.5 | 48.3 | 44.5 | - | 68.1 | 51.7 | 46.1 | - | 60.4 | 45.8 | 43.9 | - |
| CASteer | 1.0 | 71.0 | 52.0 | 52.2 | 71.3 | 48.1 | 73.6 | 54.8 | 43.0 | 22.1 | 73.8 | 48.0 | 44.8 | 13.8 | 68.1 | 51.6 | 46.2 | 15.7 | 60.2 | 46.0 | 44.3 | 15.9 |
| | 2.0 | 65.5 | 60.4 | 67.5 | 56.0 | 229.0 | 73.1 | 54.2 | 43.7 | 37.9 | 74.0 | 47.7 | 45.0 | 22.6 | 68.1 | 51.4 | 46.3 | 24.1 | 60.0 | 46.0 | 44.5 | 21.5 |
| | 3.0 | 54.9 | 68.7 | 69.5 | 51.8 | 234.6 | 72.3 | 54.0 | 45.0 | 55.8 | 74.2 | 47.4 | 45.3 | 30.5 | 68.1 | 51.3 | 46.5 | 32.2 | 59.7 | 46.3 | 44.9 | 25.9 |
| | 4.0 | 52.6 | 70.3 | 70.2 | 50.7 | 228.7 | 69.3 | 55.1 | 53.0 | 107.1 | 74.5 | 47.4 | 45.9 | 39.9 | 68.2 | 51.2 | 46.7 | 40.4 | 59.5 | 46.5 | 45.3 | 31.1 |
| | 5.0 | 51.8 | 71.0 | 70.5 | 50.3 | 224.9 | 62.3 | 54.8 | 61.7 | 167.0 | 74.7 | 47.5 | 46.9 | 48.9 | 68.2 | 51.2 | 47.0 | 51.3 | 59.3 | 46.7 | 45.8 | 36.8 |
| LEACE | 1.0 | 68.7 | 56.2 | 51.6 | 71.6 | 36.3 | 73.7 | 55.2 | 42.6 | 10.2 | 73.6 | 48.2 | 44.7 | 8.0 | 68.1 | 51.8 | 46.0 | 8.1 | 60.4 | 45.7 | 43.9 | 11.6 |
| | 2.0 | 53.1 | 70.7 | 58.7 | 69.4 | 125.0 | 73.7 | 55.2 | 42.8 | 16.2 | 73.7 | 48.1 | 44.7 | 12.6 | 68.1 | 51.8 | 46.0 | 13.1 | 60.5 | 45.7 | 43.8 | 17.0 |
| | 3.0 | 51.9 | 71.3 | 65.7 | 58.2 | 229.3 | 73.7 | 55.1 | 43.1 | 21.0 | 69.8 | 57.6 | 52.1 | 220.6 | 68.0 | 51.8 | 46.0 | 17.7 | 60.7 | 45.8 | 43.8 | 21.6 |
| | 4.0 | 51.9 | 71.1 | 67.4 | 53.7 | 242.8 | 73.7 | 55.0 | 43.3 | 25.9 | 73.9 | 48.0 | 44.9 | 20.4 | 68.0 | 51.8 | 46.0 | 21.5 | 55.1 | 52.2 | 51.9 | 294.6 |
| | 5.0 | 51.7 | 70.8 | 68.4 | 52.6 | 236.2 | 73.7 | 54.9 | 43.6 | 31.8 | 74.0 | 48.0 | 44.9 | 24.0 | 68.0 | 51.8 | 45.9 | 25.0 | 60.8 | 45.8 | 43.8 | 30.1 |
| MidSteer | 1.0 | 55.5 | 68.9 | 50.9 | 70.4 | 8.7 | 73.6 | 55.1 | 42.7 | 14.7 | 73.7 | 48.2 | 44.7 | 10.4 | 68.0 | 51.8 | 46.1 | 9.3 | 60.5 | 45.8 | 44.0 | 13.0 |
| | 2.0 | 51.7 | 71.1 | 50.9 | 70.4 | 11.7 | 73.6 | 55.0 | 43.1 | 23.2 | 73.9 | 48.1 | 44.8 | 16.5 | 68.0 | 51.7 | 46.1 | 15.0 | 60.6 | 45.8 | 43.9 | 19.1 |
| | 3.0 | 51.4 | 70.7 | 50.9 | 70.4 | 13.9 | 73.4 | 54.7 | 43.4 | 32.5 | 74.1 | 48.0 | 45.0 | 22.2 | 67.9 | 51.7 | 46.1 | 19.8 | 60.8 | 45.9 | 44.0 | 23.9 |
| | 4.0 | 51.4 | 70.9 | 50.9 | 70.4 | 15.9 | 73.1 | 54.5 | 43.9 | 43.8 | 74.3 | 48.1 | 45.3 | 27.4 | 67.9 | 51.6 | 46.1 | 24.3 | 60.8 | 45.9 | 44.1 | 28.9 |
| | 5.0 | 51.5 | 71.4 | 51.0 | 70.4 | 18.0 | 69.8 | 55.2 | 50.0 | 78.2 | 74.5 | 48.1 | 45.7 | 34.3 | 67.9 | 51.6 | 46.2 | 28.4 | 61.0 | 46.0 | 44.2 | 34.3 |

*Table 20.* Model SANA, flipping from chihuahua to muffin

| method | strength | chihuahua | | muffin | | | dog | | | | wolf | | | | cat | | | | legislator | | | |
|---|---|---|---|---|---|---|---|---|---|---|---|---|---|---|---|---|---|---|---|---|---|---|
| | | src-cs↓ | tgt-cs↑ | src-cs↓ | tgt-cs↑ | fid↓ | cs↑ | src-cs↓ | tgt-cs↓ | fid↓ | cs↑ | src-cs↓ | tgt-cs↓ | fid↓ | cs↑ | src-cs↓ | tgt-cs↓ | fid↓ | cs↑ | src-cs↓ | tgt-cs↓ | fid↓ |
| No Steering | - | 76.4 | 55.0 | 43.4 | 66.3 | - | 68.1 | 62.0 | 52.7 | - | 73.2 | 52.8 | 46.1 | - | 68.5 | 53.4 | 53.0 | - | 60.4 | 42.7 | 40.8 | - |
| CASteer | 1.0 | 75.8 | 55.8 | 45.3 | 65.6 | 46.6 | 67.7 | 58.4 | 52.9 | 39.4 | 73.1 | 52.7 | 45.8 | 16.3 | 68.3 | 52.8 | 53.5 | 24.5 | 60.2 | 42.8 | 41.0 | 13.8 |
| | 2.0 | 55.3 | 60.1 | 62.0 | 60.4 | 200.7 | 67.1 | 56.3 | 54.1 | 78.8 | 72.9 | 52.4 | 45.6 | 28.4 | 68.3 | 52.1 | 54.3 | 39.2 | 60.1 | 42.7 | 41.1 | 18.9 |
| | 3.0 | 45.4 | 60.5 | 60.0 | 47.6 | 278.5 | 50.6 | 49.0 | 59.2 | 198.9 | 72.4 | 52.4 | 45.8 | 46.2 | 67.9 | 51.0 | 55.0 | 61.7 | 59.9 | 42.8 | 41.3 | 23.4 |
| | 4.0 | 44.7 | 60.7 | 70.6 | 54.9 | 250.7 | 45.1 | 46.0 | 60.8 | 223.3 | 68.6 | 52.0 | 46.6 | 113.8 | 65.5 | 50.1 | 55.5 | 109.2 | 59.7 | 42.9 | 41.4 | 27.6 |
| | 5.0 | 44.1 | 60.7 | 72.1 | 54.2 | 260.7 | 43.4 | 45.3 | 61.1 | 229.3 | 58.9 | 51.1 | 50.3 | 216.2 | 56.9 | 48.0 | 57.6 | 179.1 | 59.9 | 42.7 | 41.6 | 31.3 |
| LEACE | 1.0 | 72.3 | 58.3 | 46.1 | 65.0 | 57.7 | 67.9 | 59.9 | 52.7 | 21.0 | 73.2 | 52.9 | 46.1 | 5.8 | 68.4 | 53.2 | 52.9 | 8.9 | 60.4 | 42.7 | 40.8 | 9.1 |
| | 2.0 | 48.9 | 61.4 | 59.8 | 63.0 | 185.8 | 67.8 | 58.3 | 52.8 | 35.1 | 73.2 | 52.9 | 46.1 | 9.4 | 68.4 | 53.1 | 53.0 | 13.6 | 60.4 | 42.5 | 40.8 | 13.7 |
| | 3.0 | 45.5 | 60.7 | 65.7 | 57.2 | 227.5 | 67.7 | 57.1 | 52.8 | 48.4 | 73.2 | 52.9 | 46.0 | 12.5 | 68.5 | 53.1 | 53.0 | 17.5 | 60.4 | 42.7 | 40.7 | 16.9 |
| | 4.0 | 44.6 | 60.5 | 68.1 | 55.8 | 240.4 | 67.7 | 56.3 | 53.0 | 60.8 | 73.2 | 52.9 | 46.0 | 15.4 | 68.5 | 53.0 | 53.0 | 21.0 | 60.4 | 42.7 | 40.7 | 19.4 |
| | 5.0 | 44.2 | 60.1 | 69.4 | 55.1 | 246.3 | 67.4 | 55.9 | 53.6 | 76.7 | 73.1 | 52.9 | 46.0 | 18.1 | 68.6 | 52.9 | 53.0 | 24.0 | 60.4 | 42.9 | 40.6 | 21.5 |
| MidSteer | 1.0 | 53.3 | 60.5 | 43.4 | 66.2 | 7.8 | 66.2 | 62.5 | 48.3 | 216.4 | 73.2 | 52.8 | 45.9 | 7.1 | 68.4 | 53.2 | 53.1 | 12.1 | 60.4 | 42.6 | 40.8 | 10.7 |
| | 2.0 | 44.4 | 58.7 | 43.4 | 66.0 | 13.1 | 67.6 | 56.8 | 53.0 | 57.9 | 73.1 | 52.8 | 45.8 | 11.2 | 68.4 | 52.9 | 53.1 | 17.8 | 60.4 | 42.5 | 40.8 | 15.8 |
| | 3.0 | 44.1 | 56.4 | 43.5 | 65.8 | 17.7 | 65.8 | 55.4 | 54.4 | 96.5 | 73.0 | 52.8 | 45.7 | 14.5 | 68.5 | 52.7 | 53.2 | 23.0 | 60.3 | 42.4 | 40.8 | 19.8 |
| | 4.0 | 45.0 | 53.8 | 43.6 | 65.6 | 22.3 | 56.4 | 51.6 | 56.3 | 154.4 | 73.0 | 52.9 | 45.6 | 17.3 | 68.5 | 52.5 | 53.2 | 27.2 | 60.2 | 42.5 | 40.8 | 22.9 |
| | 5.0 | 46.1 | 51.8 | 43.6 | 65.5 | 27.3 | 49.3 | 48.5 | 57.2 | 193.2 | 73.0 | 52.9 | 45.5 | 19.9 | 68.5 | 52.2 | 53.4 | 31.3 | 60.1 | 42.4 | 40.6 | 25.9 |

*Table 21.* Model SANA, flipping from snoopy to mickey

| method | strength | snoopy | | mickey | | | pikachu | | | | spongebob | | | | dog | | | | legislator | | | |
|---|---|---|---|---|---|---|---|---|---|---|---|---|---|---|---|---|---|---|---|---|---|---|
| | | src-cs ↓ | tgt-cs ↑ | src-cs ↓ | tgt-cs ↑ | fid ↓ | cs ↑ | src-cs ↓ | tgt-cs ↓ | fid ↓ | cs ↑ | src-cs ↓ | tgt-cs ↓ | fid ↓ | cs ↑ | src-cs ↓ | tgt-cs ↓ | fid ↓ | cs ↑ | src-cs ↓ | tgt-cs ↓ | fid ↓ |
| No Steering | - | 79.7 | 58.0 | 56.3 | 76.1 | - | 74.0 | 41.5 | 50.9 | - | 79.0 | 50.7 | 53.8 | - | 67.3 | 57.0 | 57.1 | - | 60.4 | 42.9 | 46.6 | - |
| CASteer | 1.0 | 77.9 | 61.7 | 57.8 | 76.3 | 31.8 | 74.1 | 41.5 | 50.8 | 8.6 | 79.0 | 50.7 | 53.8 | 14.5 | 68.1 | 54.0 | 52.0 | 218.8 | 60.3 | 43.0 | 46.8 | 12.5 |
| | 2.0 | 65.6 | 71.0 | 61.0 | 76.5 | 53.8 | 74.1 | 41.6 | 50.8 | 13.9 | 78.9 | 50.8 | 53.8 | 19.8 | 68.2 | 53.5 | 52.0 | 218.3 | 60.2 | 43.0 | 46.9 | 17.7 |
| | 3.0 | 55.8 | 72.3 | 71.7 | 74.8 | 78.8 | 74.2 | 41.7 | 50.7 | 18.6 | 78.9 | 50.8 | 53.8 | 23.9 | 68.1 | 53.2 | 52.1 | 218.3 | 60.1 | 43.0 | 46.8 | 21.4 |
| | 4.0 | 52.0 | 71.4 | 78.5 | 67.6 | 102.9 | 74.3 | 41.8 | 50.7 | 22.7 | 78.9 | 50.8 | 53.8 | 27.9 | 68.2 | 52.8 | 52.1 | 218.5 | 60.1 | 43.0 | 47.0 | 25.4 |
| | 5.0 | 50.0 | 70.4 | 79.8 | 61.9 | 121.0 | 74.5 | 42.0 | 50.7 | 26.7 | 79.0 | 50.8 | 53.8 | 31.8 | 68.2 | 52.4 | 52.2 | 218.0 | 60.1 | 43.0 | 47.2 | 28.5 |
| LEACE | 1.0 | 76.6 | 63.0 | 58.2 | 76.4 | 36.0 | 74.1 | 41.6 | 51.0 | 3.2 | 79.0 | 50.7 | 53.7 | 10.8 | 68.1 | 54.2 | 52.0 | 219.3 | 60.2 | 42.9 | 46.6 | 5.9 |
| | 2.0 | 62.6 | 71.3 | 57.9 | 72.9 | 170.9 | 74.1 | 41.7 | 51.0 | 5.5 | 78.9 | 50.7 | 53.7 | 15.1 | 68.0 | 54.0 | 52.0 | 220.0 | 60.2 | 42.9 | 46.6 | 9.6 |
| | 3.0 | 55.9 | 71.7 | 75.1 | 73.4 | 88.9 | 74.2 | 41.7 | 51.0 | 7.5 | 78.9 | 50.7 | 53.7 | 18.2 | 68.0 | 53.7 | 52.1 | 218.7 | 60.1 | 42.9 | 46.7 | 12.2 |
| | 4.0 | 52.9 | 68.8 | 79.2 | 66.4 | 111.0 | 74.2 | 41.8 | 51.1 | 9.5 | 78.9 | 50.8 | 53.8 | 21.1 | 68.0 | 53.6 | 52.1 | 218.8 | 60.1 | 42.9 | 46.7 | 14.6 |
| | 5.0 | 51.0 | 65.5 | 79.6 | 60.8 | 130.5 | 74.3 | 41.9 | 51.1 | 11.2 | 78.9 | 50.8 | 53.8 | 24.1 | 68.0 | 53.3 | 52.2 | 219.7 | 59.9 | 42.9 | 46.7 | 16.4 |
| MidSteer | 1.0 | 64.9 | 70.8 | 56.0 | 76.2 | 16.0 | 74.1 | 41.5 | 51.0 | 5.8 | 79.1 | 50.6 | 53.8 | 12.3 | 68.1 | 54.0 | 52.0 | 219.2 | 60.2 | 42.9 | 46.7 | 8.3 |
| | 2.0 | 53.5 | 71.1 | 55.6 | 76.2 | 25.2 | 74.2 | 41.5 | 51.1 | 10.0 | 79.2 | 50.6 | 53.9 | 17.4 | 68.1 | 53.6 | 52.1 | 218.2 | 60.1 | 42.9 | 46.7 | 12.4 |
| | 3.0 | 50.0 | 67.4 | 55.2 | 76.2 | 34.8 | 74.4 | 41.5 | 51.2 | 13.5 | 79.2 | 50.6 | 54.0 | 21.6 | 68.0 | 53.2 | 52.2 | 218.6 | 59.9 | 42.8 | 46.7 | 15.6 |
| | 4.0 | 48.4 | 62.6 | 54.9 | 76.4 | 45.4 | 74.5 | 41.6 | 51.3 | 16.8 | 79.2 | 50.5 | 54.0 | 25.8 | 67.9 | 52.6 | 52.3 | 216.6 | 59.8 | 42.8 | 46.7 | 17.8 |
| | 5.0 | 48.1 | 60.7 | 54.5 | 76.6 | 55.3 | 74.6 | 41.5 | 51.3 | 20.0 | 79.2 | 50.3 | 54.1 | 29.6 | 68.0 | 52.3 | 52.4 | 215.1 | 55.0 | 47.5 | 52.8 | 271.2 |

# L. Concept erasure

In this section, we provide experiments for concept erasure. The reason for such is to show that unified MidSteer framework, that also includes erasure in case $Z_2$ is constant, performs favourably on a variety of tasks. Note that in case of erasure, both MidSteer and LEACE provide the same solutions, so we will label the results as LEACE / MidSteer in charts.

For concept erasure, we use evaluation setup seimilar to that of concept switching. For LLM, We test erasure of the concepts $c_s \in$ ("Horse", $c_2$ ="Dog"). Corresponding testing concepts are $t_i = \{c_i\}_{i=1}^5$: $t_1$ = ("Motorcycle", "Cow", "Dog", "Pig", "Legislator"), $t_2$ = ("Cat", "Cow", "Wolf", "Pig", "Legislator"). For Diffusion Models, We test erasure of the concepts $c_s \in$ ("Horse", $c_2$ ="Chichuahua"). Corresponding testing concepts are $t_i = \{c_i\}_{i=1}^5$: $t_1$ = ("Motorcycle", "Cow", "Dog", "Pig", "Legislator"), $t_2$ = ("Muffin", "Cat", "Dog", "Wolf", "Legislator"),

The setup for erasure is similar to those of main paper experiments (Sec. 5). The only difference is that there is no target concept (i.e. it is a dummy concept). We utilise similar pairs of prompts $(c_1, c_2)$ as in switching experiments, with the goal of removing $c_1$.

Fig. 21b,21d,21a,21c present Pareto plots for concept erasure similar to that for concept switching in sec. 5. More precisely, we compare results on concept switching by applying vanilla steering, LEACE (which is equivalent to MidSteer in the case of erasure) with different values of $\beta$. We present results on LLama-2-7b and SDXL models aggregated for all three erase concepts. It can be clearly seen, that in each case, MidSteer achieves much better balance between level of concept switch between c1 and c2 and preservation of other concepts across different values of $\beta$.

Next, in sec. L.1, L.2 we provide detailed Pareto plots for each model, and in sec. L.3,L.4 we provide Tables with detailed breakdown of scores for all $\beta$ values and all erased concepts.

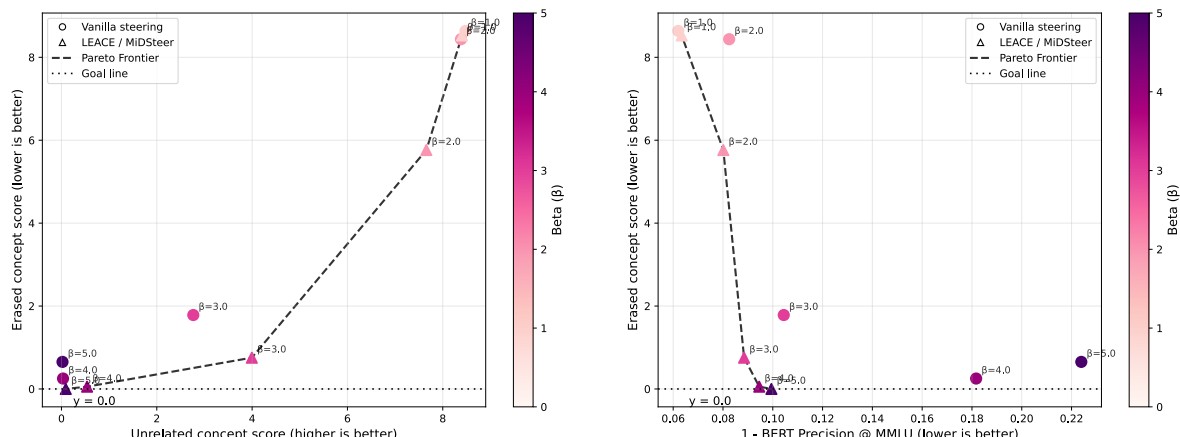

*(a)* Erased concept score vs CS of unrelated concepts on Llama-2-7b model.

*(b)* Erased concept score vs 1 - BERT Precision on MMLU on Llama-2-7b model.

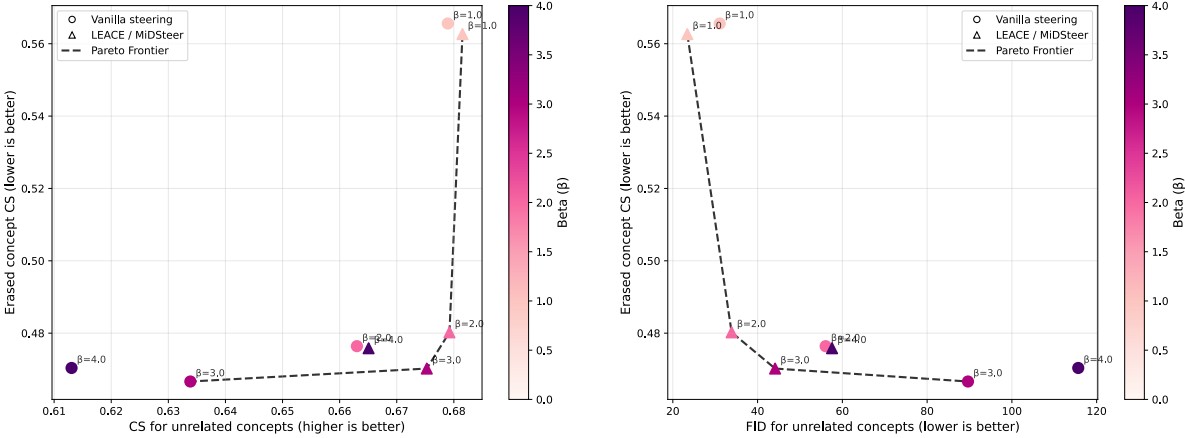

*(c)* Erased concept score vs CS of unrelated concepts on SDXL model.

*(d)* Erased concept score vs FID of unrelated concepts on SDXL model.

*Figure 21.* Pareto efficiency frontiers for concept *erasure* experiments with vanilla steering and LEACE / MidSteer highlighting different $\beta$.

## L.1. Pareto charts for LLM concept erasure

In this section, we provide more Pareto charts for LLM concept erasure. When erasing concept $c_s$, for each LLM model we provide 3 types of Pareto plots:

- 1 - BERT Precision score for unrelated concepts (horizontal axis) vs Concept Score (CS) for the erased $c_s$ concept (vertical axis) (fig. 22b,23b,24a)

- 1 - BERT Precision score for MMLU (horizontal axis) vs Concept Score (CS) for the erased $c_s$ concept (vertical axis) (fig. 22a,23a,24a)

- Average Concept Score (CS) for unrelated concepts $c_i$ (horizontal axis) vs Concept Score (CS) for the erased $c_s$ concept (vertical axis) (fig. 22c,23c,24c)

In each case, we see clear superiority of LEACE/MidSteer over other steering approaches.

We additionally provide detailed breakdown of scores for all $\beta$ values and all concepts $c_s, c_i$ in the tables in sec. L.3

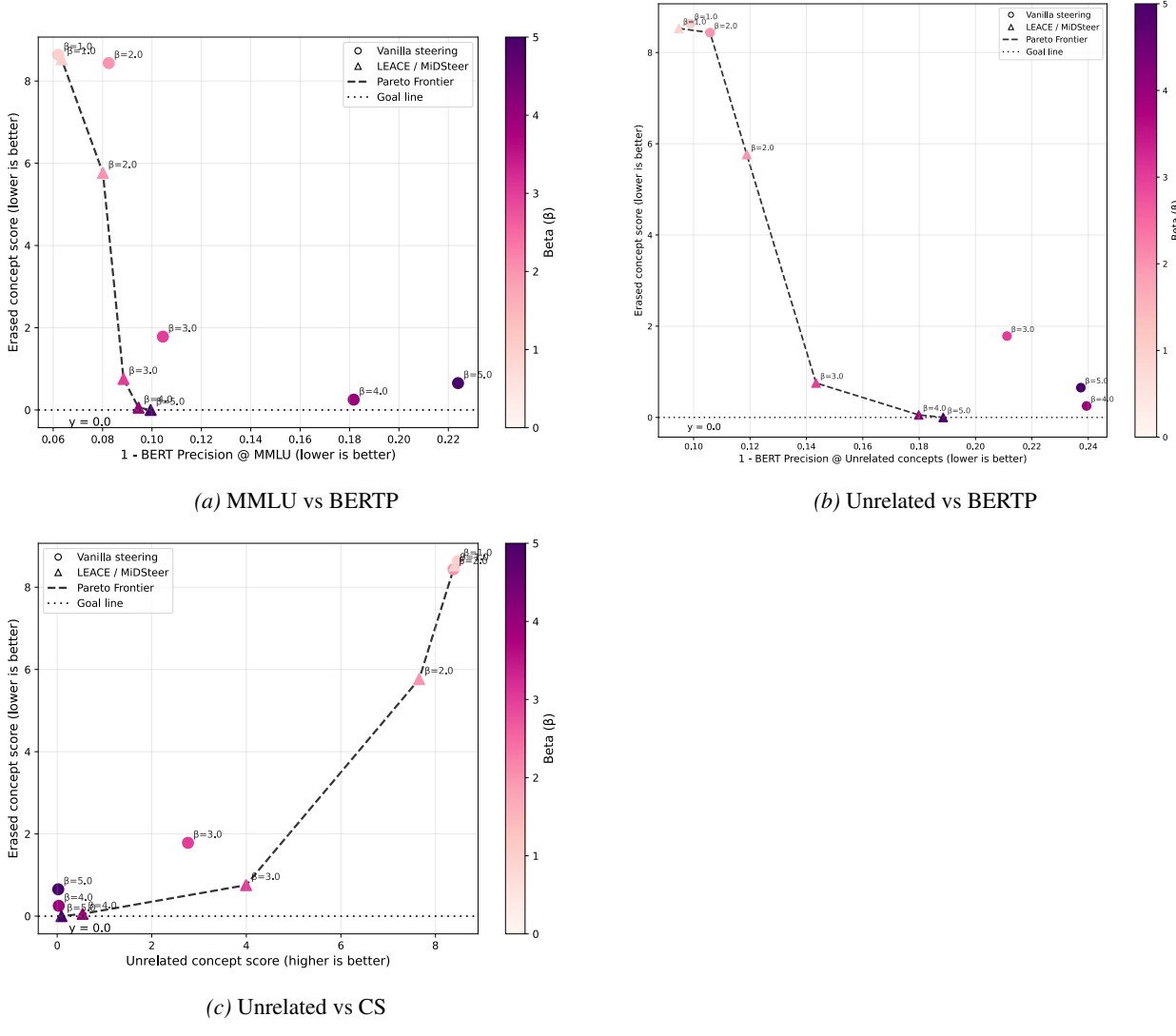

*(a)* MMLU vs BERTP

*(b)* Unrelated vs BERTP

*(c)* Unrelated vs CS

*Figure 22.* Pareto plot for concept erasure on model llama2-7b

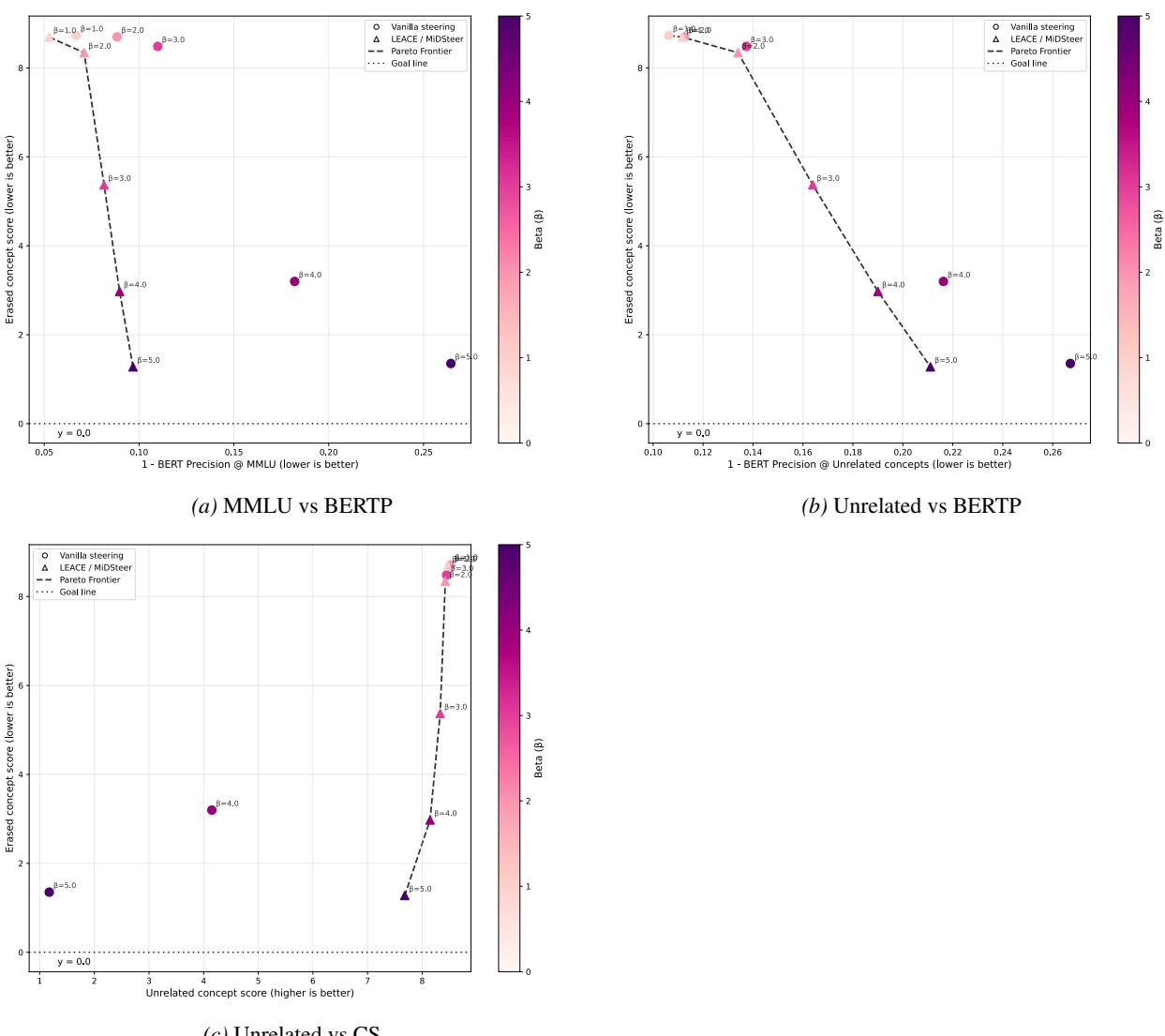

*(a)* MMLU vs BERTP

*(b)* Unrelated vs BERTP

*(c)* Unrelated vs CS

*Figure 23.* Pareto plot for concept erasure on model qwen-14b

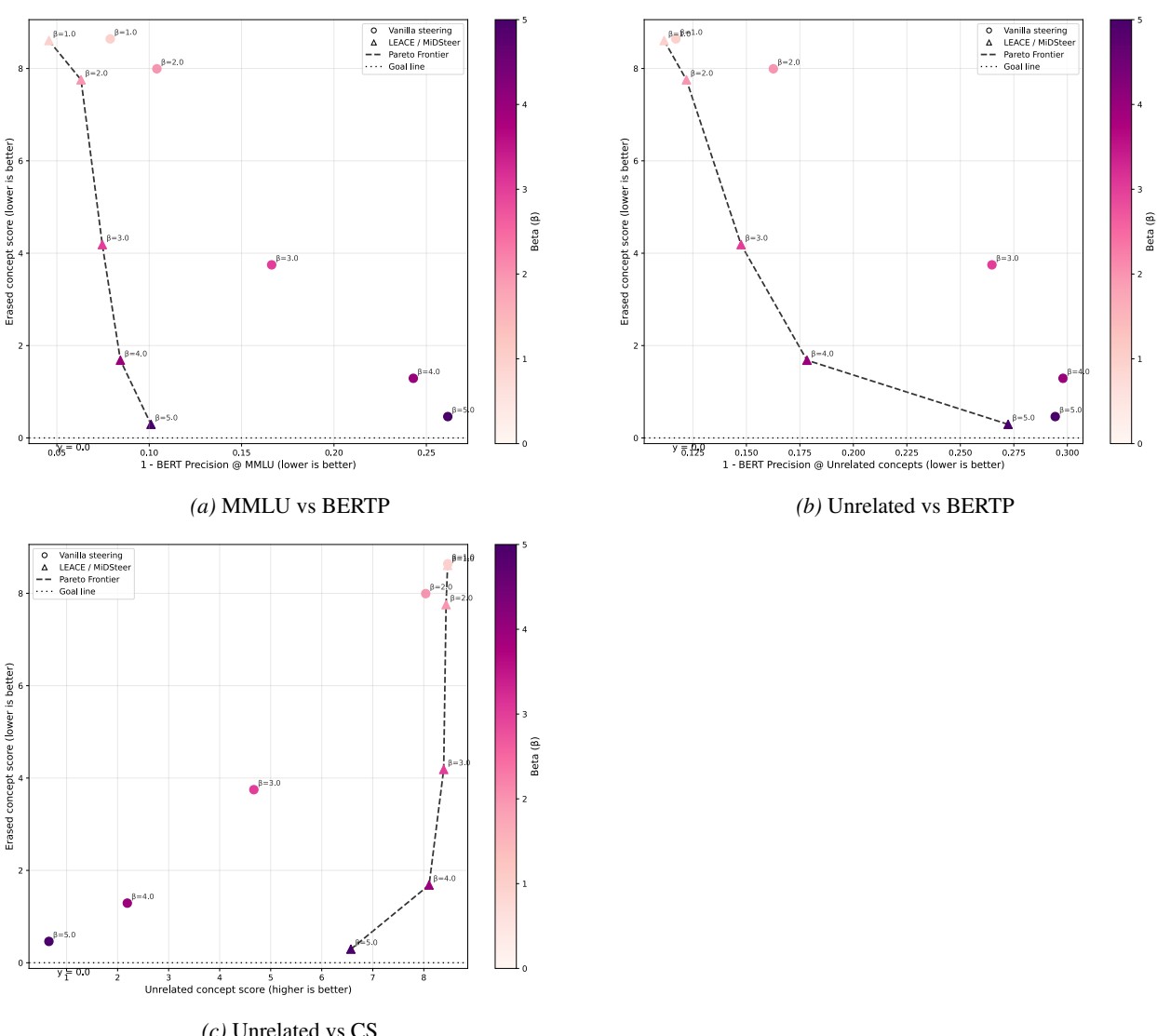

*(a)* MMLU vs BERTP

*(b)* Unrelated vs BERTP

*(c)* Unrelated vs CS

*Figure 24.* Pareto plot for concept erasure on model qwen-7b

### L.2. Pareto charts for image diffusion concept erasure

In this section, we provide more Pareto charts for LLM concept erasure. When erasing concept $c_s$, for each Diffusion model we provide 2 types of Pareto plots:

- FID score for unrelated concepts (horizontal axis) vs Concept Score (CS) for the erased $c_s$ concept (vertical axis) (fig. 26b,26b)

- Average Concept Score (CS) for unrelated concepts $c_i$ (horizontal axis) vs Concept Score (CS) for the erased $c_s$ concept (vertical axis) (fig. 26a,25a)

In each case, we see clear superiority of LEACE/MidSteer over other steering approaches.

We additionally provide detailed breakdown of scores for all $\beta$ values and all concepts $c_s, c_i$ in the tables in sec. L.4

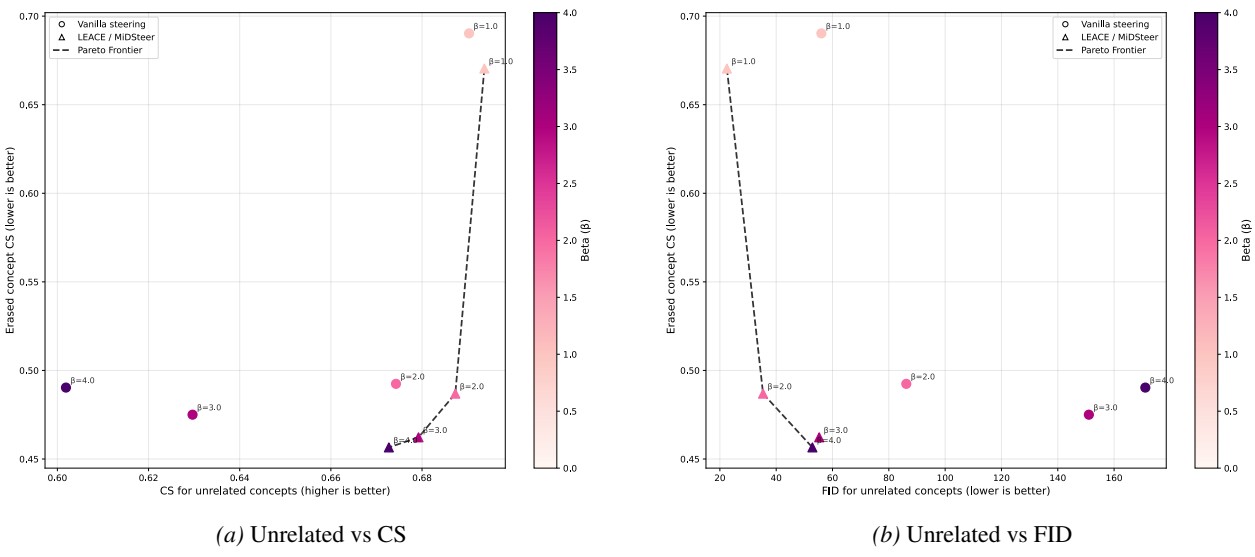

*(a)* Unrelated vs CS

*(b)* Unrelated vs FID

*Figure 25.* Pareto plot for concept erase on model sana

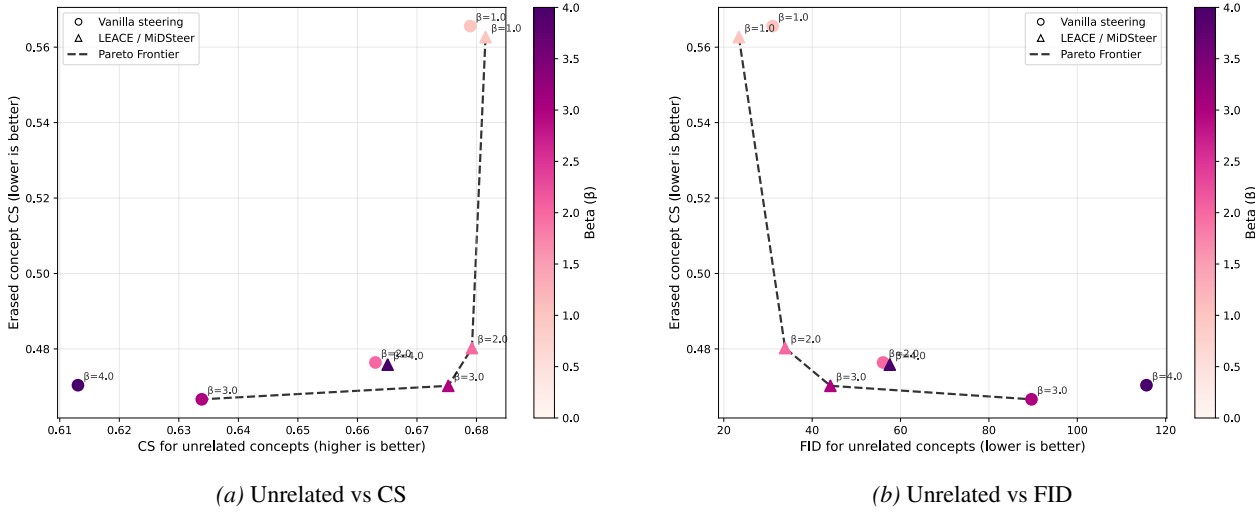

*(a)* Unrelated vs CS

*(b)* Unrelated vs FID

*Figure 26.* Pareto plot for concept erase on model sdxl

## L.3. Results for LLM concept erasure

In this section, in tab. 22,23,26,27,26,25we provide detailed breakdown of scores for all $\beta$ values and all concepts $c_s, c_i$. Pareto plots in sec. L.1 were created based on the scores provided in these tables.

*Table 22.* Model LLama2-7b, erasure of horses

| method | strength | horses cs ↓ | motorcycles cs ↑ | bertp ↑ | cows cs ↑ | bertp ↑ | pigs cs ↑ | bertp ↑ | dogs cs ↑ | bertp ↑ | legislators cs ↑ | bertp ↑ |
|---|---|---|---|---|---|---|---|---|---|---|---|---|
| No Steering | - | 8.6 | 8.5 | - | 8.4 | - | 8.5 | - | 8.7 | - | 8.4 | - |
| Steering | 1.0 | 8.6 | 8.5 | 0.91 | 8.4 | 0.90 | 8.4 | 0.90 | 8.6 | 0.90 | 8.5 | 0.89 |
| | 2.0 | 8.4 | 8.5 | 0.90 | 8.4 | 0.89 | 8.3 | 0.89 | 8.5 | 0.90 | 8.3 | 0.89 |
| | 3.0 | 1.5 | 3.4 | 0.80 | 2.4 | 0.78 | 1.4 | 0.78 | 2.9 | 0.79 | 2.6 | 0.79 |
| | 4.0 | 0.4 | 0.0 | 0.76 | 0.1 | 0.76 | 0.0 | 0.77 | 0.0 | 0.77 | 0.0 | 0.77 |
| | 5.0 | 1.0 | 0.0 | 0.76 | 0.1 | 0.77 | 0.0 | 0.77 | 0.0 | 0.77 | 0.0 | 0.77 |
| LEACE | 1.0 | 8.5 | 8.5 | 0.91 | 8.3 | 0.91 | 8.4 | 0.90 | 8.6 | 0.91 | 8.3 | 0.90 |
| | 2.0 | 5.0 | 8.3 | 0.89 | 7.3 | 0.87 | 7.4 | 0.88 | 8.0 | 0.89 | 7.8 | 0.88 |
| | 3.0 | 0.2 | 4.6 | 0.86 | 3.3 | 0.85 | 3.7 | 0.85 | 4.0 | 0.86 | 4.6 | 0.86 |
| | 4.0 | 0.0 | 1.0 | 0.82 | 0.2 | 0.81 | 0.2 | 0.81 | 0.4 | 0.82 | 1.1 | 0.82 |
| | 5.0 | 0.0 | 0.1 | 0.81 | 0.0 | 0.81 | 0.0 | 0.81 | 0.1 | 0.81 | 0.2 | 0.81 |

*Table 23.* Model LLama2-7b, erasure of dogs

| method | strength | dogs cs ↓ | cats cs ↑ | bertp ↑ | wolves cs ↑ | bertp ↑ | cows cs ↑ | bertp ↑ | pigs cs ↑ | bertp ↑ | legislators cs ↑ | bertp ↑ |
|---|---|---|---|---|---|---|---|---|---|---|---|---|
| No Steering | - | 8.6 | 8.6 | - | 8.5 | - | 8.4 | - | 8.4 | - | 8.5 | - |
| Steering | 1.0 | 8.6 | 8.5 | 0.90 | 8.4 | 0.90 | 8.4 | 0.90 | 8.4 | 0.90 | 8.5 | 0.90 |
| | 2.0 | 8.5 | 8.5 | 0.90 | 8.3 | 0.89 | 8.3 | 0.89 | 8.3 | 0.89 | 8.4 | 0.89 |
| | 3.0 | 2.1 | 3.1 | 0.79 | 3.6 | 0.80 | 3.0 | 0.79 | 1.9 | 0.78 | 3.4 | 0.79 |
| | 4.0 | 0.1 | 0.0 | 0.76 | 0.1 | 0.76 | 0.0 | 0.75 | 0.0 | 0.75 | 0.0 | 0.76 |
| | 5.0 | 0.3 | 0.0 | 0.76 | 0.0 | 0.75 | 0.0 | 0.76 | 0.0 | 0.76 | 0.0 | 0.75 |
| LEACE | 1.0 | 8.5 | 8.5 | 0.91 | 8.4 | 0.91 | 8.3 | 0.91 | 8.4 | 0.91 | 8.4 | 0.90 |
| | 2.0 | 6.5 | 7.8 | 0.88 | 7.6 | 0.88 | 7.2 | 0.88 | 7.5 | 0.88 | 7.7 | 0.88 |
| | 3.0 | 1.3 | 3.8 | 0.86 | 4.2 | 0.86 | 3.4 | 0.85 | 3.7 | 0.85 | 4.6 | 0.86 |
| | 4.0 | 0.1 | 0.4 | 0.82 | 0.2 | 0.83 | 0.5 | 0.82 | 0.3 | 0.82 | 1.0 | 0.82 |
| | 5.0 | 0.0 | 0.1 | 0.81 | 0.1 | 0.82 | 0.1 | 0.82 | 0.1 | 0.81 | 0.1 | 0.81 |

*Table 24.* Model Qwen2.5-7b, erasure of horses

| method | strength | horses cs ↓ | motorcycles cs ↑ | bertp ↑ | cows cs ↑ | bertp ↑ | pigs cs ↑ | bertp ↑ | dogs cs ↑ | bertp ↑ | legislators cs ↑ | bertp ↑ |
|---|---|---|---|---|---|---|---|---|---|---|---|---|
| No Steering | - | 8.7 | 8.6 | - | 8.4 | - | 8.4 | - | 8.7 | - | 8.5 | - |
| Steering | 1.0 | 8.6 | 8.5 | 0.89 | 8.4 | 0.88 | 8.4 | 0.88 | 8.7 | 0.88 | 8.6 | 0.88 |
|  | 2.0 | 8.0 | 7.8 | 0.84 | 8.1 | 0.84 | 7.8 | 0.83 | 8.1 | 0.83 | 8.1 | 0.83 |
|  | 3.0 | 3.6 | 4.0 | 0.72 | 5.1 | 0.73 | 4.5 | 0.73 | 4.7 | 0.73 | 3.9 | 0.72 |
|  | 4.0 | 1.2 | 2.8 | 0.69 | 2.6 | 0.70 | 1.9 | 0.70 | 2.6 | 0.70 | 2.0 | 0.69 |
|  | 5.0 | 0.6 | 1.0 | 0.70 | 0.7 | 0.70 | 0.3 | 0.70 | 0.7 | 0.70 | 0.9 | 0.69 |
| LEACE | 1.0 | 8.5 | 8.5 | 0.89 | 8.4 | 0.89 | 8.4 | 0.89 | 8.7 | 0.89 | 8.5 | 0.89 |
|  | 2.0 | 6.9 | 8.5 | 0.89 | 8.4 | 0.88 | 8.4 | 0.88 | 8.7 | 0.88 | 8.5 | 0.88 |
|  | 3.0 | 0.6 | 8.5 | 0.87 | 8.3 | 0.85 | 8.3 | 0.85 | 8.6 | 0.86 | 8.4 | 0.86 |
|  | 4.0 | 0.2 | 8.3 | 0.85 | 8.1 | 0.83 | 8.1 | 0.82 | 8.4 | 0.83 | 8.2 | 0.83 |
|  | 5.0 | 0.0 | 7.3 | 0.76 | 6.7 | 0.73 | 6.2 | 0.73 | 6.9 | 0.73 | 7.2 | 0.74 |

*Table 25.* Model Qwen2.5-7b, erasure of dogs

| method | strength | dogs cs ↓ | cats cs ↑ | bertp ↑ | wolves cs ↑ | bertp ↑ | cows cs ↑ | bertp ↑ | pigs cs ↑ | bertp ↑ | legislators cs ↑ | bertp ↑ |
|---|---|---|---|---|---|---|---|---|---|---|---|---|
| No Steering | - | 8.7 | 8.5 | - | 8.3 | - | 8.4 | - | 8.4 | - | 8.5 | - |
| Steering | 1.0 | 8.6 | 8.5 | 0.88 | 8.3 | 0.88 | 8.4 | 0.88 | 8.4 | 0.88 | 8.5 | 0.88 |
|  | 2.0 | 8.0 | 8.1 | 0.84 | 8.0 | 0.84 | 8.1 | 0.84 | 7.9 | 0.84 | 8.3 | 0.84 |
|  | 3.0 | 3.9 | 5.0 | 0.74 | 4.6 | 0.74 | 5.4 | 0.75 | 4.8 | 0.75 | 4.6 | 0.75 |
|  | 4.0 | 1.4 | 2.2 | 0.71 | 1.7 | 0.71 | 2.4 | 0.71 | 1.6 | 0.71 | 2.1 | 0.71 |
|  | 5.0 | 0.3 | 0.7 | 0.71 | 0.4 | 0.71 | 0.7 | 0.71 | 0.4 | 0.72 | 0.8 | 0.71 |
| LEACE | 1.0 | 8.7 | 8.5 | 0.89 | 8.4 | 0.88 | 8.4 | 0.89 | 8.4 | 0.88 | 8.5 | 0.89 |
|  | 2.0 | 8.7 | 8.5 | 0.88 | 8.3 | 0.87 | 8.4 | 0.88 | 8.4 | 0.87 | 8.5 | 0.88 |
|  | 3.0 | 7.8 | 8.5 | 0.85 | 8.2 | 0.84 | 8.3 | 0.85 | 8.3 | 0.85 | 8.4 | 0.86 |
|  | 4.0 | 3.2 | 8.2 | 0.83 | 7.7 | 0.79 | 8.1 | 0.83 | 7.9 | 0.81 | 8.2 | 0.82 |
|  | 5.0 | 0.6 | 6.5 | 0.72 | 5.6 | 0.71 | 6.6 | 0.72 | 5.9 | 0.72 | 6.9 | 0.72 |

*Table 26.* Model Qwen2.5-14b, erasure of horses

| method | strength | horses cs ↓ | motorcycles cs ↑ | bertp ↑ | cows cs ↑ | bertp ↑ | pigs cs ↑ | bertp ↑ | dogs cs ↑ | bertp ↑ | legislators cs ↑ | bertp ↑ |
|---|---|---|---|---|---|---|---|---|---|---|---|---|
| No Steering | - | 8.7 | 8.6 | - | 8.4 | - | 8.4 | - | 8.7 | - | 8.5 | - |
| Steering | 1.0 | 8.7 | 8.6 | 0.90 | 8.4 | 0.90 | 8.5 | 0.89 | 8.7 | 0.90 | 8.5 | 0.89 |
| | 2.0 | 8.7 | 8.6 | 0.89 | 8.4 | 0.89 | 8.5 | 0.89 | 8.7 | 0.89 | 8.5 | 0.89 |
| | 3.0 | 8.5 | 8.5 | 0.86 | 8.4 | 0.86 | 8.4 | 0.86 | 8.6 | 0.86 | 8.4 | 0.87 |
| | 4.0 | 2.9 | 4.2 | 0.78 | 5.0 | 0.78 | 5.0 | 0.78 | 4.3 | 0.79 | 4.3 | 0.79 |
| | 5.0 | 0.8 | 1.5 | 0.73 | 1.2 | 0.73 | 2.0 | 0.74 | 1.4 | 0.74 | 0.6 | 0.73 |
| LEACE | 1.0 | 8.7 | 8.6 | 0.90 | 8.4 | 0.89 | 8.4 | 0.88 | 8.7 | 0.89 | 8.5 | 0.88 |
| | 2.0 | 8.0 | 8.5 | 0.88 | 8.3 | 0.87 | 8.3 | 0.86 | 8.7 | 0.87 | 8.4 | 0.86 |
| | 3.0 | 3.0 | 8.5 | 0.85 | 8.3 | 0.84 | 8.3 | 0.83 | 8.6 | 0.84 | 8.3 | 0.83 |
| | 4.0 | 1.2 | 8.3 | 0.82 | 8.1 | 0.81 | 8.0 | 0.80 | 8.4 | 0.81 | 8.1 | 0.80 |
| | 5.0 | 0.3 | 7.8 | 0.79 | 7.5 | 0.79 | 7.5 | 0.78 | 7.9 | 0.79 | 7.7 | 0.79 |

*Table 27.* Model Qwen2.5-14b, erasure of dogs

| method | strength | dogs cs ↓ | cats cs ↑ | bertp ↑ | wolves cs ↑ | bertp ↑ | cows cs ↑ | bertp ↑ | pigs cs ↑ | bertp ↑ | legislators cs ↑ | bertp ↑ |
|---|---|---|---|---|---|---|---|---|---|---|---|---|
| No Steering | - | 8.7 | 8.5 | - | 8.4 | - | 8.4 | - | 8.5 | - | 8.5 | - |
| Steering | 1.0 | 8.7 | 8.5 | 0.90 | 8.4 | 0.89 | 8.5 | 0.89 | 8.5 | 0.89 | 8.6 | 0.89 |
| | 2.0 | 8.7 | 8.5 | 0.89 | 8.4 | 0.88 | 8.4 | 0.89 | 8.5 | 0.89 | 8.5 | 0.89 |
| | 3.0 | 8.4 | 8.5 | 0.86 | 8.3 | 0.86 | 8.4 | 0.86 | 8.4 | 0.86 | 8.4 | 0.87 |
| | 4.0 | 3.5 | 3.7 | 0.79 | 3.1 | 0.77 | 4.2 | 0.78 | 4.2 | 0.78 | 3.4 | 0.79 |
| | 5.0 | 1.9 | 0.9 | 0.73 | 1.6 | 0.73 | 0.9 | 0.73 | 1.2 | 0.73 | 0.4 | 0.73 |
| LEACE | 1.0 | 8.7 | 8.5 | 0.89 | 8.4 | 0.88 | 8.4 | 0.89 | 8.4 | 0.89 | 8.5 | 0.89 |
| | 2.0 | 8.7 | 8.4 | 0.87 | 8.4 | 0.86 | 8.3 | 0.87 | 8.4 | 0.86 | 8.4 | 0.86 |
| | 3.0 | 7.7 | 8.4 | 0.84 | 8.2 | 0.83 | 8.3 | 0.84 | 8.2 | 0.83 | 8.3 | 0.83 |
| | 4.0 | 4.7 | 8.3 | 0.82 | 7.9 | 0.81 | 8.2 | 0.82 | 8.0 | 0.80 | 8.1 | 0.80 |
| | 5.0 | 2.3 | 7.9 | 0.79 | 7.5 | 0.79 | 7.7 | 0.80 | 7.5 | 0.78 | 7.7 | 0.79 |

## L.4. Results for image diffusion concept erasure

In this section, in tab. 30,33,29,33 we provide detailed breakdown of scores for all $\beta$ values and all concepts $c_s, c_i$. Pareto plots in sec. L.2 were created based on the scores provided in these tables.

*Table 28.* Model SDXL, erasure of snoopy

| method | strength | snoopy cs ↓ | mickey cs ↑ | fid ↓ | pikachu cs ↑ | fid ↓ | spongebob cs ↑ | fid ↓ | dog cs ↑ | fid ↓ | legislator cs ↑ | fid ↓ |
|---|---|---|---|---|---|---|---|---|---|---|---|---|
| No Steering | - | 74.3 | 73.1 | - | 72.6 | - | 75.1 | - | 66.3 | - | 60.8 | - |
| CASteer | 1.0 | 55.8 | 70.1 | 54.9 | 72.5 | 30.3 | 73.9 | 50.9 | 66.2 | 30.6 | 60.9 | 22.6 |
|  | 1.5 | 49.9 | 67.9 | 71.8 | 72.5 | 39.9 | 72.8 | 66.0 | 66.2 | 39.4 | 60.9 | 27.5 |
|  | 2.0 | 47.0 | 65.2 | 90.5 | 72.6 | 51.2 | 71.0 | 85.2 | 66.2 | 48.1 | 60.8 | 31.7 |
|  | 2.5 | 45.6 | 62.2 | 111.0 | 72.5 | 65.7 | 68.6 | 109.4 | 66.1 | 58.1 | 60.9 | 35.3 |
|  | 3.0 | 45.3 | 58.8 | 132.1 | 72.2 | 83.7 | 65.3 | 138.2 | 66.2 | 68.0 | 60.8 | 38.7 |
|  | 4.0 | 45.3 | 53.5 | 169.0 | 71.6 | 123.4 | 59.0 | 189.5 | 66.1 | 83.3 | 60.9 | 45.8 |
|  | 5.0 | 45.9 | 50.7 | 195.3 | 69.3 | 153.0 | 55.7 | 218.2 | 65.6 | 99.9 | 61.0 | 52.9 |
| LEACE | 1.0 | 56.7 | 72.2 | 35.7 | 72.9 | 21.3 | 74.1 | 42.2 | 66.3 | 20.7 | 60.6 | 26.9 |
|  | 1.5 | 51.2 | 71.7 | 42.3 | 73.0 | 25.8 | 73.7 | 50.1 | 66.3 | 26.5 | 60.5 | 32.5 |
|  | 2.0 | 48.3 | 71.0 | 48.2 | 73.2 | 29.6 | 73.3 | 57.8 | 66.3 | 31.6 | 60.4 | 36.4 |
|  | 2.5 | 46.5 | 70.3 | 53.9 | 73.3 | 33.0 | 72.8 | 66.5 | 66.4 | 36.5 | 60.2 | 41.3 |
|  | 3.0 | 45.8 | 69.6 | 59.8 | 73.5 | 36.7 | 72.1 | 75.3 | 66.4 | 40.8 | 60.0 | 46.5 |
|  | 4.0 | 45.8 | 67.9 | 72.2 | 73.7 | 44.8 | 70.8 | 91.8 | 66.5 | 49.5 | 59.4 | 56.1 |
|  | 5.0 | 47.0 | 66.1 | 85.9 | 73.7 | 53.8 | 69.4 | 114.1 | 66.4 | 57.1 | 58.6 | 69.2 |

*Table 29.* Model SDXL, erasure of chihuahua

| method | strength | chihuahua cs ↓ | muffin cs ↑ | fid ↓ | dog cs ↑ | fid ↓ | wolf cs ↑ | fid ↓ | cat cs ↑ | fid ↓ | legislator cs ↑ | fid ↓ |
|---|---|---|---|---|---|---|---|---|---|---|---|---|
| No Steering | - | 75.9 | 68.2 | - | 66.3 | - | 71.8 | - | 67.5 | - | 60.8 | - |
| CASteer | 1.0 | 54.6 | 68.1 | 19.7 | 65.0 | 58.2 | 72.5 | 25.9 | 67.0 | 35.2 | 60.9 | 22.7 |
|  | 1.5 | 48.5 | 68.2 | 24.0 | 61.2 | 99.9 | 72.6 | 34.0 | 66.5 | 48.9 | 60.8 | 27.8 |
|  | 2.0 | 47.6 | 68.0 | 27.3 | 54.1 | 155.5 | 72.6 | 44.1 | 64.6 | 69.0 | 60.9 | 31.9 |
|  | 2.5 | 47.2 | 67.9 | 31.1 | 50.7 | 177.8 | 72.2 | 61.2 | 60.5 | 102.6 | 60.8 | 35.8 |
|  | 3.0 | 46.9 | 67.9 | 34.6 | 49.7 | 187.7 | 70.0 | 96.2 | 55.8 | 141.5 | 60.8 | 39.4 |
|  | 4.0 | 47.8 | 67.7 | 42.2 | 49.0 | 198.2 | 62.2 | 191.6 | 50.7 | 186.3 | 60.7 | 45.7 |
|  | 5.0 | 49.7 | 67.6 | 49.5 | 48.9 | 209.3 | 57.8 | 228.4 | 49.4 | 201.5 | 60.7 | 52.1 |
| LEACE | 1.0 | 55.0 | 68.2 | 20.0 | 65.8 | 35.2 | 72.3 | 17.1 | 67.4 | 22.1 | 60.9 | 21.8 |
|  | 1.5 | 48.5 | 68.1 | 25.0 | 65.5 | 47.6 | 72.5 | 21.3 | 67.3 | 27.1 | 60.8 | 27.0 |
|  | 2.0 | 47.4 | 68.1 | 29.0 | 65.0 | 61.1 | 72.6 | 25.0 | 67.3 | 31.4 | 60.9 | 31.0 |
|  | 2.5 | 47.0 | 68.2 | 32.6 | 64.1 | 74.8 | 72.8 | 28.6 | 67.2 | 35.1 | 60.8 | 34.2 |
|  | 3.0 | 47.2 | 68.1 | 36.0 | 62.7 | 92.4 | 72.9 | 32.4 | 67.1 | 38.5 | 60.8 | 36.5 |
|  | 4.0 | 48.6 | 68.0 | 42.3 | 57.6 | 131.4 | 73.1 | 39.4 | 66.9 | 45.1 | 60.7 | 41.9 |
|  | 5.0 | 50.2 | 68.0 | 49.4 | 53.7 | 162.6 | 73.1 | 48.3 | 66.5 | 52.5 | 60.5 | 48.3 |

*Table 30.* Model SDXL, erasure of horse

| method | strength | horse cs ↓ | motorcycle cs ↑ | fid ↓ | cow cs ↑ | fid ↓ | pig cs ↑ | fid ↓ | dog cs ↑ | fid ↓ | legislator cs ↑ | fid ↓ |
|---|---|---|---|---|---|---|---|---|---|---|---|---|
| No Steering | - | 71.0 | 70.7 | - | 72.7 | - | 71.8 | - | 66.3 | - | 60.8 | - |
| CASteer | 1.0 | 59.3 | 70.7 | 12.9 | 71.9 | 30.1 | 71.8 | 20.8 | 65.9 | 29.9 | 61.0 | 21.3 |
| | 1.5 | 49.8 | 70.7 | 15.6 | 71.2 | 46.6 | 71.8 | 27.6 | 65.8 | 36.5 | 61.1 | 26.2 |
| | 2.0 | 48.3 | 70.6 | 17.4 | 69.2 | 79.8 | 71.9 | 36.5 | 65.7 | 42.2 | 61.0 | 30.3 |
| | 2.5 | 47.9 | 70.7 | 19.5 | 62.1 | 152.5 | 72.0 | 45.9 | 65.4 | 48.4 | 60.9 | 33.6 |
| | 3.0 | 47.8 | 70.7 | 21.7 | 54.7 | 211.1 | 72.0 | 60.1 | 65.0 | 54.7 | 60.9 | 37.0 |
| | 4.0 | 48.0 | 70.7 | 26.9 | 51.1 | 227.9 | 71.8 | 92.3 | 63.9 | 69.4 | 60.8 | 43.3 |
| | 5.0 | 49.3 | 70.8 | 35.1 | 50.4 | 238.4 | 69.8 | 138.4 | 62.2 | 89.4 | 60.6 | 49.4 |
| LEACE | 1.0 | 57.1 | 70.6 | 11.5 | 72.3 | 20.5 | 71.8 | 11.6 | 66.1 | 19.7 | 60.7 | 25.1 |
| | 1.5 | 49.6 | 70.6 | 14.0 | 72.0 | 26.0 | 71.9 | 14.1 | 66.1 | 23.7 | 60.5 | 29.6 |
| | 2.0 | 48.4 | 70.6 | 15.9 | 71.8 | 33.1 | 71.9 | 16.2 | 66.0 | 28.0 | 60.4 | 34.1 |
| | 2.5 | 48.0 | 70.6 | 17.5 | 71.4 | 39.9 | 72.0 | 17.3 | 66.1 | 30.9 | 60.3 | 38.2 |
| | 3.0 | 48.1 | 70.6 | 19.3 | 70.7 | 53.5 | 72.0 | 19.1 | 66.1 | 33.6 | 60.3 | 41.6 |
| | 4.0 | 48.4 | 70.5 | 23.0 | 64.5 | 115.1 | 72.1 | 23.1 | 66.1 | 38.6 | 59.9 | 49.2 |
| | 5.0 | 49.7 | 70.3 | 27.4 | 56.4 | 197.4 | 72.2 | 27.8 | 66.1 | 44.6 | 59.4 | 58.8 |

*Table 31.* Model SDXL, erasure of horse

| method | strength | horse cs ↓ | motorcycle cs ↑ | fid ↓ | cow cs ↑ | fid ↓ | pig cs ↑ | fid ↓ | dog cs ↑ | fid ↓ | legislator cs ↑ | fid ↓ |
|---|---|---|---|---|---|---|---|---|---|---|---|---|
| No Steering | - | 72.1 | 70.5 | - | 73.8 | - | 73.5 | - | 68.1 | - | 60.4 | - |
| CASteer | 1.0 | 70.8 | 70.1 | 21.3 | 74.1 | 36.4 | 73.7 | 28.2 | 67.8 | 29.2 | 60.2 | 21.2 |
| | 2.0 | 52.2 | 70.9 | 45.3 | 72.0 | 93.0 | 74.1 | 49.3 | 67.4 | 44.2 | 59.8 | 36.3 |
| | 3.0 | 51.3 | 69.3 | 105.0 | 61.5 | 216.9 | 65.9 | 261.3 | 65.5 | 71.9 | 59.3 | 58.4 |
| | 4.0 | 56.7 | 62.3 | 186.2 | 59.0 | 242.8 | 67.3 | 175.3 | 62.5 | 118.4 | 58.9 | 90.1 |
| | 5.0 | 52.5 | 60.4 | 221.0 | 58.7 | 249.9 | 65.0 | 205.0 | 59.3 | 161.3 | 58.4 | 130.3 |
| LEACE | 1.0 | 71.1 | 70.5 | 7.5 | 73.8 | 14.4 | 73.4 | 11.4 | 68.0 | 9.2 | 60.4 | 11.1 |
| | 2.0 | 52.0 | 70.4 | 10.0 | 73.9 | 20.9 | 73.5 | 16.7 | 68.0 | 14.2 | 60.4 | 15.6 |
| | 3.0 | 49.7 | 70.3 | 12.2 | 74.0 | 25.8 | 73.5 | 21.4 | 67.9 | 18.4 | 60.3 | 19.1 |
| | 4.0 | 48.8 | 70.4 | 13.8 | 74.2 | 30.9 | 73.5 | 25.6 | 67.8 | 21.3 | 60.2 | 21.8 |
| | 5.0 | 53.7 | 70.3 | 15.2 | 74.1 | 36.2 | 73.6 | 29.7 | 67.8 | 24.4 | 60.1 | 24.2 |

*Table 32.* Model SANA, erasure of snoopy

| method | strength | snoopy cs ↓ | mickey cs ↑ | fid ↓ | pikachu cs ↑ | fid ↓ | spongebob cs ↑ | fid ↓ | dog cs ↑ | fid ↓ | legislator cs ↑ | fid ↓ |
|---|---|---|---|---|---|---|---|---|---|---|---|---|
| No Steering | - | 79.7 | 76.1 | - | 74.0 | - | 79.0 | - | 68.1 | - | 60.4 | - |
| CASteer | 1.0 | 60.6 | 75.3 | 64.0 | 74.1 | 41.3 | 79.0 | 43.9 | 68.0 | 42.1 | 60.8 | 23.3 |
| | 2.0 | 46.0 | 70.5 | 168.3 | 74.3 | 103.6 | 74.7 | 146.6 | 68.0 | 74.3 | 61.1 | 38.2 |
| | 3.0 | 42.4 | 64.2 | 189.7 | 72.0 | 164.1 | 63.4 | 222.2 | 67.7 | 100.9 | 60.8 | 55.3 |
| | 4.0 | 40.9 | 58.5 | 202.0 | 62.6 | 204.2 | 55.7 | 258.7 | 66.8 | 116.9 | 60.4 | 74.8 |
| | 5.0 | 40.9 | 55.4 | 208.1 | 55.5 | 231.6 | 52.9 | 276.5 | 65.1 | 127.6 | 60.0 | 94.8 |
| LEACE | 1.0 | 57.0 | 76.1 | 18.2 | 74.1 | 6.7 | 79.0 | 13.9 | 68.1 | 17.3 | 60.3 | 9.1 |
| | 2.0 | 44.8 | 76.2 | 30.5 | 74.1 | 11.7 | 78.9 | 19.3 | 68.1 | 25.3 | 60.2 | 13.6 |
| | 3.0 | 41.6 | 76.1 | 49.0 | 74.2 | 16.4 | 75.2 | 200.5 | 68.0 | 32.2 | 60.2 | 16.8 |
| | 4.0 | 40.9 | 75.6 | 73.4 | 74.2 | 21.3 | 78.7 | 29.0 | 68.0 | 38.0 | 60.1 | 19.5 |
| | 5.0 | 41.4 | 74.4 | 109.4 | 74.2 | 26.1 | 78.7 | 34.1 | 68.1 | 44.1 | 60.0 | 22.0 |

*Table 33.* Model SANA, erasure of chihuahua

| method | strength | chihuahua cs ↓ | muffin cs ↑ | fid ↓ | dog cs ↑ | fid ↓ | wolf cs ↑ | fid ↓ | cat cs ↑ | fid ↓ | legislator cs ↑ | fid ↓ |
|---|---|---|---|---|---|---|---|---|---|---|---|---|
| No Steering | - | 76.4 | 66.3 | - | 68.1 | - | 73.2 | - | 68.5 | - | 60.4 | - |
| CASteer | 1.0 | 75.6 | 66.6 | 19.8 | 67.4 | 49.4 | 73.6 | 25.6 | 68.3 | 32.5 | 55.3 | 268.2 |
| | 2.0 | 49.5 | 66.8 | 30.8 | 59.9 | 143.6 | 73.4 | 53.4 | 66.4 | 65.2 | 60.5 | 33.7 |
| | 3.0 | 48.9 | 67.1 | 44.1 | 52.6 | 214.4 | 64.6 | 265.3 | 58.6 | 151.2 | 60.4 | 48.6 |
| | 4.0 | 49.5 | 67.0 | 58.0 | 52.3 | 223.8 | 62.0 | 263.8 | 54.6 | 205.0 | 60.2 | 71.9 |
| | 5.0 | 50.4 | 66.3 | 76.7 | 53.2 | 233.5 | 59.8 | 282.5 | 53.6 | 220.9 | 59.8 | 102.3 |
| LEACE | 1.0 | 73.0 | 66.3 | 5.8 | 68.0 | 28.1 | 73.2 | 6.4 | 68.5 | 10.4 | 60.4 | 9.9 |
| | 2.0 | 49.3 | 66.2 | 9.7 | 67.8 | 47.1 | 73.3 | 9.7 | 68.5 | 16.0 | 60.4 | 14.7 |
| | 3.0 | 47.3 | 66.2 | 12.6 | 67.6 | 68.1 | 73.3 | 12.3 | 68.6 | 20.6 | 60.4 | 17.9 |
| | 4.0 | 47.2 | 66.1 | 15.2 | 67.1 | 88.7 | 73.3 | 14.7 | 68.6 | 24.6 | 60.3 | 20.9 |
| | 5.0 | 48.5 | 66.1 | 17.7 | 66.2 | 113.6 | 73.3 | 16.8 | 68.7 | 27.5 | 60.3 | 23.0 |

