# OpenReview forum: "MidSteer: Optimal Affine Framework for Steering Generative Models"
_ICML.cc/2026/Conference — ICML 2026 regular_

### Official Review · Reviewer_UtYM · 2026-03-13

**Soundness:** 3
**Presentation:** 3
**Significance:** 3
**Originality:** 3
**Overall Recommendation:** 5
**Confidence:** 3

**Summary:**

The authors present a theoretically well-grounded technique for steering the internal representations of neural networks. Building on the LEACE framework, they propose MidSteer, which operates on the principle of intervening on the covariances between the concepts at hand while minimizing the change to the representations at hand. This method demonstrates impressive empirical results across both textual and visual applications on a proposed benchmark for evaluating concept switching, a core manipulation technique it enables.

**Compliance With Llm Reviewing Policy:**

Affirmed.

**Final Justification:**

My concerns are addressed and I maintain my score.

**Key Questions For Authors:**

In Eq 22, what happens for $\beta \in (1,2)$?

**Limitations:**

A very mild limitation is the limited number of concepts that are manipulated in the empirical results

**Strengths And Weaknesses:**

**Soundness.** I found the empirical validation to be somewhat lacking, though given the theory present, this isn’t a major issue. The core issue is that only a few concepts are interacted with.

**Presentation.** Overall I found the paper readable, but it could be improved. There are a number of mathematical details that feel out of place. For example, it’s not entirely clear to me what the significance of guardedness is; the desiderata of controlling the covariance seems clear enough. Nevertheless, the overall narrative can be followed, and the contrast with prior work is clear.

**Significance.** I assess this as a strength. Manipulating model’s internal representations is an important area of research, and this seems to be a new approach with well-demonstrated benefits. It feels natural to seriously consider adopting it.

**Originality.** I rate this as a strength. The differences from prior work are clearly articulated, and the method seems novel.

---

> ### Author Rebuttal · Authors · 2026-03-30
>
> We thank the reviewer for **the positive assessment of our paper’s significance, originality, and theoretical soundness**. Below we address the reviewer’s main concerns.
>
> **On guardedness**
>
> We agree that the covariance-control objective is the clearest way to understand the method. Guardedness is included primarily to connect our framework to prior theoretical work on concept erasure, not because it is strictly necessary for following the main construction. We will revise the presentation to make this hierarchy clearer.
>
> **On β ∈ (1,2)**
>
> In Eq. 22, β interpolates between erasure and full switching: β=1 gives erasure, and β=2 gives full switching. For β ∈ (1,2), the concept-mediated component is sign-inverted but reduced in magnitude, so this corresponds to a partial / soft switch. Empirically, this provides a trade-off between switch strength and preservation of unrelated content. We also evaluate this intermediate regime in the Appendix (Tabs. 2, 3, 8, 9, 10), where β=1.5 behaves as expected between erasure-strength and full-switch interventions.
>
> **On a limited number of evaluation concepts**
>
> We agree that the empirical section can be strengthened with a broader range of concepts. To address this, we conducted additional experiments on safety-related abstract concepts across both LLMs and diffusion models, and the conclusions remain consistent with those for nominal concepts: **MidSteer achieves stronger target manipulation while causing smaller changes to unrelated concepts**. These additional results strengthen the empirical evidence that the benefits of MidSteer extend to abstract and safety-relevant interventions as well.
>
> Due to space constraints, we report only a subset of these results in the rebuttal. In the revised paper, we will add the full set of experiments, including Pareto plots and detailed tables for all models and concept pairs.
>
> **LLM: Toxicity → Helpfulness**
>
> We compute cross-covariances from 1,000 toxic comments (Jigsaw Toxic Comment Classification) and 1,000 helpful responses (Anthropic HH-RLHF). We evaluate toxicity removal on 500 RealToxicityPrompts using Detoxify. We also track unrelated concepts (“sarcasm”, “creativity”, “politeness”, “mathematics”) and consistency on Alpaca / MMLU as in the main paper.
>
> We report the main results on removing toxicity and preserving helpfulness on the Llama-2-7b model in Tab.1 and Tab.2. On toxicity prompts, MidSteer reduces toxicity monotonically and substantially outperforms both LEACE and vanilla steering. On helpfulness prompts, MidSteer introduces essentially no toxicity leakage, whereas vanilla steering and LEACE at high β introduce substantial leakage into target prompts.
>
> **Table 1. Toxicity score on RealToxicityPrompts (↓)**
>
> | Baseline | Steering β=3 | Steering β=5 | LEACE β=3 | LEACE β=5 | MidSteer β=3 | MidSteer β=5 |
> |---|---:|---:|---:|---:|---:|---:|
> | 0.371 | 0.369 | 0.380 | 0.344 | 0.346 | **0.309** | **0.281** |
>
> **Table 2. Toxicity score on “helpfulness” prompts (↓)**
>
> | Baseline | Steering β=3 | Steering β=5 | LEACE β=3 | LEACE β=5 | MidSteer β=3 | MidSteer β=5 |
> |---|---:|---:|---:|---:|---:|---:|
> | 0.0008 | 0.0011 | 0.0199 | 0.0020 | 0.0020 | **0.0007** | **0.0008** |
>
> We also observe that MidSteer and LEACE preserve output consistency on MMLU and Alpaca much better than vanilla steering at high strengths: (score > 0.94 for all β values for MidSteer/LEACE , vanilla steering degrades to 0.85 at β=5). Then, MidSteer introduces no leakage of source concept into unrelated prompts. We observe similar behaviour for Qwen2.5-14B.
>
> **Diffusion: Violence → Peace**
>
> We follow the diffusion setup from the paper and use four unrelated concepts: “sadness”, “calm”, “anger”, and “nature”.
>
> We present the Main results on removing violence and preserving peace concepts on SANA in Tab.3 and Tab.4. MidSteer achieves near-complete violence removal at β=3 (9.4%), while CASteer/LEACE need β=5 to reach 30%/14.4%. At the same time, MidSteer does not introduce violence into peace prompts. In contrast, both CASteer and LEACE substantially increase violence on peace prompts at higher strengths.
>
> **Table 3. CLIP violence accuracy on “violence” prompts (↓)**
> | Baseline | CASteer β=3 | CASteer β=5 | LEACE β=3 | LEACE β=5 | MidSteer β=3 | MidSteer β=5 |
> |---|---:|---:|---:|---:|---:|---:|
> | 98.8% | 74.6% | 30.0% | 60.1% | 14.4% | **9.4%** | **1.9%** |
>
> **Table 4. CLIP violence accuracy on “peace” prompts (↓)**
> | Baseline | CASteer β=3 | CASteer β=5 | LEACE β=3 | LEACE β=5 | MidSteer β=3 | MidSteer β=5 |
> |---|---:|---:|---:|---:|---:|---:|
> | 18.0% | 70.8% | 80.2% | 89.4% | 96.8% | **2.8%** | **0.8%** |
>
> On unrelated concepts, unlike MidSteer, CASteer and LEACE can also introduce violence where none existed. CASteer increases violence in calm (8→23%) and nature (2→15%) prompts at high β, LEACE also slightly  increases violence in “calm” prompts (8→10%). MidSteer reduces violence consistently. We observe similar behaviour for SDXL as well.

---

> > ### Author Rebuttal · Reviewer_UtYM · 2026-04-04
> >
> > Thank you, this addresses my concerns.

---

### Official Review · Reviewer_2Zz6 · 2026-03-23

**Soundness:** 3
**Presentation:** 4
**Significance:** 3
**Originality:** 3
**Overall Recommendation:** 5
**Confidence:** 3

**Summary:**

The paper formalizes the task of concept switching, at the crossroads of affine erasure and generative model steering of intermediate representation states. It builds upon the LEACE method to demonstrate steering is a subcase of LEACE and to extend it to concept inversion in LEACE-Switch before providing closed-form solution for concept switching while relaxing the most important restrictions of LEACE-Switch, the requirement for complete label inversion and the needs for concepts which partition the full dataset. This new framework, MIDSTEER is then applied on LLM and Diffusion models to demonstrate and evaluate the method on textual and visual data across three concept pairs.

**Compliance With Llm Reviewing Policy:**

Affirmed.

**Final Justification:**

The rebuttal adresses my main concerns :
- the second LLM judge (GPT-4o-mini) appears to confirm the relative ranking of methods.
- the authors respond to the concern about abstract and safety-relevant concepts with additional experiments.
- impact statement will be rework to include both benefits and risks of sucessful steering.
- the reason for large number of prompts for Σ_{XX} is robustness
- the trade-off between offline covariance estimation and zero inference-time overhead once the affine transform is folded into model weights is clearer and better than previously though, allowing for relatively fast inference of new concepts.
- The clarification regarding the full-rank assumption in Theorem 4.4 is helpful and reasonable.

The three first points and the need for more quotes in the two first parts of the papers are linked to a revised manuscript actually including the needed revisions, but the rebuttal resolves enough of my original uncertainty to justify increasing my rating from 4 to 5.

**Key Questions For Authors:**

1) In the ablation study (p.24, Appendix H), "increasing the number of prompts used beyond 5000 has limited impact," yet the main results are obtained using M = 50,000 prompts for sigma_XX estimation (p.6, Sec. 5.1). Is this a deliberate conservative choice, or could the main experiments use fewer samples without degradation? If so, what motivated the 10× gap?

2) Concerning the cost of using MIDSTEER: you mention in Section 3.1 (last paragraph, p.3) that both Eq. 3 and Eq. 4 can be integrated without inference overhead, but on the other hand LEACE-Switch and MIDSTEER need thousands of samples to estimate covariance matrices per concept (N = 1,000 for class-conditional, M = 50,000 for self-covariances). Wouldn't this offline cost limit the practical gains of zero inference overhead, especially if the method needs to be re-estimated for new concepts?

3) Concerning LLM steering evaluation: are the concept scores from a single LLM judge (Llama-3.1-8B-Instruct) trustworthy enough without cross-validation by a second judge, a semantic analysis, or human evaluation? Have you observed any failure modes of this judge?

4) Theorem 4.4 requires rk(sigma_XZ_1) = l (full column rank). How frequently is this assumption verified in practice across layers and concepts? RMSNorm layers in modern LLMs (used in both Llama-2 and Qwen-2.5) constrain activation norms, which could affect the rank structure of cross-covariance matrices. Do you observe any rank-deficiency issues, and if so, how do they impact the method's applicability?

**Limitations:**

No — the paper does not adequately discuss limitations. Specifically: (a) the offline computational cost of covariance estimation (N = 1,000 class-conditional samples, M = 50,000 general samples, per concept pair, per layer) is not acknowledged as a limitation despite the paper emphasizing zero inference overhead (Sec. 3.1, Sec. 4.2.1), creating a potentially misleading picture of the method's total cost; (b) no experiments are conducted on abstract or safety-relevant concepts (e.g., toxicity, helpfulness) despite the introduction's framing around harmful behavior suppression (p.1); (c) the impact statement (p.9) is generic and does not engage with the paper's own motivating claims about safety and alignment.

**Strengths And Weaknesses:**

Strengths and Weaknesses :
Strengths :
1) Overall clarity and scope - Major : The paper is clear, well structured and the scope is well defined ans delimited. The progression from corollary 4.1 (p.4) through theorem 4.2 (p.5) to theorem 4.4 (p.6) is logical and easy to follow, each step relaxing assumptions from the previous one. The explicit discussion of assumptions at each step is welcome.
2) Objective and steps are clear and ambitious - Major : The paper clearly states what it brings (four bullet points, p.2): bridging two closely related fields, relaxing constraints to create a new framework that allows use in practice, and formalizing a much needed task (concept switching). The two key relaxations are made explicit in "Scope of LEACE-Switch" (p.5)
3) Related work clear + good fit of their work - Minor : The related work section (Sec. 2) shows well how this paper fits in the literature and why it matters. Positioning with respect to INLP, RLACE and SPLINCE is explicit.
4) intro of part 4 - Minor : Quick summary at the beginning of Sec. 4, helping to clarify what will be done in this part, allowing to better understand the important theorems more easily.
5) Improvement of baseline LEACE is clear - Major : Limitations of baseline LEACE and LEACE-Switch are clearly explained in "Scope of LEACE-Switch" (p.5), which helps to see how MIDSTEER relaxes these assumptions. The two specific limitations identified (dataset partitioning and forced symmetry) directly motivate theorem 4.4. MIDSTEER has been shown (Sec. 4.3, p.6) to fall back to LEACE for pure concept erasure.
6) Reproducibility - Minor : Models, pseudocode (Algorithm 1, Appendix A), prompts (Appendix E, F, G), and evaluation templates are available for reproducibility.

Weaknesses :
7) Second paragraph of introduction needs grounding - Minor : Several sentences in the second paragraph (p.1) assert facts without citing papers. For example, "its theoretical foundations remain underdeveloped" and "naive steering often perturbs unrelated features" would benefit from specific citations or a concrete example. This could be improved to help build confidence in the introduction.
8) Only 1 LLM-as-a-Judge - Major :The paper's results are obtained using Llama-3.1-8B-Instruct as the sole LLM judge (Sec. 5.1, p.8) to score concept presence on a 0–10 scale via argmax across token probabilities (Appendix F, p.19).  Using a single LLM to evaluate performances on a given prompt makes the results susceptible to be limited by the LLM juge capabilities and biases. Paper could be strengthened by the use of human as a judge even for a limited set of prompts.
9) Lack of abstract / difficult concepts switching - Major : The introduction motivates the work around "suppress harmful ones" and mentions "toxicity, nudity" (p.1), yet all three evaluation pairs((Horse,Motorcycle), (Dog,Cat), (Chihuahua,Muffin) (Sec. 5.1, p.7)) involve concrete visual/nominal concepts. Abstract concepts like (toxicity,helpfulness), (violence, peace) or (unsafe,safe), which are the paper's stated motivation, are never tested. The paper would benefit from applying its method to abstract concepts or at least discussing their feasibility. The limited number of pairs could also weaken representativeness of results.
10) Impact statement is vague - Minor : The impact statement (p.9) reads "There are many potential societal consequences of our work, none of which we feel must be specifically highlighted here." This is generic and does not engage with the safety-relevant framing of the introduction. It could benefit from being aligned with the introduction's claims about alignment and safety before opening up to other domains.

---

> ### Author Rebuttal · Authors · 2026-03-30
>
> We thank the reviewer for the thoughtful and detailed feedback. We are particularly encouraged by **the positive assessment of the paper’s clarity, structure, and theoretical progression**. Below we address the reviewer’s main concerns.
>
> **Grounding the introduction**  We agree that the second paragraph of the introduction would benefit from more explicit grounding. We will revise it by adding citations and concrete examples.
>
> **Single LLM as judge**
>
> We cross-validated our evaluation with GPT-4o-mini as a second independent judge, using the same prompt and 0–10 scoring protocol as in Appendix F. We evaluated all three concept pairs on Llama-2-7B-Chat for all the methods across β ∈ {1,2,3,4,5}. Due to rebuttal space limits, we report partial results in Tab. 1, and will include all the full cross-validation results in the revised paper. The rankings are consistent with those obtained using Llama-3.1-8B-Instruct: MidSteer achieves the highest target-concept scores, the lowest residual source-concept scores at moderate-to-high strengths, and preserves unrelated concepts.
>
> **Table 1. GPT-4o-mini judge — Concept Scores (Source CS / Target CS)**
> | Concept pair | Baseline | CASteer β=3 | LEACE β=3 | MidSteer β=3 | CASteer β=5 | LEACE β=5 | MidSteer β=5 |
> |---|---|---|---|---|---|---|---|
> | Horse → Motorcycle | 9.8 / 0.1 | 0.2 / 9.2 | 0.3 / 9.3 | 0.1 / 9.5 | 0.2 / 9.5 | 0.1 / 9.6 | 0.1 / 9.7 |
> | Dog → Cat | 9.7 / 0.1 | 2.4 / 8.0 | 3.2 / 7.0 | 1.0 / 8.9 | 1.0 / 9.1 | 1.7 / 8.2 | 0.1 / 9.7 |
> | Chihuahua → Muffin | 9.9 / 0.0 | 0.3 / 6.9 | 0.7 / 7.5 | 0.0 / 8.9 | 0.1 / 8.7 | 0.1 / 8.7 | 0.0 / 9.3 |
>
> We also emphasize that the judge is used primarily for relative comparison between methods, not to claim absolute ground-truth concept scores. The evaluation protocol is fixed, so the key question is whether the judge ranks the methods consistently.
>
> **Abstract/safety-relevant concepts**
>
> To address it, we conducted additional experiments on abstract and safety-relevant concepts across both LLMs and diffusion models. The conclusions are consistent with those in the main paper: MidSteer achieves stronger target manipulation with smaller changes to unrelated concepts than the baselines. Due to the rebuttal length limit, we refer the reviewer to our response to Reviewer UtYM for the detailed tables and discussion. We will include the full results in the revised version, and are happy to provide more results during the author-reviewer discussion.
>
> **The impact statement**
>
> We will revise the impact statement to better reflect the paper’s alignment and safety motivation. In particular, we will note both the positive potential of principled steering for post-deployment control of harmful or undesired behaviors, and the dual-use risks of manipulating model behavior or bypassing safeguards.
>
> **The number of prompts for Σ_{XX} estimation**
>
> This was a deliberate conservative choice. Appendix H shows that performance largely stabilizes around 5,000 prompts in the tested Llama-2-7B setting. We nevertheless used M=50,000 in the main experiments to make Σ_{XX} as stable as possible across models, layers, and concepts. Importantly, Σ_{XX} is concept-agnostic and computed only once per model, so this is not a per-concept bottleneck. This was also practical in our setup: estimating Σ_{XX} for 50,000 samples took under 15 minutes for Llama2-7b/SANA and under 30 minutes for SDXL/Qwen2.5-14B on a single H100 GPU.
>
> **Offline cost vs. inference overhead**
>
> Our claim of “zero inference overhead” refers specifically to deployment: once the affine transform is estimated, it can be folded into the projection weights, so generation runs at the same inference-time cost as the original model. Σ_{XX} is computed once per model, and only the cross-covariances are concept-specific. Cross-covariance estimation takes about one minute for 1000 prompts on a single H100 GPU.  Thus, the trade-off is one-time offline estimation per concept pair versus zero runtime overhead after. This is particularly attractive in repeated-use settings, such as model-level safety policies.
>
> **The full-rank assumption in Theorem 4.4**
>
> The full-rank assumption is used in the proof to derive the closed-form affine solution. In our setup, we consider only one source and one target concept, so l=1, and the condition reduces to Σ_{XZ_1} ≠ 0, i.e. the source concept has a nontrivial linear footprint at the chosen layer. RMSNorm does not by itself imply rank deficiency: the rank of Σ_{XZ_1} depends on whether the concept induces nonzero covariance directions across samples, not on the overall norm scale. If a concept has essentially no linear signature at a layer, the affine update becomes degenerate, so MidSteer would not be meaningfully applied there. In multi-concept settings, rank deficiency indicates linear redundancy among source concepts, in which case we can restrict to an independent subset or work in the effective subspace via the pseudoinverse.

---

> > ### Author Rebuttal · Reviewer_2Zz6 · 2026-04-06
> >
> > Thank you for answers and the clarification they have provided. I will increase the rating accordingly.

---

> > > ### Author Response · Authors · 2026-04-06
> > >
> > > We are glad to see that we have addressed all the reviewer's concerns and that they have increased their score. We believe that their review, followed by addressing those points, improved our paper. Of course, we will integrate all these changes in the eventual camera-ready version of the paper.

---

### Official Review · Reviewer_bFSx · 2026-03-24

**Soundness:** 3
**Presentation:** 3
**Significance:** 2
**Originality:** 3
**Overall Recommendation:** 4
**Confidence:** 3

**Summary:**

This paper focuses on steering the internal representations of generative models. The authors first formalize vanilla model steering as a special case of affine transformations. Building upon LEACE, an existing closed-form concept erasure technique, they introduce LEACE-Switch to fully utilize optimal affine transformations for concept swapping. However, this approach relies on the strict assumption that the source and target concepts perfectly partition the dataset. To overcome this limitation, they introduce MidSteer, a generalized framework that relaxes this constraint for directed, asymmetric switching.

They evaluated their methods on both large language models (e.g., Llama-2, Qwen) and vision diffusion models (e.g., SDXL, SANA) across various concept-switching tasks. The empirical results demonstrated that MidSteer consistently outperforms both vanilla steering and LEACE-Switch; it successfully executes targeted concept swaps while effectively minimizing the disturbance to orthogonal, unrelated features in the representation space, all without adding inference overhead.

**Compliance With Llm Reviewing Policy:**

Affirmed.

**Final Justification:**

My concerns are fully addressed.

**Key Questions For Authors:**

The connection between steering and LEACE (Corollary 4.1) relies on the assumption that representations are standardized (zero mean, identity covariance). How realistic is this assumption for actual intermediate representations in LLMs and diffusion models

**Limitations:**

Yes

**Strengths And Weaknesses:**

### Strengths
1. Formalizing the vanilla steering
2. Zero inference overhead
3. Reducing damage to orthogonal features.
4. Can be applied to most neural network architectures.

### Weaknesses
1. From mechanistic interpretability research, we know with some confidence that language models encode their features not orthogonal to each other. They encode features in a way that is mostly orthogonal but not fully orthogonal so that they can fit more features into their representation space. This paper assumes a linear encoding of features.
2. The paper claims there is no benchmark for concept editing. However, this is not current. There are many benchmarks designed for concept/knowledge editing. For example, AxBench [1]. Hence, the paper only evaluates using its own tasks.
2. All concept pairs tested involve concrete, visually or semantically distinct object categories (horse→motorcycle, dog→cat). I think you should considere evaluating on more abstract or safety-relevant concepts (e.g., toxicity, bias, truthfulness) where steering methods are arguably most needed.



[1]: Wu, Zhengxuan, et al. "Axbench: Steering llms? even simple baselines outperform sparse autoencoders." arXiv preprint arXiv:2501.17148 (2025).

---

> ### Author Rebuttal · Authors · 2026-03-30
>
> We thank the reviewer for the detailed feedback and are encouraged **by the positive assessment of the formalization, zero inference overhead, and preservation of unrelated features**. Below we address the reviewer’s main concerns.
>
> **On orthogonality**
>
> Our framework does not assume that concepts are mutually orthogonal in representation space. Orthogonality and linearity are distinct: concepts may be encoded along correlated or overlapping directions, yet still be well captured by linear or affine statistics. Our results operate in this affine regime, following prior work on linear concept erasure and activation steering. In particular, LEACE [1] gives an optimal closed-form linear erasure operator, and our paper extends this affine viewpoint to switching and more general concept manipulation. More broadly, prior work provides both empirical and theoretical evidence that approximate linear structure is often meaningful in LLM representations: simple linear directions can strongly control behavior [2], and the Linear Representation Hypothesis [3] relates such structure to both probing and steering. We do not claim that all concepts are exactly linear in all models; rather, we characterize the optimal affine intervention under this practically useful approximation.
>
> [1] Belrose, et al. "LEACE: Perfect linear concept erasure in closed form"
>
> [2] Arditi, et al. "Refusal in Language Models Is Mediated by a Single Direction"
>
> [3] Park, et al. "The Linear Representation Hypothesis and the Geometry of Large Language Models"
>
> **On benchmarks for concept editing**
>
> Our original wording was indeed broad. AxBench is an important benchmark for concept steering/editing in LLMs, and we will revise the paper accordingly. However, we note that AxBench does not directly evaluate the setting studied here: directed concept switching from a source concept c_s to a target concept c_t with minimal disturbance. Our evaluation explicitly checks (i) source-concept erasure, (ii) target-concept addition, and (iii) preservation of unrelated concepts. By contrast, AxBench is organized around introducing or suppressing individual concepts, rather than paired source-to-target switching.
>
> Furthermore, MidSteer is not designed for pure concept addition: Theorem 4.4 assumes that the source-concept cross-covariance has full column rank, which fails in the degenerate “no source concept” case. In contrast, for concept erasure (no target concept), MidSteer reduces to LEACE [1], as discussed in Sec. 4.3. For this special case, we additionally evaluated LEACE on AxBench’s concept-erasure task, following the official protocol on Gemma-2-2B. We present the results in Tab.1. LEACE achieves HM = 1.394, matching prompt-based suppression (1.396) and outperforming the best trained representation-steering baseline reported in [4], RePS SV r=1 (0.929). We will add this evaluation and clarify that, while steering/editing benchmarks do exist, to the best of our knowledge, there is no standard benchmark tailored to directed source-to-target concept switching.
>
> **Table 1. Concept suppression on AxBench**
> | Method | Type | Training | HM (↑) |
> |---|---|---|---:|
> | Prompt | System prompt instruction | None | 1.396 |
> | **LEACE** | **Analytical projection** | **None** | **1.394** |
> | Lang SV r=1 | DPO-trained vector | 18 epochs | 0.936 |
> | RePS SV r=1 | DPO-trained vector | 18 epochs | 0.929 |
>
> [4] Wu, et al. "Improved Representation Steering for Language Models"
>
> **On steering abstract concepts**.
>
> We agree that abstract and safety-relevant concepts are an important testbed. To address this, we conducted additional experiments on such concepts across both LLMs and diffusion models. The results support the same conclusion as in the main paper: MidSteer consistently achieves stronger target manipulation with smaller changes to unrelated concepts than the baselines. Due to the rebuttal length limit, we refer the reviewer to our response to Reviewer UtYM for the detailed tables and discussion. We will include the full results in the revised version. We are happy to provide more results during the author-reviewer discussion if the reviewer asks for them.
>
> **On standardized representations**
>
> Corollary 4.1 does not assume that intermediate representations in LLMs or diffusion models are already standardized. Rather, it shows that vanilla steering is a special case of LEACE when the representation has zero mean and identity covariance; in that setting, steering in erasure mode coincides exactly with LEACE. Hidden representations are not expected to satisfy these conditions, so the corollary should be read as an equivalence result, not a literal description of raw activations. More generally, Theorems 3.3, 4.2, and 4.4 show that the optimal affine transformations explicitly involve whitening to account for the anisotropic second-order structure of the representation space. This more realistic covariance-aware setting is precisely what our framework is designed for.

---

> > ### Author Rebuttal · Reviewer_bFSx · 2026-04-03
> >
> > Thank you for your response. All my concerns are addressed and I will increase my score.

---

> > > ### Author Response · Authors · 2026-04-04
> > >
> > > We thank the reviewer for appreciating our rebuttal and increasing their score. We believe that their review, followed by addressing those points, improved our paper. Of course, we will integrate all these changes in the eventual camera-ready version of the paper.

---

### Decision · Program_Chairs · 2026-04-30

**Decision:**

Accept (regular)

**Comment:**

This manuscript formalizes model steering by linking it to concept erasure and then introducing a conceptual framework for concept switching. They provide convincing empirical evidence in favor of their framework, on both LLMs and vision diffusion models.

Reviewers have aknowledge the strenghts of the proposed framework with respect to previous work and foudn the experiments convincing. Their concerns were address during the discussion phase and I therefore recommend acceptance.